# An analysis of the noise schedule for score-based generative models

Stanislas Strasman[†], Antonio Ocello[*], Claire Boyer[+,°], Sylvain Le Corff[†], and Vincent Lemaire[†]

[†]LPSM, Sorbonne Université, UMR CNRS 8001, Paris, France.
[*]CMAP, École Polytechnique, Institut Polytechnique de Paris, France.
[+]LMO, Université Paris-Saclay, UMR CNRS 8628, Orsay, France.
[°]IUF, Institut Universitaire de France.

**Reviewed on OpenReview:** https://openreview.net/forum?id=BlYIPaOFx1&noteId=BlYIPaOFx1

## Abstract

Score-based generative models (SGMs) aim at estimating a target data distribution by learning score functions using only noise-perturbed samples from the target. Recent literature has focused extensively on assessing the error between the target and estimated distributions, gauging the generative quality through the Kullback-Leibler (KL) divergence and Wasserstein distances. Under mild assumptions on the data distribution, we establish an upper bound for the KL divergence between the target and the estimated distributions, explicitly depending on any time-dependent noise schedule. Under additional regularity assumptions, taking advantage of favorable underlying contraction mechanisms, we provide a tighter error bound in Wasserstein distance compared to state-of-the-art results. In addition to being tractable, this upper bound jointly incorporates properties of the target distribution and SGM hyperparameters that need to be tuned during training. Finally, we illustrate these bounds through numerical experiments using simulated and CIFAR-10 datasets, identifying an optimal range of noise schedules within a parametric family.

## 1 Introduction

Recent years have seen impressive advances in machine learning and artificial intelligence, with one of the most notable breakthroughs being the success of diffusion models, introduced by Sohl-Dickstein et al. (2015). Diffusion models in generative modeling refer to a class of algorithms that generate new samples given training samples of an unknown distribution $\pi_{\text{data}}$. This method is now recognized for its ability to produce high-quality images that appear genuine to human observers (see *e.g.*, Ramesh et al., 2022, for text-to-image generation). Its range of applications is expanding rapidly, yielding impressive outcomes in areas such as computer vision (Li et al., 2022; Lugmayr et al., 2022) or natural language generation (Gong et al., 2023), among others, see Yang et al. (2023) for a comprehensive overview of the latest advances in this topic.

**Score-based generative models (SGMs).**    Generative diffusion models aim at creating synthetic instances of a target distribution when only a genuine sample (*e.g.*, a dataset of real-life images) is accessible. It is crucial to note that the complexity of real data prohibits a thorough depiction of the distribution $\pi_{\text{data}}$ through standard non-parametric density estimation strategies. Score-based Generative Models (SGMs) are probabilistic models designed to address this challenge using two main phases. The first phase, the noising phase (also referred to as the forward phase), involves progressively perturbing the empirical distribution by adding noise to the training data until its distribution approximately reaches an easy-to-sample distribution $\pi_\infty$. The second phase involves learning to reverse this noising dynamics by sequentially removing the noise, which is referred to as the sampling phase (or backward phase). Reversing the dynamics during the backward phase would require in principle knowledge of the score function, i.e., the gradient of the logarithm of the density at each time step of the diffusion. To circumvent this issue, the score function is learned based on the evolution of the noised data samples and using a deep neural network architecture. When applying these learned reverse dynamics to samples from $\pi_\infty$, we obtain a generative distribution that approximates $\pi_{\text{data}}$.

**Related works.**    Significant attention has been paid to understanding the sources of errors that affect the quality of data generation associated with SGMs (Block et al., 2020; De Bortoli, 2022; Lee et al., 2022; 2023; Chen et al., 2023a;b). In particular, a key area of interest has been the derivation of upper bounds for distances or pseudo-distances between the training and generated sample distributions. Note that all the mathematical theory for diffusion models developed so far covers general time discretizations of time-homogeneous SGMs (see Song and Ermon, 2019, in the variance-preserving case), which means that the strength of the noise is prescribed to be constant during the forward phase. De Bortoli et al. (2021); Chen (2023) provided upper bounds in terms of total variation, by assuming smoothness properties of the score and its derivatives. On the other hand, the upper bounds in total variation and Wasserstein distances provided by Lee et al. (2023); Gao et al. (2023) also require smoothness assumptions on the data distribution, either involving non-explicit constants, or focusing on iteration complexity sharpness. More recently, Conforti et al. (2023); Benton et al. (2024) established an upper bound in terms of Kullback–Leibler (KL) divergence avoiding strong assumptions about the score regularity, and relying on mild conditions about the data distribution (e.g., assumed to be of finite Fisher information w.r.t. the Gaussian distribution). Regarding time-inhomogeneous SGMs, the central role of the noise schedule has already been exhibited in numerical experiments, see for instance Chen (2023); Nichol and Dhariwal (2021); Guo et al. (2023). However, a rigorous theoretical analysis of it is still missing.

**Contributions.**    In this paper, we conduct a thorough mathematical analysis of the role of the noise schedule in score-based generative models. We propose a unified framework for time-inhomogeneous SGMs, to conduct joint theoretical analyses in KL and Wasserstein metrics, with state-of-the-art set of assumptions, using exponential integration of the backward process. In our opinion, these upper-bounds provide numerical insights into proper SGM training.

- We establish an upper bound on the Kullback-Leibler divergence between the data distribution and the law of the SGM. This bound holds under the mildest assumptions used in the SGM literature and explicitly depends on the noise schedule used to train the SGM. The proof follows the same steps as Conforti et al. (2023). However, it requires to establish a Kullback-Leibler upper bound for an inhomogeneous forward diffusion which involves

determining a non-asymptotic rate of convergence for the mixing time using Fokker-Planck equations and a log-Sobolev inequality that depends on the noise schedule, and not only on the diffusion time horizon, see Lemma B.1. In addition, taking into account the backward contraction for the diffusion process (Proposition C.1) provides state-of-the-art results on mixing time convergence for SGM under the Ornstein-Uhlenbeck forward process, whether inhomogeneous or not.

- By making additional assumptions on the Lipschitz and strong log-concavity properties of the score function, we establish a bound in terms of Wasserstein distance explicitly depending on the noise schedule. This extends the similar result for the KL in the Gaussian setting. These results are in the same line of work as Bruno et al. (2023); Gao et al. (2023), incorporating to the time inhomogeneous setting a refinement of the mixing time error based on an analysis of the modified score function.

- We illustrate, through numerical experiments, the upper bounds obtained in practice in regard of the effective empirical KL divergences and Wasserstein metrics. These simulations highlights the relevancy of the upper bound, reflecting in practice the effect of the noise schedule on the quality of the generative distribution. Additionally, the simulations conducted provide theoretically-inspired guidelines for improving SGM training. For reproducibility purposes, the code for the numerical experiments is available at `https://github.com/StanislasStrasman/Noise_Schedule_for_Score-based_Generative_Models`.

## 2 Mathematical framework for SGMs

**Forward process.** Denote as $\beta : [0, T] \mapsto \mathbb{R}_{>0}$ the noise schedule, assumed to be continuous and non decreasing. Although originally developed using a finite number of noising steps (Sohl-Dickstein et al., 2015; Song and Ermon, 2019; Ho et al., 2020; Song et al., 2021), most recent approaches consider time-continuous noise perturbations through the use of stochastic differential equations (SDEs) (Song et al., 2021). Consider, therefore, a forward process given by

$$\mathrm{d}\overrightarrow{X}_t = -\frac{\beta(t)}{2\sigma^2}\overrightarrow{X}_t\mathrm{d}t + \sqrt{\beta(t)}\mathrm{d}B_t, \quad \overrightarrow{X}_0 \sim \pi_{\mathrm{data}}. \tag{1}$$

We denote by $p_t$ the density of $\overrightarrow{X}_t$ at time $t \in (0, T]$. Note that, up to the time change $t \mapsto \int_0^t \beta(s)/2\mathrm{d}s$, this process corresponds to the standard Ornstein–Uhlenbeck (OU) process, solution to

$$\mathrm{d}\overrightarrow{X}_t = -\frac{1}{\sigma^2}\overrightarrow{X}_t\mathrm{d}t + \sqrt{2}\mathrm{d}B_t, \quad \overrightarrow{X}_0 \sim \pi_{\mathrm{data}},$$

see, e.g., Karatzas and Shreve (2012, Chapter 3). Due to the linear nature of the drift with respect to $(X_t)_t$, it is well-known that an exact simulation can be performed for this process (Section E.1.2). The stationary distribution $\pi_\infty$ of the forward process is the Gaussian distribution with mean 0 and variance $\sigma^2 \mathrm{I}_d$. In the literature, when $\beta(t)$ is constant equal to 2 (meaning that there is no time change), this diffusion process is referred to as the Variance-Preserving SDE (VPSDE, De Bortoli et al., 2021; Conforti et al., 2023; Chen et al., 2023b), leading to the so-called Denoising Diffusion Probabilistic Models (DDPM, Ho et al., 2020). Understanding the effects of the general diffusion model (1), in particular when reversing the dynamic, remains a challenging problem, to which we devote the rest of our analysis.

**Backward process.** The corresponding backward process is given by

$$
\begin{cases}
\mathrm{d}\overleftarrow{X}_t = \eta(t, \overleftarrow{X}_t)\mathrm{d}t + \sqrt{\bar{\beta}(t)}\mathrm{d}B_t, \\
\overleftarrow{X}_0 \sim \pi_\infty,
\end{cases}
\quad \text{with} \quad
\begin{cases}
\bar{\beta}(t) := \beta(T-t) \\
\eta(t, \overleftarrow{X}_t) := \frac{\bar{\beta}(t)}{2\sigma^2}\overleftarrow{X}_t + \bar{\beta}(t)\nabla \log p_{T-t}\left(\overleftarrow{X}_t\right).
\end{cases}
$$

We consider the marginal time distribution of the forward process divided by the density of its stationary distribution, introducing

$$
\forall x \in \mathbb{R}^d, \quad \tilde{p}_t(x) := p_t(x)/\varphi_{\sigma^2}(x), \tag{2}
$$

where $\varphi_{\sigma^2}$ denote the density function of $\pi_\infty$, a Gaussian distribution with mean 0 and variance $\sigma^2 \mathrm{I}_d$. Thus, the backward process can be rewritten as

$$
\mathrm{d}\overleftarrow{X}_t = \bar{\eta}\left(t, \overleftarrow{X}_t\right)\mathrm{d}t + \sqrt{\bar{\beta}(t)}\mathrm{d}B_t, \quad \overleftarrow{X}_0 \sim \pi_\infty, \tag{3}
$$

where $\bar{\eta}(t, \overleftarrow{X}_t) := -\frac{\bar{\beta}(t)}{2\sigma^2}\overleftarrow{X}_t + \bar{\beta}(t)\nabla \log \tilde{p}_{T-t}(\overleftarrow{X}_t)$. The benefit of using the renormalization $\tilde{p}_t$ in our analysis results in considering the backward equation as a perturbation of an OU process. This trick is crucial to highlight the central role of the relative Fisher information in the performance of the SGM. It has already been used by Conforti et al. (2023).

**Score estimation.** Simulating the backward process means knowing how to operate the score. However, the (modified) score function $\nabla \log \tilde{p}_t(x) = \nabla \log p_t(x) + x/\sigma^2$ cannot be evaluated directly, because it depends on the unknown data distribution. To work around this problem, the score function $\nabla \log p_t$ needs to be estimated. In Hyvärinen and Dayan (2005), the authors proposed to estimate the score function associated with a distribution by minimizing the expected $\mathrm{L}^2$-squared distance between the true score function and the proposed approximation. In the context of diffusion models, this is typically done with the use of a deep neural network architecture $s_\theta : [0,T] \times \mathbb{R}^d \mapsto \mathbb{R}^d$ parameterized by $\theta \in \Theta$, and trained to minimize:

$$
\mathcal{L}_{\text{explicit}}(\theta) = \mathbb{E}\left[\left\|s_\theta\left(\tau, \overrightarrow{X}_\tau\right) - \nabla \log p_\tau\left(\overrightarrow{X}_\tau\right)\right\|^2\right], \tag{4}
$$

with $\tau \sim \mathcal{U}(0,T)$ independent of the forward process $(\overrightarrow{X}_t)_{t\geq 0}$. However, this estimation problem still suffers from the fact that the regression target is not explicitly known. A tractable optimization problem sharing the same optima can be defined though, through the marginalization over $\pi_{\text{data}}$ of $p_\tau$ (see Vincent, 2011; Song et al., 2021):

$$
\mathcal{L}_{\text{score}}(\theta) = \mathbb{E}\left[\left\|s_\theta\left(\tau, \overrightarrow{X}_\tau\right) - \nabla \log p_\tau\left(\overrightarrow{X}_\tau|X_0\right)\right\|^2\right], \tag{5}
$$

where $\tau$ is uniformly distributed on $[0,T]$, and independent of $X_0 \sim \pi_{\text{data}}$ and $\overrightarrow{X}_\tau \sim p_\tau(\cdot|X_0)$. This loss function is appealing as it only requires to know the transition kernel of the forward process. In (1), this is a Gaussian kernel with explicit mean and variance.

**Discretization.** Once the score function is learned, it remains that, in most cases, the backward dynamics no longer enjoys a linear drift, which makes its exact simulation challenging. To address

this issue, one solution is to discretize the continuous dynamics of the backward process. In this way, Song et al. (2021) propose an Euler-Maruyama (EM) discretization scheme in which both the drift and the diffusion coefficients are discretized recursively (see (50)). The Euler Exponential Integrator (EI, see Durmus and Moulines, 2015), as already used in Conforti et al. (2023), only requires to discretize the part associated with the modified score function. Introduce $\tilde{s}_\theta(t, x) := s_\theta(t, x) + x/\sigma^2$ and consider the regular time discretization $0 = t_0 \leq t_1 \leq \cdots \leq t_N = T$. Then, $(\overleftarrow{X}_t^\theta)_{t \in [0,T]}$ is such that, for $t \in [t_k, t_{k+1}]$,

$$\mathrm{d}\overleftarrow{X}_t^\theta = \bar{\beta}(t) \left( -\frac{1}{2\sigma^2} \overleftarrow{X}_t^\theta + \tilde{s}_\theta \left( T - t_k, \overleftarrow{X}_{t_k}^\theta \right) \right) \mathrm{d}t + \sqrt{\bar{\beta}(t)} \mathrm{d}B_t, \quad \overleftarrow{X}_0^\theta \sim \pi_\infty. \tag{6}$$

This scheme can be seen as a refinement of the classical EM one as it handles the linear drift term by integrating it explicitly. In addition, $(\overleftarrow{X}_t^\theta)_{t \in \{t_0, \ldots, t_N\}}$ can be sampled exactly, see Appendix A. We consider therefore such a scheme in our further theoretical developments.

## 3 Non-asymptotic Kullback-Leibler bound

In this section, we provide a theoretical analysis of the effect of the noise schedule used when training an SGM. Its impact is scrutinized through a bound on the KL divergence between the data distribution and the generative one.

**Statement.** The data distribution $\pi_{\mathrm{data}}$ is assumed to be absolutely continuous with respect to the Gaussian measure $\pi_\infty$. Define the relative Fisher information $\mathcal{I}(\pi_{\mathrm{data}}|\pi_\infty)$ by

$$\mathcal{I}(\pi_{\mathrm{data}}|\pi_\infty) := \int \left\| \nabla \log \left( \frac{\mathrm{d}\pi_{\mathrm{data}}}{\mathrm{d}\pi_\infty} \right) \right\|^2 \mathrm{d}\pi_{\mathrm{data}},$$

and consider the following assumptions.

**H1** The noise schedule $\beta$ is continuous, positive, non decreasing and such that $\int_0^\infty \beta(t)\mathrm{d}t = \infty$.

**H2** The data distribution is such that $\mathcal{I}(\pi_{\mathrm{data}}|\pi_\infty) < \infty$.

**H3** The NN parameter $\theta \in \Theta$ and the schedule $\beta$ satisfy

$$\mathbb{E} \left[ \exp \left\{ \frac{1}{2} \int_0^T \bar{\beta}(t) \left\| \left( \tilde{s} \left( T - t, \overleftarrow{X}_t \right) - \tilde{s}_\theta \left( T - t, \overleftarrow{X}_t \right) \right) \right\|^2 \mathrm{d}t \right\} \right] < \infty,$$

with $\tilde{s}(t, x) := \nabla \log \tilde{p}_t(x)$ and $\tilde{p}_t$ defined in (2).

Assumption H1 is necessary to ensure that the forward process converges to the stationary distribution when the diffusion time tends to infinity. Assumption H2 is inherent to the data distribution, as it involves only the $L^2$-integrability of the score function. Such a kind of hypothesis has already been considered in the literature, see Conforti et al. (2023). We stress that, in this section, we do not require extra assumptions about the smoothness of the score function. Lastly, Assumption H3 is a condition on the approximation of the score by the neural network $\tilde{s}_\theta$, weighted by the level of noise in play. We are now in position to provide an upper bound for the relative entropy between the distribution $\widehat{\pi}_N^{(\beta,\theta)}$ of samples obtained from (6), and the target data distribution $\pi_{\mathrm{data}}$.

**Theorem 3.1.** *Assume that H1, H2 and H3 hold. Then,*

$$\mathrm{KL}\left(\pi_{\mathrm{data}}\middle\|\widehat{\pi}_N^{(\beta,\theta)}\right) \leq \mathcal{E}_1^{\mathrm{KL}}(\beta) + \mathcal{E}_2^{\mathrm{KL}}(\theta,\beta) + \mathcal{E}_3^{\mathrm{KL}}(\beta),$$

*where*

$$\mathcal{E}_1^{\mathrm{KL}}(\beta) = \mathrm{KL}\left(\pi_{\mathrm{data}}\middle\|\pi_\infty\right)\exp\left\{-\frac{1}{\sigma^2}\int_0^T \beta(s)\mathrm{d}s\right\},$$

$$\mathcal{E}_2^{\mathrm{KL}}(\theta,\beta) = \sum_{k=0}^{N-1}\mathbb{E}\left[\left\|\nabla\log\tilde{p}_{T-t_k}\left(\overrightarrow{X}_{T-t_k}\right) - \tilde{s}_\theta\left(T-t_k,\overrightarrow{X}_{T-t_k}\right)\right\|^2\right]\int_{T-t_{k+1}}^{T-t_k}\beta(t)\mathrm{d}t,$$

$$\mathcal{E}_3^{\mathrm{KL}}(\beta) = 2h\beta(T)\mathcal{I}(\pi_{\mathrm{data}}|\pi_\infty),$$

*with $h := \sup_{k\in\{1,\ldots,N\}}(t_k - t_{k-1})$ small enough and $t_0 := 0$.*

The obtained bound is composed of three terms, all depending on the noise schedule, through either its integrated version over the diffusion time, or its final value at time $T$. The result was derived for the EI discretization scheme, but it could be adapted to the Euler scheme up to minor technicalities. Remark also that using Pinsker's inequality, the obtained bound could be transferred in terms of total variation.

**Dissecting the upper bound.** The upper bound of Theorem 3.1 involves three different types of error that affect the training of an SGM. The term $\mathcal{E}_1^{\mathrm{KL}}$ represents the *mixing time* of the OU forward process, arising from the practical limitation of considering the forward process up to a finite time $T$. Indeed, $\mathcal{E}_1^{\mathrm{KL}}$ is shrinked to 0 when $T$ grows to infinity. Note that the multiplicative term in $\mathcal{E}_1^{\mathrm{KL}}$ corresponds to the KL divergence between $\pi_{\mathrm{data}}$ and $\pi_\infty$ which is ensured to be finite by Assumption H2. The second term $\mathcal{E}_2^{\mathrm{KL}}$ corresponds to the *approximation error*, which stems from the use of a deep neural network to estimate the score function. Note that if we assume that the error of the score approximation is uniformly (in time) bounded by $M_\theta$ (see De Bortoli et al., 2021, Equation (8)), the term $\mathcal{E}_2^{\mathrm{KL}}$ admits as a crude bound $M_\theta\int_0^T\beta(t)\mathrm{d}t$, with the disadvantage of exploding when $T \to +\infty$. Otherwise, by considering Conforti et al. (2023, Assumption H3), one can make this bound finer and finite, by balancing the quality of the score approximation, the discretization grid and the final time $T$. Finally, $\mathcal{E}_3^{\mathrm{KL}}$ is the *discretization error* of the EI discretization scheme. This last term vanishes as the discretization grid is refined (i.e., $h \to 0$).

**Comparison with existing bounds.** Under perfect score approximation,

and infinitely precise discretization (i.e., when $\mathcal{E}_2^{\mathrm{KL}}(\theta,\beta) = \mathcal{E}_3^{\mathrm{KL}}(\beta) = 0$), we recover that the Variance Preserving SDE (VPSDE, De Bortoli et al., 2021; Conforti et al., 2023; Chen et al., 2023b) converges exponentially fast to the target distribution. Beyond this idealized setting, the bound established in Theorem 3.1 recovers that of Conforti et al. (2023, Theorem 1) when choosing $\beta(t) = 2$, $\sigma^2 = 1$, $T = 1$, and using a discretization step size $h \leq 1$.

**Refined analysis of the mixing time error** Still assuming "perfect score approximation" and infinitely precise discretization (i.e., $\mathcal{E}_2^{\mathrm{KL}}(\theta,\beta) = \mathcal{E}_3^{\mathrm{KL}}(\beta) = 0$), one can assess the sharpness of the term $\mathcal{E}_1^{\mathrm{KL}}(\beta)$ in the upper bound of Theorem 3.1. In particular, when restricting the data distribution to be Gaussian $\mathcal{N}(\mu_0, \Sigma_0)$, one can exploit the backward contraction assuming that

$\lambda_{\max}(\Sigma_0) \leq \sigma^2$, where $\lambda_{\max}(\Sigma_0)$ denotes the largest eigenvalue of $\Sigma_0$. In this specific case, we can obtain a refined version for $\mathcal{E}_1^{\mathrm{KL}}$ (see Proposition C.1), given by

$$\mathrm{KL}\left(\pi_{\mathrm{data}}\|\pi_\infty Q_T\right) \leq \mathrm{KL}\left(\pi_{\mathrm{data}}\|\pi_\infty\right) \exp\left(-\frac{2}{\sigma^2}\int_0^T \beta(s)\mathrm{d}s\right), \tag{7}$$

where $(Q_t)_{0 \leq t \leq T}$ is the Markov semi-group associated with the backward SDE. This idea is exploited in Section 4 to establish Wasserstein bounds for more general data distributions than Gaussian, but requiring extra regularity of the score.

## 4 Non-asymptotic Wasserstein bound

In the literature, much attention is paid to derive upper bounds with other metrics such as the $\mathcal{W}_2$ distance, which has the advantage to be a distance and to have easier-to-handle and implementable estimators. In Lee et al. (2023), the authors obtain a control for the 2-Wasserstein and total variation distances. However, those results rely on additional assumptions on $\pi_{\mathrm{data}}$ (which is assumed to have bounded support for instance in De Bortoli (2022)).

**Regularity assumptions.** We consider extra regularity assumptions about the modified marginal density $\tilde{p}_t$ at any time of the diffusion.

**H4** (i) For all $t \geq 0$, there exists $C_t \geq 0$ such that for all $x, y \in \mathbb{R}^d$,

$$(\nabla \log \tilde{p}_t(y) - \nabla \log \tilde{p}_t(x))^\top (x - y) \geq C_t \|x - y\|^2 .$$

(ii) For all $t \geq 0$, there exists $L_t \geq 0$ such that $\nabla \log \tilde{p}_t$ is $L_t$-Lipschitz continuous.

The strong log-concavity (i) (see, e.g., Saumard and Wellner, 2014) plays a crucial role in terms of contraction of the backward SDE. Classical distributions satisfying H4(i) include logistic densities restricted to a compact set, or Gaussian laws with a positive definite covariance matrix, see Saumard and Wellner (2014) for other examples. We observe, notably, that when the density of the data distribution is log-concave, this property propagates within the probability flow $(\tilde{p}_t)_{0 \leq t \leq T}$ (see Proposition D.1). Similar conclusions can be drawn regarding the Lipschitz continuity of the score (Proposition D.2). This property is formalized in the Lemma 4.1 for Gaussian distributions.

**Lemma 4.1.** *Assume that $\pi_{\mathrm{data}}$ is a Gaussian distribution $\mathcal{N}(\mu_0, \Sigma_0)$, such that $\Sigma_0$ is invertible and $\lambda_{\max}(\Sigma_0) < \sigma^2$. Let $m_t := \exp(-\int_0^t \beta(s)/2\sigma^2 \mathrm{d}s)$. Then, the probability flow $\tilde{p}_t$ given by (1) initialized at $\pi_{\mathrm{data}}$ is $C_t$-strongly log concave, with*

$$C_t := \frac{m_t^2\left(\sigma^2 - \lambda_{\max}(\Sigma_0)\right)}{m_t^2 \lambda_{\max}(\Sigma_0) + \sigma^2\left(1 - m_t^2\right)} .$$

*In addition, the associated score $\nabla \log \tilde{p}_t$ is $L_t$-Lipschitz continuous with*

$$L_t := \min\left\{\frac{1}{\sigma^2\left(1 - m_t^2\right)}; \frac{1}{\lambda_{\min}(\Sigma_0)m_t^2}\right\} + \frac{1}{\sigma^2} .$$

This result, restricted to the Gaussian case, sets the focus on the importance of calibrating the parameter $\sigma^2$ depending on the covariance structure of the data distribution, in order to enhance strong log concavity of the probability flow through the diffusion.

**Error bound.** To establish a 2-Wasserstein bound explicitly depending on the noise schedule, we consider the following additional assumptions, respectively about uniform approximation of the score, and Lipschitz continuity in time of the renormalized score.

**H5** There exists $\varepsilon \geq 0$ such that $\displaystyle\sup_{k \in \{0,..,N-1\}} \left\| \tilde{s}\left(T - t_k, \bar{X}_{t_k}^\theta\right) - \tilde{s}_\theta\left(T - t_k, \bar{X}_{t_k}^\theta\right) \right\|_{L_2} \leq \varepsilon$.

**H6** For a regular discretization $\{t_k, 0 \leq k \leq N\}$ of $[0, T]$ of constant step size $h$, there exists $M \geq 0$ such that

$$\sup_{k \in \{0,..,N-1\}} \sup_{t_k \leq t \leq t_{k+1}} \left\| \tilde{s}\left(T - t, x\right) - \tilde{s}\left(T - t_k, x\right) \right\|_{L_2} \leq Mh(1 + \|x\|).$$

We now have all the ingredients to present our theoretical guarantee in terms of Wasserstein distance.

**Theorem 4.2.** *Assuming H4, H5 and H6 and that the time step $h$ is small enough, it holds that*

$$\mathcal{W}_2\left(\pi_{\text{data}}, \widehat{\pi}_N^{(\beta, \theta)}\right) \leq \mathcal{E}_1^{\mathcal{W}_2}(\beta) + \mathcal{E}_2^{\mathcal{W}_2}(\theta, \beta) \tag{8}$$

*with*
$$\mathcal{E}_1^{\mathcal{W}_2}(\beta) = \mathcal{W}_2\left(\pi_{\text{data}}, \pi_\infty\right) \exp\left(-\int_0^T \frac{\beta(t)}{\sigma^2}\left(1 + C_t \sigma^2\right) dt\right),$$

$$\mathcal{E}_2^{\mathcal{W}_2}(\theta, \beta) = \sum_{k=0}^{N-1} \left(\int_{t_k}^{t_{k+1}} \bar{L}_t \bar{\beta}(t) dt\right) \left(\frac{\sqrt{2h\beta(T)}}{\sigma} + \frac{h\beta(T)}{2\sigma^2} + \int_{t_k}^{t_{k+1}} 2\bar{L}_t \bar{\beta}(t) dt\right) B$$
$$+ \varepsilon T \beta(T) + MhT\beta(T)\left(1 + 2B\right),$$

$B = \left(\mathbb{E}[\|X_0\|^2] + \sigma^2 d\right)^{1/2}$, *and for all* $t \in [0, T], \bar{L}_t = L_{T-t}$.

In Theorem 4.2, we exploit the contraction entailed by Assumption H4(i) of the backward diffusion processes on top of that of the forward phase. This idea leads to an improvement of all the existing bounds in Wasserstein metrics, by refining their mixing time term. The previous result can be established when the target distribution has a Lipschitz continuous score and is strongly log-concave: by propagating these properties, the constants $L_t$ and $C_t$ can be characterized as a function of $L_0$ and $C_0$ (see Propositions D.1 and D.2). The propagation of the log-concave property was also established in Saremi et al. (2023).

**Corollary 4.3.** *Assume that $\nabla \log \tilde{p}_{\text{data}}$ is $L_0$-Lipschitz, that $\log \tilde{p}_{\text{data}}$ is $C_0$-strongly concave such that $C_0 > 1/\sigma^2$. Under Assumption H5 and H6, with a time step $h$ small enough,*

$$\mathcal{W}_2\left(\pi_{\text{data}}, \widehat{\pi}_N^{(\beta, \theta)}\right) \leq \mathcal{W}_2\left(\pi_{\text{data}}, \pi_\infty\right) \exp\left(-\int_0^T \frac{\beta(t)}{\sigma^2}\left(1 + C_t \sigma^2\right) dt\right) + c_1 \sqrt{h} + c_2 h + \varepsilon T \beta(T),$$

*with $c_1 = L_0 \beta(T) T \sqrt{2\beta(T)}/\sigma$ and $c_2 = \beta(T) T \left(L_0\left(1/(2\sigma^2) + 2L_0\right) \beta(T) B + M(1 + 2B)\right)$.*

This provides an easy-to-handle upper bound in Wasserstein distance, encompassing the three types of error (e.g., mixing time, score approximation and discretization error), for Lipschitz scores and strongly-log concave distributions. We remark that it also exhibits an extra term in $\sqrt{h}$ compared to the more general KL bound obtained under milder assumptions. Note however that this term is in line with what can be found in the literature for Wasserstein bounds for SDE approximation (see Alfonsi et al., 2015).

**Discussion and comparison with other works.** Theorem 4.2 requires more stringent assumptions on the regularity of the score function than Theorem 3.1. However, these assumptions are not specific to our setting. In particular, the strong log-concavity assumption has proven to be a key property for the fast convergence of sampling algorithms (see Dalalyan, 2017; Durmus and Moulines, 2017; Dwivedi et al., 2019). While this is a strong assumption to require on the data density, this can be mitigated.

In Benton et al. (2024), the authors propose quantitative bounds for the Kullback-Leibler divergence only assuming a finite second moments of the data distribution and do not use any smoothness assumption. The authors use early stopping and stop the backward sampling at small time $\delta > 0$ to avoid the score explosion in the neighborhood of 0. In another line of work, Chen et al. (2023a) used a high-probability bound on the Hessian matrix of $p_t$ to avoid additional assumptions (such as a bounded support) on the data distribution to obtain Kullback-Leibler upper bounds. These works offer promising perspective to obtain Wasserstein bounds under weaker assumptions.

The analysis of the modified score functions in the Gaussian case reveals that by properly adjusting the variance of the stationary distribution of the forward process and rescaling the target distribution, we can attain the desired properties for the score function. This observation is specific to the Gaussian case as one can easily derive values for $C_0$ and $L_0$ with the eigenvalues of covariance matrix of the target distribution. However, we would like to strengthen the fact that a similar preprocessing has been applied to more complex distributions in Section 5.

## 5  Evaluation of the theoretical upper bounds

The goal of this section is to numerically illustrate the validity of the theoretical bounds obtained in Theorem 3.1 and Theorem 4.2. More precisely, we aim at unraveling the contributions of each error term of the upper bounds. We consider a simulation design where the target distribution is known, and the associated constants of interest (i.e., the strong log concavity parameter, the Lipschitz constant, $\mathcal{W}_2\left(\pi_{\text{data}}, \pi_\infty\right)$, $\mathcal{I}\left(\pi_{\text{data}}|\pi_\infty\right)$ or $\mathrm{KL}\left(\pi_{\text{data}}||\pi_\infty\right)$) can be evaluated. The error bounds are assessed for different choices of noise schedules of the form

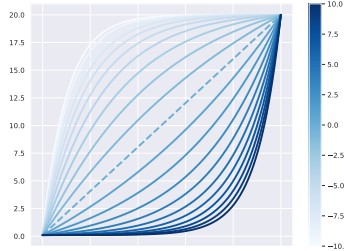

$$\beta_a(t) \propto (\mathrm{e}^{at} - 1)/(\mathrm{e}^{aT} - 1), \qquad (9)$$

with $a \in \mathbb{R}$ ranging from $-10$ to $10$ with a unit step size. We set $T = 1$ and adjust schedules so that they all start at $\beta(0) = 0.1$ and end at $\beta(1) = 20$ (see Figure 1). This choice has been made so that when $a = 0$ the schedule is linear and matches exactly the classical VPSDE implementation (Song and Ermon, 2019; Song et al., 2021).

Figure 1: Noise schedule $\beta_a$ over time for $a \in \{-10, -9, .., 10\}$ with the linear schedule $a = 0$ shown as a dashed line.

### 5.1  Gaussian setting

**Target distributions.** We consider the setting where the true distribution $\pi_{\text{data}}$ is Gaussian in dimension $d = 50$ with mean $\mathbf{1}_d$ and different choices of covariance structure:

1. (Isotropic, denoted by $\pi_{\text{data}}^{(\text{iso})}$) $\Sigma^{(\text{iso})} = 0.5\mathrm{I}_d$.

2. (Heteroscedastic, denoted by $\pi_{\text{data}}^{(\text{heterosc})}$) $\Sigma^{(\text{heterosc})} \in \mathbb{R}^{d \times d}$ is a diagonal matrix such that $\Sigma_{jj}^{(\text{heterosc})} = 1$ for $1 \leq j \leq 5$, and $\Sigma_{jj}^{(\text{heterosc})} = 0.01$ otherwise.

3. (Correlated, denoted by $\pi_{\text{data}}^{(\text{corr})}$) $\Sigma^{(\text{corr})} \in \mathbb{R}^{d \times d}$ is a full matrix whose diagonal entries are equal to one and the off-diagonal terms are $\Sigma_{jj'}^{(\text{corr})} = 1/\sqrt{|j - j'|}$ for $1 \leq j \neq j' \leq d$.

**SGM simulations.** We simulate $\widehat{\pi}_N^{(\beta_a, \theta)}$ from SGM using the forward process defined in (1) with $t \mapsto \beta_a(t)$ for the noise schedule. The score is learned via a dense neural network with 3 hidden layers of width 256 over 150 epochs (see Figure 11) trained to optimize $\mathcal{L}_{\text{explicit}}$ (4). This is feasible because the score is analytically derived when $\pi_{\text{data}}$ is Gaussian (Lemma E.1). Numerical experiments have also been run with the commonly used conditional loss $\mathcal{L}_{\text{score}}$, without changing the nature of the conclusions, see Appendix F. For backward process simulation, we use an Euler-Maruyama scheme with 500 steps, as being the most encountered discretization in practice (with these discretization steps the difference with Exponential Integrator scheme is minimal as highlighted in the appendix). For each value of $a$, and each data distribution, we train the SGM using $n = 10000$ training samples.

**KL bound.** In Figure 2 (top), we compare the empirical KL divergence between $\pi_{\text{data}}$ and samples from $\widehat{\pi}_N^{(\beta_a, \theta)}$ to the upper bound from Theorem 3.1. We refer the reader to Appendix E.2.1 for implementation details. For Gaussian distributions, both the bound and KL divergence can be computed using closed-form expressions (see Lemma E.2 and E.4). In all scenarios the noise schedule significantly impacts the value of $\text{KL}(\pi_{\text{data}} \| \widehat{\pi}_N^{(\beta_a, \theta)})$, and thereby the quality of the learned distribution. Moreover, in all three cases taking into acccount the contraction argument (7) is key to properly align the upper bound trend with the generation results. In all these experiments, the KL upper bound indicates possible values for $a$ improving over the classical linear noise schedule.

**2-Wasserstein bound.** In Figure 2 (bottom), we compare the empirical $\mathcal{W}_2$ distance between $\pi_{\text{data}}$ and samples from $\widehat{\pi}_N^{(\beta_a, \theta)}$ to the upper bound from Theorem 4.2. For Gaussian distributions, both the bound and the $\mathcal{W}_2$ distance can be computed using closed-form expressions (see Lemma 4.1, E.3, and E.5). For the isotropic case, the proposed $\mathcal{W}_2$ upper bound reflects the SGM performances, as already highlighted by the KL bound. However, in non-isotropic cases, the raw distributions $\pi_{\text{data}}^{(\text{heterosc})}$ and $\pi_{\text{data}}^{(\text{corr})}$ do not directly satisfy Assumption 4 (i) when the variance of the stationary distribution is set to 1. Therefore, scaling the distributions in play becomes crucial for the theoretical $\mathcal{W}_2$ upper bound to hold. That is why we propose the following preprocessing: train an SGM with centered and standardized samples of covariance $\Sigma^{(\text{stand})}$ rescaled in turn by a factor $1/(2\lambda_{\max}(\Sigma^{(\text{stand})}))^{1/2}$. This choice ensures that $\lambda_{\max}\left(\Sigma^{(\text{scaled})}\right) < \sigma^2 = 1$, for $\Sigma^{(\text{scaled})}$ the resulting covariance matrix, and thus the strong log-concavity of $\tilde{p}_0 = p_{\text{data}}/\varphi_{\sigma^2}$. We call $\widehat{\pi}_{N,\text{scaled}}^{(\beta_a, \theta)}$ the resulting generative distribution, and the evaluated metrics is adjusted (see (51)) to ensure a fair numerical comparison. After this preprocessing, not only the $\mathcal{W}_2$ upper bound of Theorem 4.2 aligns with the empirical performances but the SGM performances can be also boosted (see degraded empirical performances on raw distributions in Appendix E.2.2). This highlights the importance of properly calibrating the training sample to the stationary distribution of the SGM. Note that data normalization does not only enforce the strong log-concavity of the modified score at time 0, but can lower the ratio $L_0/C_0$. To see this, consider the heteroscedastic case, for which $\lambda_{\min}(\Sigma^{(\text{heterosc})})/\lambda_{\max}(\Sigma^{(\text{heterosc})}) = 100$, whereas $\lambda_{\min}(\Sigma^{(\text{scaled})})/\lambda_{\max}(\Sigma^{(\text{scaled})}) = 1$ after scaling. This Gaussian set-up reveals that data

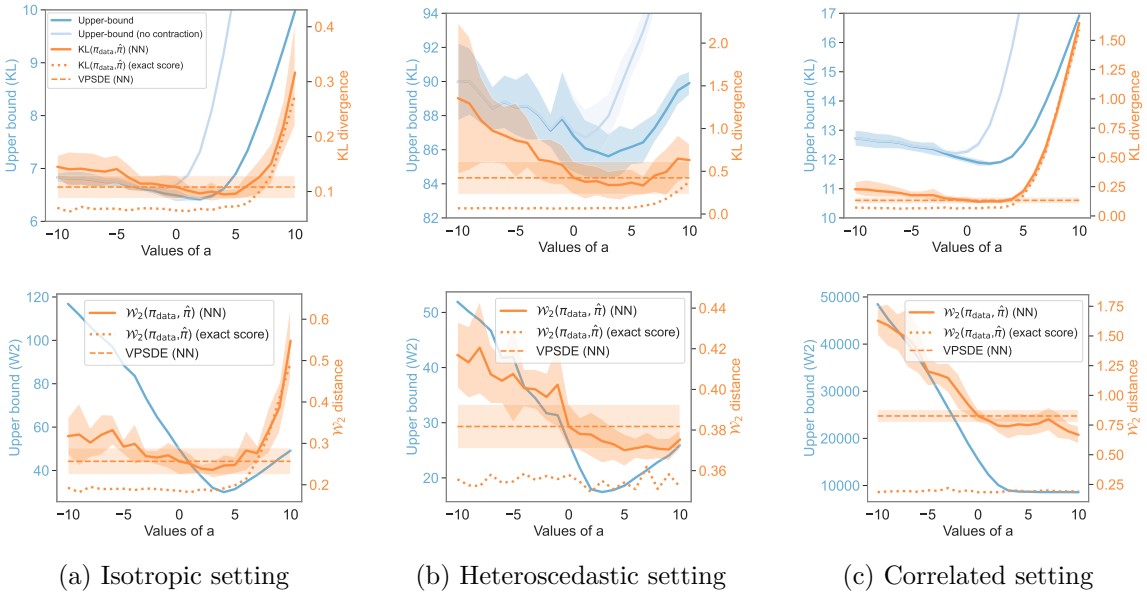

(a) Isotropic setting      (b) Heteroscedastic setting      (c) Correlated setting

Figure 2: Comparison of the empirical KL divergence (mean $\pm$ std over 30 runs) (top) and $\mathcal{W}_2$ distance (mean $\pm$ std over 10 runs)(bottom) between $\pi_{\mathrm{data}}$ and $\widehat{\pi}_N^{(\beta,\theta)}$ (orange) and the related upper bounds (blue) from Theorem 3.1 and Theorem 4.2 across parameter $a$ for noise schedule $\beta_a$, $d = 50$. In the KL case, the upper bounds in lighter blue are the theoretical upper bounds without taking into the contraction argument (7). We also show the metrics for the linear VPSDE model (dashed line) and our model (dotted line) with exact score evaluation.

renormalization improves the conditioning of the covariance matrix, and thereby the conditioning of SGM training. In particular, this is captured in the upper bound of Theorem 4.2 by limiting the growth of $L_t$ and inducing a more balanced second term.

We now consider a varying dimension in $\{5, 10, 25, 50\}$, and we compare the empirical $\mathcal{W}_2$ distance obtained by (i) $\beta_0$ the classical VPSDE (Song et al., 2021), with a linear noise schedule (i.e., $a = 0$), (ii) $\beta_{\cos}$ the SGM with cosine schedule (Nichol and Dhariwal, 2021), and (iii) $\beta_{a^\star}$ the SGM with parametric schedule with $a = a^\star$ approximately minimizing the upper bound from Theorem 3.1. In Figure 3, we observe that the SGMs run with $\beta_{a^\star}$ consistently outperforms those run with linear schedule $\beta_0$ slightly improving the data generation quality. It displays lower average $\mathcal{W}_2$ distances between $\pi_{\mathrm{data}}$ and the generated sample distribution, but also reduces the standard deviation of the resulting $\mathcal{W}_2$ distances yielding more stable generation (see Table 2). These performances are comparable to, and often surpass, those achieved with state-of-the-art schedules like the cosine schedule, particularly in higher dimensions.

## 5.2 More general target distributions

Beyond Gaussian distributions, numerical analysis in terms of KL divergence is not tractable as standard estimators of the KL terms do not scale well with dimen-

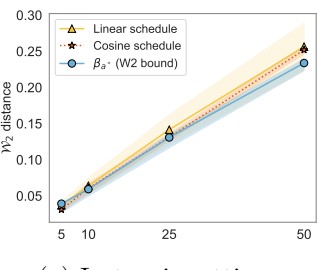
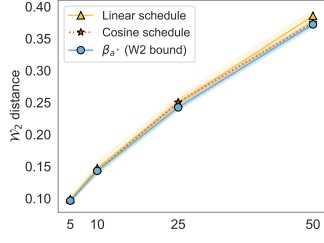
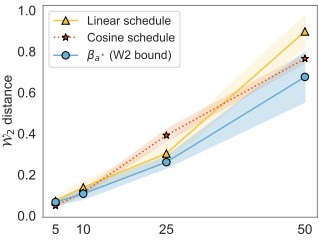

(a) Isotropic setting      (b) Rescaled heterosc. setting      (c) Rescaled correlated setting

Figure 3: Comparison of the empirical $\mathcal{W}_2$ distance (mean value $\pm$ std over 10 runs) between $\pi_{\text{data}}$ and the generative distribution $\widehat{\pi}_N^{(\beta,\theta)}$ across various dimensions. The distributions compared include SGMs with different noise schedules: $\beta_{a^\star}$ (blue solid), $\beta_0$ (yellow dashed), and $\beta_{\cos}$ (orange dotted).

sion. On the contrary, there exist computationally-efficient estimators of Wasserstein distances, as for instance the sliced $\mathcal{W}_2$ estimate (Flamary et al., 2021).

We use the latter to assess the relevancy of Theorem 4.2 when the target distribution corresponds to a 50-dimensional Funnel distribution defined as: $\pi_{\text{data}}(x) = \varphi_{a^2}(x_1)\prod_{j=2}^{d}\varphi_{\exp(2bx_1)}(x_j)$, with $a = 1$ and $b = 0.5$ (see Section E.2.3 for more details and additional experiments on a Gaussian mixture model). As previously, the samples are standardized and rescaled. In Figure 4, empirical results demonstrate that the minimum of the upper bound closely aligns with that of the empirical sliced 2-Wasserstein distance between the simulated and training data. Moreover, implementing SGM with the optimal parameter $a$ yields consistent improvements of the data generation quality across different metrics w.r.t. to classical noise schedule competitors (linear or cosine). These experiments not only

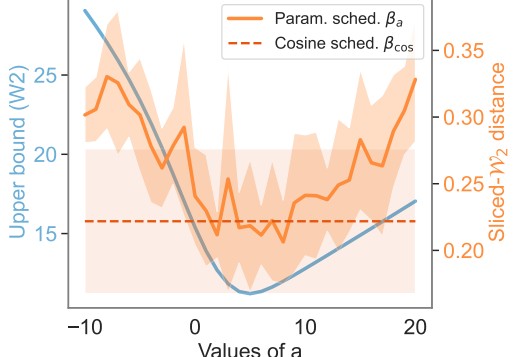

Figure 4: Upper bound and sliced 2-Wasserstein distance on a Funnel dataset in dimension 50.

support the relevance of the theoretical upper bound beyond the assumptions required in Section 4, but also the validity of theoretically-inspired data preprocessing for improving SGM training with arbitrary target distributions.

When dealing with high-dimensional real-world datasets, directly evaluating our theoretical upper bounds (Theorems 3.1 and 4.2) becomes more challenging because relevant quantities (distances and constants) are either poorly estimated or unavailable. As a first step toward real data, we evaluate the impact of the noise schedule on the sampling quality of models pre-trained using CIFAR-10 dataset. In Figure 5, we display the FID score with 50,000 generated samples using Euler-Maruyama discretization scheme for various noise schedules drawn from the parametric family in Equation (9). Additional implementation details are available in Appendix E.3. Although the assumptions underpinning our results cannot be verified in this setting, the empirical performance trends mirror closely those observed in the simulated settings.

This consistency highlights that analyzing and optimizing noise schedules could be a promising direction for improving SGM-based generation in more complex scenarios.

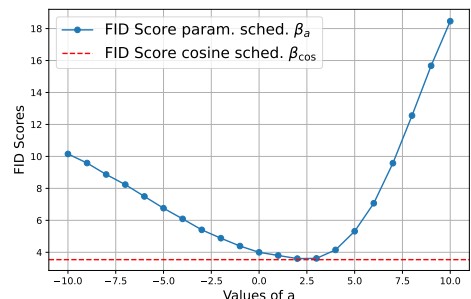

Figure 5: FID Scores using 50,000 generated samples for the parametric and cosine schedules (CIFAR-10 dataset).

## 6 Discussion

In this paper, we propose a unified framework to analyze the impact of the noise schedule for time-inhomogeneous SGMs, providing upper bounds in KL and Wasserstein metrics. The KL upper bound follows the steps of recent works using the mildest assumptions used in the SGM literature. We also provide an improved upper bound in the Gaussian setting with numerical experiments highlighting the impact of the backward contraction of the forward noise process. Following Bruno et al. (2023); Gao et al. (2023), under additional assumptions on the Lipschitz and strong log-concavity properties of the score function, we establish upper bounds for the Wasserstein distance. This bound highlights the role of the noise schedule and provides a detailed analysis based on the modified score function. Our results are supported by numerical experiments in simple settings to highlight the several terms of the upper bounds and the role of the noise schedule. There are many perspectives to this work. Studying multi-dimensional noise schedules is of particular interest. Indeed, they could be useful to understand how to deal with target distributions with complex covariance structures, and thereby an alternative solution to data normalization issues. Establishing upper bounds for Wasserstein distances under milder assumptions remains an exciting open problem, which would shed light on the performances and limitations of score-based generative models. A specific perspective would be to adapt our result using early-stopping to avoid the explosion of error terms in the neighborhood of 0 and to provide other assumptions to control the corresponding error close to 0.

**Acknowledgements**

We would like to thank Gabriel Victorino Cardoso for his valuable insights and thoughtful help on the numerical experiments involving real-world datasets.

Antonio Ocello was funded by the European Union (ERC-2022-SYG-OCEAN-101071601). Views and opinions expressed are however those of the author only and do not necessarily reflect those of the European Union or the European Research Council Executive Agency. Neither the European Union nor the granting authority can be held responsible for them.

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

# A    Notations and assumptions.

Consider the following notations, used throughout the appendices. For all $d \geq 1$, $\mu \in \mathbb{R}^d$ and definite positive matrices $\Sigma \in \mathbb{R}^{d \times d}$, let $\varphi_{\mu, \Sigma}$ be the probability density function of a Gaussian random variable with mean $\mu$ and variance $\Sigma$. We also use the notation $\varphi_{\sigma^2} = \varphi_{0, \sigma^2 I_d}$. When the context is clear, we may indifferently use the measure and the associated density w.r.t. the reference measure. For all twice-differentiable real-valued function $f$, let $\Delta f$ be the Laplacian of $f$. For all matrix $A \in \mathbb{R}^{m \times n}$, $\|A\|_{\mathrm{Fr}}$ is the Frobenius norm of $A$, i.e., $\|A\|_{\mathrm{Fr}} = (\sum_{i=1}^m \sum_{j=1}^n |A_{i,j}|^2)^{1/2}$. For all time-dependent real-valued functions $h : t \mapsto h_t$ or $f : t \mapsto f(t)$, we write $\bar{h}_t = h_{T-t}$ and $\bar{f}(t) = f(T-t)$ for all $t \in [0, T]$.

Let $\pi_0$ be a probability density function with respect to the Lebesgue measure on $\mathbb{R}^d$ and $\alpha : \mathbb{R} \to \mathbb{R}$ and $g : \mathbb{R} \to \mathbb{R}$ be two continuous and increasing functions. Consider the general forward process

$$\mathrm{d}\overrightarrow{X}_t = -\alpha(t)\overrightarrow{X}_t \mathrm{d}t + g(t)\mathrm{d}B_t, \quad \overrightarrow{X}_0 \sim \pi_0, \tag{10}$$

and introduce $\tilde{p}_t : x \mapsto p_t(x)/\varphi_{\sigma^2}(x)$, where $p_t$ is the probability density function of $\overrightarrow{X}_t$. The backward process associated with (10) is referred to as $(\overleftarrow{X}_t)_{t \in [0,T]}$ and given by

$$\mathrm{d}\overleftarrow{X}_t = \left\{ \left( \bar{\alpha}(t) - \frac{\bar{g}^2(t)}{\sigma^2} \right) \overleftarrow{X}_t + \bar{g}^2(t)\nabla \log \tilde{p}_{T-t}\left(\overleftarrow{X}_t\right) \right\} \mathrm{d}t + \bar{g}(t)\mathrm{d}\bar{B}_t \quad \overleftarrow{X}_0 \sim p_T, \tag{11}$$

with $\bar{B}$ a standard Brownian motion in $\mathbb{R}^d$. Moreover, consider

$$\sigma_t^2 := \exp\left( -2\int_0^t \alpha(s)\mathrm{d}s \right) \int_0^t g^2(s)\exp\left( 2\int_0^s \alpha(u)\mathrm{d}u \right) \mathrm{d}s. \tag{12}$$

The approximate EI discretization of (11) considered in this paper is, for $t_k \leq t \leq t_{k+1}$, $0 \leq k \leq N-1$,

$$\mathrm{d}\overleftarrow{X}_t^\theta = \left\{ \bar{\alpha}(t)\overleftarrow{X}_t^\theta + \bar{g}^2(t)s_\theta(T-t_k, \overleftarrow{X}_{t_k}^\theta) \right\} \mathrm{d}t + \bar{g}(t)\mathrm{d}\bar{B}_t.$$

Sampling from this backward SDE is possible recursively for $k \in \{0, \dots, N-1\}$, with $(Z_k)_{1 \leq k \leq N} \overset{\mathrm{i.i.d}}{\sim} \mathcal{N}(0, I_d)$. For $k \in \{0, \dots, N-1\}$, writing $\tau_k = T - t_k$,

$$\overleftarrow{X}_{t_{k+1}}^\theta = \mathrm{e}^{-\int_{\tau_k}^{\tau_{k+1}} \alpha(s)\mathrm{d}s} \overleftarrow{X}_{t_k}^\theta + s_\theta(\tau_k, \overleftarrow{X}_{t_k}^\theta)\mathrm{e}^{-\int_{\tau_k}^{\tau_{k+1}} \alpha(s)\mathrm{d}s} \int_{\tau_k}^{\tau_{k+1}} g^2(t)\mathrm{e}^{\int_{\tau_k}^t \alpha(v)\mathrm{d}v}\mathrm{d}t$$

$$+ \left( \mathrm{e}^{-2\int_{\tau_k}^{\tau_{k+1}} \alpha(s)\mathrm{d}s} \int_{\tau_{k+1}}^{\tau_k} \mathrm{e}^{2\int_{\tau_k}^t \alpha(s)\mathrm{d}s} g^2(t)\mathrm{d}t \right)^{1/2} Z_{k+1}.$$

We denote by $\mathbb{Q}_T \in \mathcal{P}(C([0, T], \mathbb{R}^d))$ the path measure associated with the backward diffusion and by $(Q_t)_{0 \leq t \leq T}$ its Markov semi-group. We also write $\overleftarrow{X}_T^\infty \sim \pi_\infty Q_T$ and, for each time step $t_k$ for $0 \leq k \leq N$, $\overleftarrow{X}_{t_k}^\infty \sim \pi_\infty Q_{t_k}$. For each time step $t_k$ for $0 \leq k \leq N$, the kernel associated with the backward discretization is denoted by $Q_{t_k}^{N,\theta}$, so that we have $\bar{X}_{t_k}^\theta \sim \pi_\infty Q_{t_k}^{N,\theta}$.

In Appendix C, these notations are used for the specific case where $\alpha : t \mapsto \beta(t)/(2\sigma^2)$ and $g : t \mapsto \beta(t)^{1/2}$ and the associated backward discretization is given in (31).

## B  Proofs of Section 3

### B.1  Proof of Theorem 3.1

We are interested in the relative entropy of the training data distribution $\pi_{\text{data}}$ with respect to the generated data distribution $\widehat{\pi}_N^{(\beta,\theta)}$. Leveraging the time-reverse property we have:

$$\text{KL}\left(\pi_{\text{data}}\Big\|\widehat{\pi}_N^{(\beta,\theta)}\right) = \text{KL}\left(p_T Q_T\Big\|\widehat{\pi}_N^{(\beta,\theta)}\right).$$

By the data processing inequality,

$$\text{KL}\left(p_T Q_T\Big\|\widehat{\pi}_N^{(\beta,\theta)}\right) \leq \text{KL}\left(p_T \mathbb{Q}_T\Big\|\pi_\infty \mathbb{Q}_T^{N,\theta}\right).$$

where $\mathbb{Q}_T$ and $\mathbb{Q}_T^{N,\theta}$ denote the path measures of, respectively, the backward process and the SGM generation. Writing the backward time $\tau_t = T - t$ and its discretized version $\tau_k = T - t_k$, with $0 = t_0 < t_1 < \ldots < t_N = T$, we have (by Lemma B.5) that

$$\text{KL}\left(\pi_{\text{data}}\|\widehat{\pi}_N^{(\beta,\theta)}\right) \leq \text{KL}\left(p_T\|\pi_\infty\right) + \frac{1}{2}\int_0^T \frac{1}{\bar{\beta}(t)}\mathbb{E}\left[\left\|\frac{-\bar{\beta}(t)}{2\sigma^2}\overleftarrow{X}_t + \bar{\beta}(t)\nabla\log\tilde{p}_{\tau_t}\left(\overleftarrow{X}_t\right)\right.\right.$$
$$\left.\left. - \left(-\frac{\bar{\beta}(t)}{2\sigma^2}\overleftarrow{X}_t + \bar{\beta}(t)\tilde{s}_\theta\left(\tau_k, \overleftarrow{X}_{t_k}\right)\right)\right\|^2\right]\mathrm{d}t.$$

From there, the KL divergence can be split into the theoretical mixing time of the forward OU process and the approximation error for the score function made by the neural network, as follows:

$$\text{KL}\left(\pi_{\text{data}}\|\widehat{\pi}_N^{(\beta,\theta)}\right) \leq \text{KL}\left(p_T\|\pi_\infty\right) + \frac{1}{2}\int_0^T \frac{1}{\bar{\beta}(t)}\mathbb{E}\left[\left\|\bar{\beta}(t)\left(\tilde{s}\left(\tau_t, \overleftarrow{X}_t\right) - \tilde{s}_\theta(\tau_k, \overleftarrow{X}_{t_k})\right)\right\|^2\right]\mathrm{d}t.$$

By using the regular discretization of the interval $[0, T]$, one can disentangle the last term as follows:

$$\text{KL}\left(\pi_{\text{data}}\Big\|\widehat{\pi}_N^{(\beta,\theta)}\right) \leq \text{KL}\left(p_T\|\pi_\infty\right) + \frac{1}{2}\sum_{k=0}^{N-1}\int_{t_k}^{t_{k+1}}\bar{\beta}(t)\mathbb{E}\left[\left\|\tilde{s}\left(\tau_t, \overleftarrow{X}_t\right) - \tilde{s}_\theta\left(\tau_k, \overleftarrow{X}_{t_k}\right)\right\|^2\right]\mathrm{d}t$$
$$\leq E_1(\beta) + E_2(\theta,\beta) + E_3(\beta),$$

where

$$E_1(\beta) = \text{KL}\left(p_T\|\varphi_{\sigma^2}\right), \tag{13}$$

$$E_2(\theta,\beta) = \sum_{k=0}^{N-1}\int_{t_k}^{t_{k+1}}\bar{\beta}(t)\mathbb{E}\left[\left\|\tilde{s}\left(\tau_k, \overleftarrow{X}_{t_k}\right) - \tilde{s}_\theta\left(\tau_k, \overleftarrow{X}_{t_k}\right)\right\|^2\right]\mathrm{d}t, \tag{14}$$

$$E_3(\beta) = \sum_{k=0}^{N-1}\int_{t_k}^{t_{k+1}}\bar{\beta}(t)\mathbb{E}\left[\left\|\tilde{s}\left(\tau_t, \overleftarrow{X}_t\right) - \tilde{s}\left(\tau_k, \overleftarrow{X}_{t_k}\right)\right\|^2\right]\mathrm{d}t. \tag{15}$$

Finishing the proof of Theorem 3.1 amounts to obtaining upper bounds for $E_1(\beta)$, $E_2(\theta, \beta)$ and $E_3(\beta)$. This is done in Lemmas B.1, B.2 and B.3, so that $E_1(\beta) \leq \mathcal{E}_1(\beta)$, $E_2(\theta, \beta) \leq \mathcal{E}_2(\theta, \beta)$ and $E_3(\beta) \leq \mathcal{E}_3(\beta)$.

**Lemma B.1.** *For any noise schedule $\beta$,*

$$E_1(\beta) = \mathrm{KL}\left(p_T \| \pi_\infty\right) \leq \mathrm{KL}\left(\pi_{\mathrm{data}} \| \pi_\infty\right) \exp\left(-\frac{1}{\sigma^2}\int_0^T \beta(s)\mathrm{d}s\right).$$

*Proof.* The proof follows the same lines as Franzese et al. (2023, Lemma 1). The Fokker-Planck equation associated with (1) is

$$\partial_t p_t(x) = \frac{\beta(t)}{2\sigma^2}\mathrm{div}\left(xp_t(x)\right) + \frac{\beta(t)}{2}\Delta p_t(x) = \frac{\beta(t)}{2}\mathrm{div}\left(\frac{1}{\sigma^2}xp_t(x) + \nabla p_t(x)\right),$$

for $t \in [0, T], x \in \mathbb{R}^d$. Combing this with the derivation under the integral theorem, we get

$$
\begin{aligned}
\frac{\partial}{\partial t}\mathrm{KL}\left(p_t \| \varphi_{\sigma^2}\right) &= \frac{\partial}{\partial t}\int_{\mathbb{R}^d}\log\frac{p_t(x)}{\varphi_{\sigma^2}(x)}p_t(x)\mathrm{d}x \\
&= \int_{\mathbb{R}^d}\frac{\partial}{\partial t}p_t(x)\log\frac{p_t(x)}{\varphi_{\sigma^2}(x)}\mathrm{d}x + \int_{\mathbb{R}^d}\frac{p_t(x)\partial_t p_t(x)}{p_t(x)}\mathrm{d}x \\
&= \int_{\mathbb{R}^d}\frac{\partial}{\partial t}p_t(x)\log\frac{p_t(x)}{\varphi_{\sigma^2}(x)}\mathrm{d}x + \int_{\mathbb{R}^d}\frac{\partial}{\partial t}p_t(x)\mathrm{d}x \\
&= \int_{\mathbb{R}^d}\frac{\beta(t)}{2}\mathrm{div}\left(\frac{x}{\sigma^2}p_t(x) + \nabla p_t(x)\right)\log\frac{p_t(x)}{\varphi_{\sigma^2}(x)}\mathrm{d}x \\
&= \frac{\beta(t)}{2}\int_{\mathbb{R}^d}\mathrm{div}\left(-\nabla\log\varphi_{\sigma^2}(x)\,p_t(x) + \nabla p_t(x)\right)\log\frac{p_t(x)}{\varphi_{\sigma^2}(x)}\mathrm{d}x \\
&= -\frac{\beta(t)}{2}\int_{\mathbb{R}^d}\left(-\nabla\log\varphi_{\sigma^2}(x)\,p_t(x) + \nabla p_t(x)\right)^\top\nabla\log\frac{p_t(x)}{\varphi_{\sigma^2}(x)}\mathrm{d}x \\
&= -\frac{\beta(t)}{2}\int_{\mathbb{R}^d}p_t(x)\left(-\nabla\log\varphi_{\sigma^2}(x) + \nabla\log p_t(x)\right)^\top\nabla\log\frac{p_t(x)}{\varphi_{\sigma^2}(x)}\mathrm{d}x \\
&= -\frac{\beta(t)}{2}\int_{\mathbb{R}^d}p_t(x)\left\|\nabla\log\frac{p_t(x)}{\varphi_{\sigma^2}(x)}\right\|^2\mathrm{d}x.
\end{aligned}
$$

Using the Stam-Gross logarithmic Sobolev inequality given in Proposition B.6, we get

$$\frac{\partial}{\partial t}\mathrm{KL}\left(p_t \| \varphi_{\sigma^2}\right) \leq -\frac{\beta(t)}{\sigma^2}\mathrm{KL}\left(p_t \| \varphi_{\sigma^2}\right).$$

Applying Grönwall's inequality, we obtain

$$\mathrm{KL}\left(p_T \| \varphi_{\sigma^2}\right) \leq \mathrm{KL}\left(p_0 \| \varphi_{\sigma^2}\right)\exp\left\{-\frac{1}{\sigma^2}\int_0^T\beta(s)\mathrm{d}s\right\},$$

which concludes the proof. $\qquad\square$

**Lemma B.2.** *For all $\theta$ and all $\beta$,*

$$E_2(\theta, \beta) = \sum_{k=1}^{N} \mathbb{E}\left[\left\|\nabla \log \tilde{p}_{t_k}\left(\overrightarrow{X}_{t_k}\right) - \tilde{s}_\theta\left(t_k, \overrightarrow{X}_{t_k}\right)\right\|^2\right] \int_{t_k}^{t_{k+1}} \bar{\beta}(t)\mathrm{d}t \,,$$

*where $E_2(\theta, \beta)$ is defined by* (14).

*Proof.* By definition of $E_2(\theta, \beta)$,

$$\begin{aligned}
E_2(\theta, \beta) &= \sum_{k=0}^{N-1} \int_{t_k}^{t_{k+1}} \bar{\beta}(t)\mathbb{E}\left[\left\|\nabla \log \tilde{p}_{T-t_k}\left(\overleftarrow{X}_{t_k}\right) - \tilde{s}_\theta\left(T-t_k, \overleftarrow{X}_{t_k}\right)\right\|^2\right]\mathrm{d}t \\
&= \sum_{k=0}^{N-1} \mathbb{E}\left[\left\|\nabla \log \tilde{p}_{T-t_k}\left(\overleftarrow{X}_{t_k}\right) - \tilde{s}_\theta\left(T-t_k, \overleftarrow{X}_{t_k}\right)\right\|^2\right] \int_{t_k}^{t_{k+1}} \bar{\beta}(t)\mathrm{d}t \\
&= \sum_{k=0}^{N-1} \mathbb{E}\left[\left\|\nabla \log \tilde{p}_{t_k}\left(\overrightarrow{X}_{t_k}\right) - \tilde{s}_\theta\left(t_k, \overrightarrow{X}_{t_k}\right)\right\|^2\right] \int_{t_k}^{t_{k+1}} \bar{\beta}(t)\mathrm{d}t \,,
\end{aligned}$$

where the last equality comes from the fact that the forward and backward processes have same marginals since $\overrightarrow{X}_T \sim p_T$. $\qquad\square$

**Lemma B.3.** *Assume that H1 holds. For all $T, \sigma > 0$, $\theta$ and all $\beta$,*

$$E_3(\beta) \le 2h\beta(T) \max\left\{\frac{h\beta(T)}{4\sigma^2}; 1\right\} \mathcal{I}(\pi_{\mathrm{data}}|\pi_\infty) \,,$$

*where $E_3(\beta)$ is defined by* (15).

*Proof.* By Lemma B.7, with $Y_t := \nabla \log \tilde{p}_{T-t}(\overleftarrow{X}_t)$,

$$\mathrm{d}Y_t = \frac{\bar{\beta}(t)}{2\sigma^2} Y_t \mathrm{d}t + \sqrt{\bar{\beta}(t)} Z_t \mathrm{d}B_t \,.$$

By applying Itô's lemma to the function $x \mapsto \|x\|^2$, we obtain

$$\mathrm{d}\|Y_t\|^2 = \left(\frac{\bar{\beta}(t)}{\sigma^2}\|Y_t\|^2 + \bar{\beta}(t)\|Z_t\|_{\mathrm{Fr}}^2\right)\mathrm{d}t + \sqrt{\bar{\beta}(t)} Y_t^\top Z_t \mathrm{d}B_t \,.$$

Fix $\delta > 0$. From Baldi (2017, Theorem 7.3, p.193), we have that $\left(\int_0^t g(s) Y_s^\top Z_s \mathrm{d}B_s\right)_{t \in [0, T-\delta]}$ is a square integrable martingale if

$$\mathbb{E}\left[\int_0^{T-\delta} g^2(s)\left\|Y_s^\top Z_s\right\|^2 \mathrm{d}s\right] < \infty \,.$$

From the Cauchy-Schwarz inequality, we get that

$$\mathbb{E}\left[\left\|Y_s^\top Z_s\right\|_2^2\right] \le \mathbb{E}\left[\|Y_s\|_2^2 \|Z_s\|_{\mathrm{Fr}}^2\right] \le \mathbb{E}\left[\|Y_s\|_2^4\right]^{1/2} \mathbb{E}\left[\|Z_s\|_{\mathrm{Fr}}^4\right]^{1/2} \,.$$

Applying Lemma B.8 and B.9, we get that both $\mathbb{E}[\|Y_s\|_2^4]$ and $\mathbb{E}[\|Z_s\|_2^4]$ are bounded by a quantity depending on $\sigma_{T-t}^{-8}$. As the term $\sigma_{T-t}^{-8}$ is uniformly bounded in $[0, T-\delta]$ and by Fubini's theorem, $\mathbb{E}[\int_0^T g^2(s)\|Y_s^\top Z_s\|^2 \mathrm{d}s] = \int_0^T g^2(s)\mathbb{E}[\|Y_s^\top Z_s\|^2]\mathrm{d}s < \infty$. Therefore, $(\int_0^t g(s)Y_s^\top Z_s \mathrm{d}B_s)_{t\in[0,T-\delta]}$ is a square integrable martingale. Therefore, we have

$$\mathbb{E}\left[\|Y_t\|^2\right] - \mathbb{E}\left[\|Y_{t_k}\|^2\right] = \mathbb{E}\left[\int_{t_k}^t \frac{\bar{\beta}(s)}{\sigma^2}\|Y_s\|^2 \mathrm{d}s + \int_{t_k}^t \bar{\beta}(s)\|Z_s\|_{\mathrm{Fr}}^2 \mathrm{d}s\right],$$

and

$$
\begin{aligned}
\mathbb{E}\left[\|Y_t - Y_{t_k}\|^2\right] &= \mathbb{E}\left[\left\|\int_{t_k}^t \frac{\bar{\beta}(s)}{2\sigma^2}Y_s \mathrm{d}s + \int_{t_k}^t \sqrt{\bar{\beta}(s)}Z_s \mathrm{d}B_s\right\|^2\right] \\
&\leq 2\mathbb{E}\left[\left\|\int_{t_k}^t \frac{\bar{\beta}(s)}{2\sigma^2}Y_s \mathrm{d}s\right\|^2\right] + 2\mathbb{E}\left[\int_{t_k}^t \left\|\sqrt{\bar{\beta}(s)}Z_s \mathrm{d}B_s\right\|^2\right] \\
&\leq 2\mathbb{E}\left[\left\|\frac{1}{2\sigma}\int_{t_k}^t \sqrt{\bar{\beta}(s)}\frac{\sqrt{\bar{\beta}(s)}}{\sigma}Y_s \mathrm{d}s\right\|^2\right] + 2\mathbb{E}\left[\int_{t_k}^t \left\|\sqrt{\bar{\beta}(s)}Z_s \mathrm{d}B_s\right\|^2\right] \\
&\leq \frac{1}{2\sigma^2}\int_{t_k}^{t_{k+1}} \bar{\beta}(s)\mathrm{d}s\,\mathbb{E}\left[\int_{t_k}^{t_{k+1}} \frac{\bar{\beta}(s)}{\sigma^2}\|Y_s\|^2\,ds\right] + 2\mathbb{E}\left[\int_{t_k}^{t_{k+1}} \bar{\beta}(s)\|Z_s\|_{\mathrm{Fr}}^2\,\mathrm{d}s\right] \\
&\leq 2\max\left\{\frac{\int_{t_k}^{t_{k+1}}\bar{\beta}(s)\mathrm{d}s}{4\sigma^2},1\right\}\left(\mathbb{E}\left[\|Y_{t_{k+1}}\|^2\right] - \mathbb{E}\left[\|Y_{t_k}\|^2\right]\right).
\end{aligned}
\tag{16}
$$

Without loss of generality, we have that $t_{N-1} = T - \delta$. Then, the discretization error can be bounded as follows

$$
\begin{aligned}
&\sum_{k=0}^{N-1}\int_{t_k}^{t_{k+1}} \bar{\beta}(t)\mathbb{E}\left[\left\|\nabla\log\tilde{p}_{T-t}\left(\overleftarrow{X}_t\right) - \nabla\log\tilde{p}_{T-t_k}\left(\overleftarrow{X}_{t_k}\right)\right\|^2\right]\mathrm{d}t \\
&= \sum_{k=0}^{N-1}\int_{t_k}^{t_{k+1}} \bar{\beta}(t)\mathbb{E}\left[\|Y_t - Y_{t_k}\|^2\right]\mathrm{d}t \\
&\leq 2\sum_{k=0}^{N-1}\int_{t_k}^{t_{k+1}} \bar{\beta}(t)\max\left\{\frac{\int_{t_k}^{t_{k+1}}\bar{\beta}(s)\mathrm{d}s}{4\sigma^2},1\right\}\left(\mathbb{E}\left[\|Y_{t_{k+1}}\|^2\right] - \mathbb{E}\left[\|Y_{t_k}\|^2\right]\right)\mathrm{d}t \\
&\leq 2\sum_{k=0}^{N-1}\max\left\{\frac{\int_{t_k}^{t_{k+1}}\bar{\beta}(s)\mathrm{d}s}{4\sigma^2},1\right\}\left(\mathbb{E}\left[\|Y_{t_{k+1}}\|^2\right] - \mathbb{E}\left[\|Y_{t_k}\|^2\right]\right)\int_{t_k}^{t_{k+1}} \bar{\beta}(t)\mathrm{d}t \\
&\leq 2\sum_{k=0}^{N-1}\max\left\{\frac{\left(\int_{t_k}^{t_{k+1}}\bar{\beta}(s)\mathrm{d}s\right)^2}{4\sigma^2},\int_{t_k}^{t_{k+1}}\bar{\beta}(s)\mathrm{d}s\right\}\left(\mathbb{E}\left[\|Y_{t_{k+1}}\|^2\right] - \mathbb{E}\left[\|Y_{t_k}\|^2\right]\right) \\
&\leq 2\max_{0\leq k\leq N-1}\left\{\max\left\{\frac{\left(\int_{t_k}^{t_{k+1}}\bar{\beta}(s)\mathrm{d}s\right)^2}{4\sigma^2},\int_{t_k}^{t_{k+1}}\bar{\beta}(s)\mathrm{d}s\right\}\right\}\mathbb{E}\left[\left\|\nabla\log\tilde{p}_{T-t_{N-1}}\left(\overleftarrow{X}_{t_{N-1}}\right)\right\|^2\right].
\end{aligned}
$$

By H1, $t \mapsto \beta(t)$ is increasing, so that $t \mapsto \bar{\beta}(t)$ is decreasing. Therefore, using that since $\overleftarrow{X}_0 \sim p_T$, $\overleftarrow{X}_{T-\delta}$ and $\overrightarrow{X}_\delta$ have the same distribution, yields,

$$
\sum_{k=0}^{N-1} \int_{t_k}^{t_{k+1}} \bar{\beta}(t) \mathbb{E}\left[\left\|\nabla \log \tilde{p}_{T-t}\left(\overleftarrow{X}_t\right) - \nabla \log \tilde{p}_{T-t_k}\left(\overleftarrow{X}_{t_k}\right)\right\|^2\right] \mathrm{d}t
$$

$$
\leq 2 \max_{0 \leq k \leq N-1}\left\{\max\left\{\frac{\left((t_{k+1}-t_k)\bar{\beta}(t_k)\right)^2}{4\sigma^2}, (t_{k+1}-t_k)\bar{\beta}(t_k)\right\}\right\}
$$

$$
\times \mathbb{E}\left[\left\|\nabla \log \tilde{p}_{T-t_{N-1}}\left(\overleftarrow{X}_{t_{N-1}}\right)\right\|^2\right]
$$

$$
\leq 2 \max_{0 \leq k \leq N-1}\left\{\max\left\{\frac{h^2\bar{\beta}^2(t_k)}{4\sigma^2}, h\bar{\beta}(t_k)\right\}\right\} \mathcal{I}(p_T Q_{T-\delta}|\pi_\infty)
$$

$$
\leq 2h\bar{\beta}(0)\max\left\{\frac{h\bar{\beta}(0)}{4\sigma^2}, 1\right\} \mathcal{I}(p_T Q_{T-\delta}|\pi_\infty)
$$

$$
\leq 2h\beta(T)\max\left\{\frac{h\beta(T)}{4\sigma^2}, 1\right\} \mathcal{I}(p_T Q_{T-\delta}|\pi_\infty)\,.
$$

Finally, following the steps of the proof of Conforti et al. (2023, Lemma 2), we can consider the limit when $\delta$ goes to zero, under Assumption H2, concluding the proof.

$\square$

### B.2 Technical results

**Lemma B.4.** *Assume that H1 and H2 hold. Let $(\overrightarrow{X}_t)_{t \geq 0}$ be a weak solution to the forward process (1). Then, the stationary distribution of $(\overrightarrow{X}_t)_{t \geq 0}$ is Gaussian with mean 0 and variance $\sigma^2 \mathrm{I}_d$.*

*Proof.* Consider the process

$$
\bar{X}_t = \exp\left(\frac{1}{2\sigma^2}\int_0^t \beta(s)\mathrm{d}s\right) \overrightarrow{X}_t\,.
$$

Itô's formula yields

$$
\overrightarrow{X}_t = \exp\left(-\frac{1}{2\sigma^2}\int_0^t \beta(s)\mathrm{d}s\right)\left(\overrightarrow{X}_0 + \int_0^t \sqrt{\beta(s)}\exp\left(\int_0^s \beta(u)/(2\sigma^2)\mathrm{d}u\right)\mathrm{d}B_s\right). \qquad (17)
$$

First, we have that

$$
\lim_{t\to\infty} \exp\left(-\frac{1}{2\sigma^2}\int_0^t \beta(s)\mathrm{d}s\right)\overrightarrow{X}_0 = 0\,.
$$

Secondly, we have that the second term in the r.h.s. of (17), by property of the Wiener integral, is Gaussian with mean 0 and variance $\sigma_t^2 \mathrm{I}_d$, where

$$
\sigma_t^2 = \exp\left(-\frac{1}{\sigma^2}\int_0^t \beta(s)\mathrm{d}s\right)\int_0^t \beta(s)e^{\int_0^s \beta(u)/\sigma^2\mathrm{d}u}\mathrm{d}s = \sigma^2\left(1 - \exp\left(-\frac{1}{\sigma^2}\int_0^t \beta(s)\mathrm{d}s\right)\right).
$$

By H1, $\lim_{t\to\infty}\sigma_t^2 = \sigma^2$, which concludes the proof. $\square$

**Lemma B.5.** *Let $T > 0$ and $b_1, b_2 : [0, T] \times C([0, T], \mathbb{R}^d) \to \mathbb{R}^d$ be measurable functions such that for $i \in \{1, 2\}$,*

$$\mathrm{d}X_t^{(i)} = b_i \left( t, (X_s^{(i)})_{s \in [0,T]} \right) \mathrm{d}t + \sqrt{\beta(T-t)} \mathrm{d}B_t \tag{18}$$

*admits a unique strong solution with $X_0^{(i)} \sim \pi_0^{(i)}$. Suppose that $(b_i(t, (X_s^{(i)})_{s \in [0,t]}))_{t \in [0,T]}$ is progressively measurable, with Markov semi-group $(P_t^{(i)})_{t \geq 0}$. In addition, assume that*

$$\mathbb{E} \left[ \exp \left\{ \frac{1}{2} \int_0^T \frac{1}{\beta(T-s)} \left\| b_1 \left( s, \left( X_u^{(1)} \right)_{u \in [0,s]} \right) - b_2 \left( s, \left( X_u^{(1)} \right)_{u \in [0,s]} \right) \right\|^2 \mathrm{d}s \right\} \right] < \infty. \tag{19}$$

*Then,*

$$\mathrm{KL} \left( \pi_0^{(1)} P_T^{(1)} \| \pi_0^{(2)} P_T^{(2)} \right) \leq \mathrm{KL} \left( \pi_0^{(1)} \| \pi_0^{(2)} \right)$$
$$+ \frac{1}{2} \int_0^T \frac{1}{\beta(T-t)} \mathbb{E} \left[ \left\| b_1 \left( s, \left( X_u^{(1)} \right)_{u \in [0,s]} \right) - b_2 \left( s, \left( X_u^{(1)} \right)_{u \in [0,s]} \right) \right\|^2 \right] \mathrm{d}t. \tag{20}$$

*Proof.* Consider the probability space $(\Omega, (\mathcal{F}_t)_{0 \leq t \leq T}, \mathbb{P})$ and for $i \in \{1, 2\}$, let $\mu^{(i)}$ be the distribution of $(X_t^{(i)})_{t \in [0,T]}$ on the Wiener space $(C([0, T]; \mathbb{R}^d), \mathcal{B}(C([0, T]; \mathbb{R}^d)))$ with $X_0^{(i)} \sim \pi_0^{(i)}$. Define $u(t, \omega)$ as

$$u(t, \omega) := \beta(T-t)^{-1/2} \left( b_1 \left( t, \left( X_u^{(1)} \right)_{u \in [0,t]} \right) - b_2 \left( t, \left( X_u^{(1)} \right)_{u \in [0,t]} \right) \right),$$

and define $\mathrm{d}\mathbb{Q}/\mathrm{d}\mathbb{P}(\omega) = M_T(\omega)$ where, for $t \in [0, T]$,

$$M_t(\omega) = \exp \left\{ - \int_0^t u(s, \omega)^\top \mathrm{d}B_s - \frac{1}{2} \int_0^t \| u(s, \omega) \|^2 \mathrm{d}s \right\}.$$

From (19), the Novikov's condition is satisfied (Karatzas and Shreve, 2012, Chapter 3.5.D), thus the process $(M_t)_{0 \leq t \leq T}$ is a martingale. Applying Girsanov theorem, $\mathrm{d}\bar{B}_t = \mathrm{d}B_t + u(t, (X_s^{(1)})_{s \in [0,t]})\mathrm{d}t$ is a Brownian motion under the measure $\mathbb{Q}$. Therefore,

$$\mathrm{d}X_t^{(1)} = b_1 \left( t, \left( X_u^{(1)} \right)_{u \in [0,t]} \right) \mathrm{d}t + \sqrt{\beta(T-t)} \mathrm{d}B_t = b_2 \left( t, \left( X_u^{(1)} \right)_{u \in [0,t]} \right) \mathrm{d}t + \sqrt{\beta(T-t)} \mathrm{d}\bar{B}_t.$$

Using the uniqueness in law of (18), the law of $X^{(1)}$ under $\mathbb{P}$ is the same as the one of $\bar{X}^{(2)}$ under $\mathbb{Q}$, with $\bar{X}^{(2)}$ solution of (18) with $i = 2$ and $\bar{X}_0^{(2)} = \pi_0^{(1)}$. Denote by $\bar{\mu}^{(2)}$ the law of $\bar{X}^{(2)}$. Therefore,

$$\mu^{(1)}(A) = \mathbb{P}(X^{(1)} \in A) = \mathbb{Q}(\bar{X}^{(2)} \in A) = \int \mathbb{1}_A(\bar{X}^{(2)}(\omega)) \mathbb{Q}(\mathrm{d}\omega),$$

which implies that

$$\frac{\mathrm{d}\bar{\mu}^{(2)}}{\mathrm{d}\mu^{(1)}} = M_T.$$

Hence, we obtain that

$$
\begin{aligned}
\mathrm{KL}\left(\mu^{(1)}\middle\|\mu^{(2)}\right) &= \mathrm{KL}\left(\pi_0^{(1)}\middle\|\pi_0^{(2)}\right) + \mathbb{E}\left[\log\left(\frac{\mathrm{d}\mu^{(1)}}{\mathrm{d}\bar{\mu}^{(2)}}\right)\right] \\
&= \mathrm{KL}\left(\pi_0^{(1)}\middle\|\pi_0^{(2)}\right) + \mathbb{E}\left[\int_0^t u(s,\omega)^\top \mathrm{d}B_s + \frac{1}{2}\int_0^t \|u(s,\omega)\|^2 \mathrm{d}s\right] \\
&= \mathrm{KL}\left(\pi_0^{(1)}\middle\|\pi_0^{(2)}\right) + \frac{1}{2}\int_0^T \frac{1}{\beta(T-t)}\mathbb{E}\left[\left\|b_1(t,(X_s^{(1)})_{s\in[0,t]}) - b_2(t,(X_s^{(1)})_{s\in[0,t]})\right\|^2\right]\mathrm{d}t\,,
\end{aligned}
$$

which concludes the proof. $\qquad\square$

**Lemma B.6.** *Let $p$ be a probability density function on $\mathbb{R}^d$. For all $\sigma^2 > 0$,*

$$
\mathrm{KL}\left(p\|\varphi_{\sigma^2}\right) = \int p(x)\log\frac{p(x)}{\varphi_{\sigma^2}(x)}\,\mathrm{d}x \le \frac{\sigma^2}{2}\int \left\|\nabla\log\frac{p(x)}{\varphi_{\sigma^2}(x)}\right\|^2 p(x)\,\mathrm{d}x.
$$

*Proof.* Define $f_{\sigma^2} : x \mapsto p(x)/\varphi_{\sigma^2}(x)$. Since $\nabla^2 \log \varphi_{\sigma^2}(x) = -\sigma^{-2}\mathrm{I}_d$, the Bakry-Emery criterion is satisfied with constant $\sigma^{2^{-1}}$, see Bakry et al. (2014); Villani (2021); Talagrand (1996). By the classical logarithmic Sobolev inequality,

$$
\int f_{\sigma^2}(x)\log f_{\sigma^2}(x)\varphi_{\sigma^2}(x)\mathrm{d}x \le \frac{\sigma^2}{2}\int\frac{\|\nabla f_{\sigma^2}(x)\|^2}{f_{\sigma^2}(x)}\varphi_{\sigma^2}(x)\mathrm{d}x\,,
$$

which concludes the proof. $\qquad\square$

**Lemma B.7.** *Define $Y_t := \nabla\log\tilde{p}_{T-t}(\overleftarrow{X}_t)$ and $Z_t := \nabla^2\log\tilde{p}_{T-t}(\overleftarrow{X}_t)$, where $\{\overleftarrow{X}_t\}_{t\ge 0}$ is a weak solution to (10). Then,*

$$
\mathrm{d}Y_t = \left(\frac{\bar{g}^2(t)}{\sigma^2} - \bar{\alpha}(t)\right)Y_t\mathrm{d}t - \frac{2}{\sigma^2}\left(\frac{\bar{g}^2(t)}{2\sigma^2} - \bar{\alpha}(t)\right)\overleftarrow{X}_t\mathrm{d}t + \bar{g}(t)Z_t\mathrm{d}\bar{B}_t\,. \tag{21}
$$

*Proof.* The Fokker-Planck equation associated with the forward process (10) is

$$
\partial_t p_t(x) = \alpha(t)\mathrm{div}\left(xp_t(x)\right) + \frac{g^2(t)}{2}\Delta p_t(x)\,, \tag{22}
$$

for $x \in \mathbb{R}^d$. First, we prove that $\tilde{p}_t$ satisfies the following PDE

$$
\begin{aligned}
\partial_t \log\tilde{p}_t(x) = d\left(\bar{\alpha}(t) - \frac{\bar{g}^2(t)}{2\sigma^2}\right) + \langle\nabla\log\tilde{p}_t(x),x\rangle\left(\bar{\alpha}(t) - \frac{\bar{g}^2(t)}{\sigma^2}\right) + \frac{\|x\|^2}{\sigma^2}\left(\frac{\bar{g}^2(t)}{2\sigma^2} - \bar{\alpha}(t)\right) \\
+ \frac{\bar{g}^2(t)}{2}\frac{\Delta\tilde{p}_t(x)}{\tilde{p}_t(x)}\,.
\end{aligned} \tag{23}
$$

Using that $\nabla\log\varphi_{\sigma^2}(x) = -x/\sigma^2$, we have

$$
\begin{aligned}
\mathrm{div}(xp_t(x)) &= d\,p_t(x) + p_t(x)\,x^\top\nabla\log p_t(x) \\
&= \varphi_{\sigma^2}(x)\left(d\,\tilde{p}_t(x) + \tilde{p}_t(x)\nabla\log\tilde{p}_t(x)^\top x - \frac{\|x\|}{\sigma^2}\right) \\
&= \varphi_{\sigma^2}(x)\left(d\,\tilde{p}_t(x) + \nabla\tilde{p}_t(x)^\top x - \frac{\|x\|}{\sigma^2}\tilde{p}_t(x)\right)\,.
\end{aligned}
$$

Then, since $\Delta\varphi_{\sigma^2}(x) = (\varphi_{\sigma^2}(x)/\sigma^2)\left(\|x\|^2/\sigma^2 - d\right)$, we get

$$\Delta p_t(x) = \tilde{p}_t(x)\Delta\varphi_{\sigma^2}(x) + 2\nabla\tilde{p}_t(x)^\top\nabla\varphi_{\sigma^2}(x) + \varphi_{\sigma^2}(x)\Delta\tilde{p}_t(x)$$

$$= \varphi_{\sigma^2}(x)\left(\frac{\tilde{p}_t(x)}{\sigma^2}\left(\frac{\|x\|^2}{\sigma^2} - d\right) - \frac{2}{\sigma^2}\nabla\tilde{p}_t(x)^\top x + \Delta\tilde{p}_t(x)\right).$$

Combining these results with (22), we obtain

$$\partial_t\tilde{p}_t(x) = d\,\tilde{p}_t(x)\left(\alpha(t) - \frac{g^2(t)}{2\sigma^2}\right) + \nabla\tilde{p}_t(x)^\top x\left(\alpha(t) - \frac{g^2(t)}{\sigma^2}\right)$$

$$+ \tilde{p}_t(x)\frac{\|x\|^2}{\sigma^2}\left(\frac{g^2(t)}{2\sigma^2} - \alpha(t)\right) + \frac{g^2(t)}{2}\Delta\tilde{p}_t(x).$$

Hence, diving by $\tilde{p}_t$ yields (23).

The previous computation, together with the fact that $\Delta\tilde{p}_t/\tilde{p}_t = \Delta\log\tilde{p}_t + \|\nabla\log\tilde{p}_t\|^2$, yields that the function $\phi_t(x) := \log\tilde{p}_{T-t}(x)$ is a solution to the following PDE

$$\partial_t\phi_t(x) = -d\left(\bar{\alpha}(t) - \frac{\bar{g}^2(t)}{2\sigma^2}\right) - \nabla\phi_t(x)^\top x\left(\bar{\alpha}(t) - \frac{\bar{g}^2(t)}{\sigma^2}\right) \tag{24}$$

$$- \frac{\|x\|^2}{\sigma^2}\left(\frac{\bar{g}^2(t)}{2\sigma^2} - \bar{\alpha}(t)\right) - \frac{\bar{g}^2(t)}{2}\left(\Delta\phi_t(x) + \|\nabla\phi_t(x)\|^2\right). \tag{25}$$

Following the lines of Conforti et al. (2023, Proposition 1), we get that, since $\alpha$ and $g$ are continuous and non-increasing, the map $p_t$, solution to (22), belongs to $C^{1,2}((0,T]\times\mathbb{R}^d)$. By (11), as $Y_t = \nabla\phi_t(\overleftarrow{X}_t)$, we can apply Itô's formula and obtain, writing $\bar{\gamma}(t) = \bar{\alpha}(t) - \bar{g}(t)^2/\sigma^2$,

$$dY_t = \left[\partial_t\nabla\phi_t\left(\overleftarrow{X}_t\right) + \nabla^2\phi_t\left(\overleftarrow{X}_t\right)\left(\bar{\gamma}(t)\overleftarrow{X}_t + \bar{g}^2(t)\nabla\phi_t\left(\overleftarrow{X}_t\right)\right) + \frac{\bar{g}^2(t)}{2}\Delta\nabla\phi_t\left(\overleftarrow{X}_t\right)\right]dt$$

$$+ \bar{g}(t)\nabla^2\phi_t\left(\overleftarrow{X}_t\right)d\bar{B}_t$$

$$= \left[\nabla\left(\partial_t\phi_t\left(\overleftarrow{X}_t\right) + \frac{\bar{g}^2(t)}{2}\left(\Delta\phi_t\left(\overleftarrow{X}_t\right) + \left\|\nabla\phi_t\left(\overleftarrow{X}_t\right)\right\|^2\right)\right) + \bar{\gamma}(t)\nabla^2\phi_t\left(\overleftarrow{X}_t\right)\overleftarrow{X}_t\right]dt$$

$$+ \bar{g}(t)\nabla^2\phi_t\left(\overleftarrow{X}_t\right)d\bar{B}_t,$$

using that $2\nabla^2\phi_t(x)\nabla\phi_t(x) = \nabla\|\nabla\phi_t(x)\|^2$. Using (24), we get

$$dY_t = \left[-\bar{\gamma}(t)\nabla\psi_t\left(\overleftarrow{X}_t\right) + \frac{2}{\sigma^2}\left(\bar{\alpha}(t) - \frac{\bar{g}^2(t)}{2\sigma^2}\right)\overleftarrow{X}_t + \bar{\gamma}(t)\nabla^2\phi_t\left(\overleftarrow{X}_t\right)\overleftarrow{X}_t\right]dt$$

$$+ \bar{g}(t)\nabla^2\phi_t\left(\overleftarrow{X}_t\right)d\bar{B}_t,$$

with $\psi_t(x) := \nabla\phi_t(x)^\top x$. With the identity $\nabla\left(x^\top\nabla\phi_t(x)\right) = \nabla\phi_t(x) + \nabla^2\phi_t(x)x$, we have

$$dY_t = \left[\left(\frac{\bar{g}^2(t)}{\sigma^2} - \bar{\alpha}(t)\right)\nabla\phi_t\left(\overleftarrow{X}_t\right) + \frac{2}{\sigma^2}\left(\bar{\alpha}(t) - \frac{\bar{g}^2(t)}{2\sigma^2}\right)\overleftarrow{X}_t\right]dt + \bar{g}(t)\nabla^2\phi_t\left(\overleftarrow{X}_t\right)d\bar{B}_t$$

$$= \left[\left(\frac{\bar{g}^2(t)}{\sigma^2} - \bar{\alpha}(t)\right)Y_t + \frac{2}{\sigma^2}\left(\bar{\alpha}(t) - \frac{\bar{g}^2(t)}{2\sigma^2}\right)\overleftarrow{X}_t\right]dt + \bar{g}(t)Z_t d\bar{B}_t,$$

which concludes the proof. $\qquad\square$

**Lemma B.8.** *Let $Y_t := \nabla \log \tilde{p}_{T-t}(\overleftarrow{X}_t)$, with $\overleftarrow{X}$ satisfying* (11). *There exists a constant $C > 0$ such that*

$$\mathbb{E}\left[\|Y_t\|^4\right] \leq C\left(\sigma_{T-t}^{-4}\mathbb{E}\left[\|N\|^4\right] + \sigma^{-8}\mathbb{E}\left[\left\|\overrightarrow{X}_0\right\|^4\right]\right), \tag{26}$$

*with $N \sim \mathcal{N}(0, \mathrm{I}_d)$ and $\sigma_t^2$ as in* (12).

*Proof.* The transition density $q_t(y, x)$ associated with the semi-group of the process (10) is given by

$$q_t(y, x) = \left(2\pi\sigma_t^2\right)^{-d/2} \exp\left(\frac{-\left\|x - y\exp\left(-\int_0^t \alpha(s)\mathrm{d}s\right)\right\|^2}{2\sigma_t^2}\right).$$

Therefore, we have

$$\nabla \log p_{T-t}(x) = \frac{1}{p_{T-t}(x)} \int p_0(y)\nabla_x q_{T-t}(y, x)\mathrm{d}y$$

$$= \frac{1}{p_{T-t}(x)} \int p_0(y)\frac{y\exp\left(-\int_0^{T-t}\alpha(u)\mathrm{d}u\right) - x}{\sigma_{T-t}^2}q_{T-t}(y, x)\mathrm{d}y.$$

This, together with the definition of $\tilde{p}$, yields

$$\nabla \log \tilde{p}_{T-t}\left(\overrightarrow{X}_{T-t}\right) = \sigma_{T-t}^{-2}\mathbb{E}\left[\overrightarrow{X}_0 e^{-\int_0^{T-t}\alpha(u)\mathrm{d}u} - \overrightarrow{X}_{T-t} \middle| \overrightarrow{X}_{T-t}\right] + \sigma^{-2}\overrightarrow{X}_{T-t}.$$

Using Jensen's inequality for conditional expectation, there exists a constant $C > 0$ (which may change from line to line) such that

$$\left\|\nabla \log \tilde{p}_{T-t}\left(\overrightarrow{X}_{T-t}\right)\right\|^4 \leq C\left(\sigma_{T-t}^{-8}\left\|\mathbb{E}\left[\overrightarrow{X}_0 e^{-\int_0^{T-t}\alpha(s)\mathrm{d}s} - \overrightarrow{X}_{T-t}\middle|\overrightarrow{X}_{T-t}\right]\right\|^4 + \sigma^{-8}\left\|\overrightarrow{X}_{T-t}\right\|^4\right)$$

$$\leq C\left(\sigma_{T-t}^{-8}\mathbb{E}\left[\left\|\overrightarrow{X}_0 e^{-\int_0^{T-t}\alpha(s)\mathrm{d}s} - \overrightarrow{X}_{T-t}\right\|^4\middle|\overrightarrow{X}_{T-t}\right] + \sigma^{-8}\left\|\overrightarrow{X}_{T-t}\right\|^4\right).$$

Note that $\overrightarrow{X}_t$ has the same law as $\exp(-\int_0^t \alpha(s)\mathrm{d}s)\overrightarrow{X}_0 + \sigma_t N$, with $N \sim \mathcal{N}(0, \mathrm{I}_d)$. This means that we have that

$$\mathbb{E}\left[\left\|\nabla \log p_{T-t}\left(\overrightarrow{X}_{T-t}\right)\right\|^4\right] \leq C\sigma_{T-t}^{-4}\left(\mathbb{E}\left[\|N\|^4\right] + \mathbb{E}\left[\left\|\overrightarrow{X}_0\right\|^4\right]\right).$$

Finally,

$$\mathbb{E}\left[\|Y_t\|^4\right] = \mathbb{E}\left[\left\|\nabla \log \tilde{p}_{T-t}\left(\overleftarrow{X}_t\right)\right\|^2\right] = \mathbb{E}\left[\left\|\nabla \log \tilde{p}_{T-t}\left(\overrightarrow{X}_{T-t}\right)\right\|^4\right]$$

$$\leq \sigma_{T-t}^{-4}\mathbb{E}\left[\|N\|^4\right]$$

$$\leq C\left(\sigma_{T-t}^{-4}\mathbb{E}\left[\|N\|^4\right] + \sigma^{-8}\mathbb{E}\left[\left\|\overrightarrow{X}_0\right\|^4\right]\right),$$

which concludes the proof. $\qquad\square$

**Lemma B.9.** *Let $Z_t := \nabla^2 \log \tilde{p}_{T-t}(\overleftarrow{X}_t)$, where $\{\overleftarrow{X}_t\}_{t \geq 0}$ is a weak solution to (11). There exists a constant $C > 0$ such that*

$$\mathbb{E}\left[\|Z_t\|^4\right] \leq C\left(\sigma_{T-t}^{-8} + \sigma^{-8}\right)\left(\mathbb{E}\left[\|Z\|_2^8\right] + d^4\right), \tag{27}$$

*with $Z \sim \mathcal{N}(0, \mathrm{I}_d)$ and $\sigma_t^2$ as in (12).*

*Proof.* Let $q_t(y, x)$ be the transition density associated to the semi-group of the process (10). Write

$$\nabla^2 \log p_{T-t}(x)$$
$$= \nabla\left(\frac{1}{p_{T-t}(x)}\int p_0(y)\frac{ye^{-\int_0^{T-t}\alpha(s)\mathrm{d}s} - x}{\sigma_{T-t}^2}q_{T-t}(y, x)\mathrm{d}y\right)$$
$$= -\frac{\nabla p_{T-t}(x)}{p_{T-t}^2(x)}\left(\int p_0(y)\frac{ye^{-\int_0^{T-t}\alpha(s)\mathrm{d}s} - x}{\sigma_{T-t}^2}q_{T-t}(y, x)\mathrm{d}y\right)^{\top}$$
$$+ \frac{1}{p_{T-t}(x)}\nabla\int p_0(y)\frac{ye^{-\int_0^{T-t}\alpha(s)\mathrm{d}s} - x}{\sigma_{T-t}^2}q_{T-t}(y, x)\mathrm{d}y$$
$$= \frac{1}{\sigma_{T-t}^2\, p_{T-t}(x)}\left(-\int\left(\frac{\nabla p_{T-t}(x)}{p_{T-t}(x)}\right)\left(\frac{ye^{-\int_0^{T-t}\alpha(s)\mathrm{d}s} - x}{\sigma_{T-t}^2}\right)^{\top}q_{T-t}(y, x)p_0(y)\mathrm{d}y\right.$$
$$\left.- \mathrm{I}_d + \int\frac{1}{\sigma_{T-t}^2}\left(ye^{-\int_0^{T-t}\alpha(s)\mathrm{d}s} - x\right)\left(ye^{-\int_0^{T-t}\alpha(s)\mathrm{d}s} - x\right)^{\top}q_{T-t}(y, x)p_0(y)\mathrm{d}y\right).$$

Therefore,

$$\nabla^2 \log \tilde{p}_{T-t}\left(\overrightarrow{X}_{T-t}\right)$$
$$= -\frac{1}{\sigma_{T-t}^2}\left(\mathbb{E}\left[\left(\frac{\nabla p_{T-t}\left(\overrightarrow{X}_{T-t}\right)}{p_{T-t}\left(\overrightarrow{X}_{T-t}\right)}\right)\left(\overrightarrow{X}_0 e^{-\int_0^{T-t}\alpha(s)\mathrm{d}s} - \overrightarrow{X}_{T-t}\right)^{\top}\middle|\overrightarrow{X}_{T-t}\right] + \mathrm{I}_d\right)$$
$$+ \sigma_{T-t}^{-4}\mathbb{E}\left[\left(\overrightarrow{X}_0 e^{-\int_0^{T-t}\alpha(s)\mathrm{d}s} - \overrightarrow{X}_{T-t}\right)\left(\overrightarrow{X}_0 e^{-\int_0^{T-t}\alpha(s)\mathrm{d}s} - \overrightarrow{X}_{T-t}\right)^{\top}\middle|\overrightarrow{X}_{T-t}\right] + \sigma^{-2}\mathrm{I}_d$$
$$= -\sigma_{T-t}^{-4}\left(\mathbb{E}\left[\overrightarrow{X}_0 e^{-\int_0^{T-t}\alpha(s)\mathrm{d}s} - \overrightarrow{X}_{T-t}\middle|\overrightarrow{X}_{T-t}\right]\right)\left(\mathbb{E}\left[\overrightarrow{X}_0 e^{-\int_0^{T-t}\alpha(s)\mathrm{d}s} - \overrightarrow{X}_{T-t}\middle|\overrightarrow{X}_{T-t}\right]\right)^{\top}$$
$$+ \left(\sigma^{-2} - \sigma_{T-t}^{-2}\right)\mathrm{I}_d$$
$$+ \sigma_{T-t}^{-4}\mathbb{E}\left[\left(\overrightarrow{X}_0 e^{-\int_0^{T-t}\alpha(s)\mathrm{d}s} - \overrightarrow{X}_{T-t}\right)\left(\overrightarrow{X}_0 e^{-\int_0^{T-t}\alpha(s)\mathrm{d}s} - \overrightarrow{X}_{T-t}\right)^{\top}\middle|\overrightarrow{X}_{T-t}\right].$$

There exists a constant $C > 0$ (which may change from line to line) such that

$$
\mathbb{E}\left[\left\|\nabla^2 \log p_{T-t}\left(\overrightarrow{X}_{T-t}\right)\right\|_{\mathrm{Fr}}^4\right]
$$

$$
\leq \frac{C}{\sigma_{T-t}^{16}} \mathbb{E}\left[\left\|\mathbb{E}\left[\overrightarrow{X}_0 e^{-\int_0^{T-t}\alpha(s)\mathrm{d}s} - \overrightarrow{X}_{T-t}\,\middle|\,\overrightarrow{X}_{T-t}\right]\mathbb{E}\left[\overrightarrow{X}_0 e^{-\int_0^{T-t}\alpha(s)\mathrm{d}s} - \overrightarrow{X}_{T-t}\,\middle|\,\overrightarrow{X}_{T-t}\right]^\top\right\|_{\mathrm{Fr}}^4\right]
$$

$$
+ C\left(\sigma_{T-t}^{-8} + \sigma^{-8}\right)d^4
$$

$$
+ \frac{C}{\sigma_{T-t}^{16}} \mathbb{E}\left[\left\|\mathbb{E}\left[\left(\overrightarrow{X}_0 e^{-\int_0^{T-t}\alpha(s)\mathrm{d}s} - \overrightarrow{X}_{T-t}\right)\left(\overrightarrow{X}_0 e^{-\int_0^{T-t}\alpha(s)\mathrm{d}s} - \overrightarrow{X}_{T-t}\right)^\top\,\middle|\,\overrightarrow{X}_{T-t}\right]\right\|_{\mathrm{Fr}}^4\right].
$$

As in the previous proof, we note that $\overrightarrow{X}_t$ has the same law as $e^{-\int_0^t\alpha(s)\mathrm{d}s}\overrightarrow{X}_0 + \sigma_t Z$, with $Z \sim \mathcal{N}(0, \mathrm{I}_d)$ independent of $\overrightarrow{X}_0$. Therefore, using Jensen's inequality,

$$
\mathbb{E}\left[\left\|\mathbb{E}\left[\overrightarrow{X}_0 e^{-\int_0^{T-t}\alpha(s)\mathrm{d}s} - \overrightarrow{X}_{T-t}\,\middle|\,\overrightarrow{X}_{T-t}\right]\mathbb{E}\left[\overrightarrow{X}_0 e^{-\int_0^{T-t}\alpha(s)\mathrm{d}s} - \overrightarrow{X}_{T-t}\,\middle|\,\overrightarrow{X}_{T-t}\right]^\top\right\|_{\mathrm{Fr}}^4\right]
$$

$$
\leq \mathbb{E}\left[\left\|\mathbb{E}\left[\overrightarrow{X}_0 e^{-\int_0^{T-t}\alpha(s)\mathrm{d}s} - \overrightarrow{X}_{T-t}\,\middle|\,\overrightarrow{X}_{T-t}\right]\right\|_2^4 \left\|\mathbb{E}\left[\overrightarrow{X}_0 e^{-\int_0^{T-t}\alpha(s)\mathrm{d}s} - \overrightarrow{X}_{T-t}\,\middle|\,\overrightarrow{X}_{T-t}\right]\right\|_2^4\right]
$$

$$
\leq \mathbb{E}\left[\mathbb{E}\left[\left\|\overrightarrow{X}_0 e^{-\int_0^{T-t}\alpha(s)\mathrm{d}s} - \overrightarrow{X}_{T-t}\right\|_2^8\,\middle|\,\overrightarrow{X}_{T-t}\right]\right]
$$

$$
\leq \sigma_t^8 \mathbb{E}\left[\|Z\|_2^8\right]
$$

and

$$
\mathbb{E}\left[\left\|\mathbb{E}\left[\left(\overrightarrow{X}_0 e^{-\int_0^{T-t}\alpha(s)\mathrm{d}s} - \overrightarrow{X}_{T-t}\right)\left(\overrightarrow{X}_0 e^{-\int_0^{T-t}\alpha(s)\mathrm{d}s} - \overrightarrow{X}_{T-t}\right)^\top\,\middle|\,\overrightarrow{X}_{T-t}\right]\right\|_{\mathrm{Fr}}^4\right]
$$

$$
\leq \mathbb{E}\left[\mathbb{E}\left[\left\|\left(\overrightarrow{X}_0 e^{-\int_0^{T-t}\alpha(s)\mathrm{d}s} - \overrightarrow{X}_{T-t}\right)\left(\overrightarrow{X}_0 e^{-\int_0^{T-t}\alpha(s)\mathrm{d}s} - \overrightarrow{X}_{T-t}\right)^\top\right\|_{\mathrm{Fr}}^4\,\middle|\,\overrightarrow{X}_{T-t}\right]\right]
$$

$$
= \mathbb{E}\left[\left\|\left(\overrightarrow{X}_0 e^{-\int_0^{T-t}\alpha(s)\mathrm{d}s} - \overrightarrow{X}_{T-t}\right)\right\|_2^8\right]
$$

$$
\leq \sigma_t^8 \mathbb{E}\left[\|Z\|_2^8\right].
$$

Hence, we can conclude that

$$
\mathbb{E}\left[\|Z_t\|_{\mathrm{Fr}}^4\right] = \mathbb{E}\left[\left\|\nabla^2 \log \tilde{p}_{T-t}\left(\overrightarrow{X}_{T-t}\right)\right\|_{\mathrm{Fr}}^4\right] \leq C\left(\sigma_{T-t}^{-8} + \sigma^{-8}\right)\left(\mathbb{E}\left[\|Z\|_2^8\right] + d^4\right).
$$

$\square$

## C Proofs of Section 4

### C.1 Gaussian case: proof of Lemma 4.1

In the case where $\pi_{\mathrm{data}}$ is the Gaussian probability density with mean $\mu_0$ and variance $\Sigma_0$, we have

$$\nabla \log \tilde{p}_t(x) = -\left(m_t^2 \Sigma_0 + \sigma_t^2 \mathrm{I}_d\right)^{-1} (x - m_t \mu_0) + \sigma^{-2} x\,,$$

with $m_t = \exp\left(-\int_0^t \beta(s)\mathrm{d}s/(2\sigma^2)\right)$ and $\sigma_t = \sigma^2(1 - m_t^2)$. Let $\overrightarrow{\Sigma}_t = m_t^2 \Sigma_0 + \sigma_t^2 \mathrm{I}_d$ be the covariance of the forward process $\overrightarrow{X}_t$ and $b_t = \overrightarrow{\Sigma}_t^{-1} m_t \mu_0$ so that

$$\nabla \log \tilde{p}_t(x) = A_t x + b_t \quad \text{with} \quad A_t = -\left(\overrightarrow{\Sigma}_t^{-1} - \sigma^{-2} \mathrm{I}_d\right)\,. \tag{28}$$

Note that, if we denote by $\lambda_0^1 \leq \cdots \leq \lambda_0^d$ the eigenvalues of $\Sigma_0$, which are positive as $\Sigma_0$ is positive definite, we have that the eigenvalues of $A_t$ are

$$\lambda_t^i := -\frac{1}{m_t^2 \lambda_0^i + \sigma_t^2} + \frac{1}{\sigma^2}\,.$$

It is straightforward to see that $\lambda_t^1 \leq \cdots \leq \lambda_t^d$. Moreover, we always have that in this case

$$(\nabla \log \tilde{p}_t(x) - \nabla \log \tilde{p}_t(y))^\top (x - y) \leq \lambda_t^d \|x - y\|^2\,,$$
$$\|\nabla \log \tilde{p}_t(x) - \nabla \log \tilde{p}_t(y)\| \leq \max\left\{\left|\lambda_t^1\right|, \left|\lambda_t^d\right|\right\} \|x - y\|\,,$$

which entails that we can define

$$L_t := \max\left\{\left|\lambda_t^1\right|, \left|\lambda_t^d\right|\right\}\,, \qquad C_t := -\lambda_t^d\,,$$

and apply Proposition C.2.

The condition $\lambda_t^d \leq 0$, or equivalently $\sigma^2 \geq \lambda_{\max}(\Sigma_0)$, yields a contraction in 2–Wasserstein distance in the backward process as well in the forward process from Proposition C.2. This shows that, in specific cases, with an appropriate calibration of the variance of the stationary law with respect to the initial law, we have a contraction both in the forward and in the backward flows.

As a consequence, note that

$$\mathcal{W}_2\left(\pi_{\mathrm{data}}, \pi_\infty Q_T\right)^2 \leq \mathcal{W}_2\left(p_T, \pi_\infty\right)^2 \exp\left(-\frac{1}{\sigma^2}\int_0^T \beta(t)(1 + 2C_t\sigma^2)\mathrm{d}t\right)\,.$$

Using Talagrand's $T_2$ inequality for the Gaussian measure $\mathcal{W}_2\left(\mu, \pi_\infty\right)^2 \leq 2\sigma^2 \mathrm{KL}(\mu\|\pi_\infty)$ and Lemma B.1 we get

$$\mathcal{W}_2\left(\pi_{\mathrm{data}}, \pi_\infty Q_T\right)^2 \leq 2\sigma^2 \mathrm{KL}\left(\pi_{\mathrm{data}}\|\pi_\infty\right) \exp\left(-\frac{2}{\sigma^2}\int_0^T \beta(t)(1 + 2C_t\sigma^2)\mathrm{d}t\right)\,.$$

**Proposition C.1.** *Assume that $\pi_{\mathrm{data}}$ is a Gaussian distribution $\mathcal{N}(\mu_0, \Sigma_0)$ such that $\lambda_{\max}(\Sigma_0) \leq \sigma^2$ where $\lambda_{\max}(\Sigma_0)$ denotes the largest eigenvalue of $\Sigma_0$. Then,*

$$\mathrm{KL}\left(\pi_{\mathrm{data}}\|\pi_\infty Q_T\right) \leq \mathrm{KL}\left(\pi_{\mathrm{data}}\|\varphi_{\sigma^2}\right) \exp\left(-\frac{2}{\sigma^2}\int_0^T \beta(s)\mathrm{d}s\right)\,.$$

*Proof.* In this Gaussian case, the backward process is linear (see (28)) and the associated infinitesimal generator writes, for $g \in \mathcal{C}^2$,

$$\overleftarrow{\mathcal{L}}_t g(x) = \nabla g(x)^\top \left( -\frac{\bar{\beta}(t)}{2\sigma^2} + \bar{\beta}(t)(\bar{A}_t x + \bar{b}_t) \right) + \frac{1}{2}\bar{\beta}(t)\Delta g(x),$$

where $\bar{A}_t = A_{T-t}$ and $\bar{b}_t = b_{T-t}$.

Our objective is to monitor the evolution of the Kullback-Leibler divergence, $\mathrm{KL}(p_T Q_t \| \varphi_{\sigma^2} Q_t)$, for $t \in [0, T]$. We follow Del Moral et al. (2003, Section 6) (see also Collet and Malrieu, 2008). Let $q_t = p_T Q_t$ and $\phi_t = \varphi_{\sigma^2} Q_t$ two densities that satisfy the Fokker-Planck equation, involving the dual operator $\overleftarrow{\mathcal{L}}_t^*$ of the infinitesimal generator $\overleftarrow{\mathcal{L}}$

$$\partial_t q_t = \overleftarrow{\mathcal{L}}_t^* q_t, \qquad q_0(x) = p_T(x)$$
$$\partial_t \phi_t = \overleftarrow{\mathcal{L}}_t^* \phi_t, \qquad \phi_0(x) = \varphi_{\sigma^2}(x).$$

Let $f_t = q_t/\phi_t$. By definition of $\mathrm{KL}(q_t \| \phi_t) = \int \ln(f_t(x)) q_t(x)\mathrm{d}x$ we have

$$\partial_t \mathrm{KL}(q_t \| \phi_t) = \int \ln(f_t(x)) \partial_t q_t(x)\mathrm{d}x + \int \partial_t \ln(f_t(x)) q_t(x)\mathrm{d}x$$
$$= \int \ln(f_t(x)) \partial_t q_t(x)\mathrm{d}x - \int f_t(x)\partial_t \phi_t(x)\mathrm{d}x.$$

By employing the Fokker-Planck equation and the adjoint relation, which states that $\int f(x)\overleftarrow{\mathcal{L}}_t^*(g)(x)\mathrm{d}x = \int \overleftarrow{\mathcal{L}}_t f(x)g(x)\mathrm{d}x$ we obtain

$$\partial_t \mathrm{KL}(q_t \| \phi_t) = \int \overleftarrow{\mathcal{L}} \ln(f_t)(x)q_t(x)\mathrm{d}x - \int \overleftarrow{\mathcal{L}} f_t(x)\phi_t(x)\mathrm{d}x.$$

The infinitesimal generator $\overleftarrow{\mathcal{L}}$ satisfies the change of variables formula (see Bakry et al., 2014) so that

$$\overleftarrow{\mathcal{L}}_t(\ln(f)) = \frac{1}{f}\overleftarrow{\mathcal{L}}_t f - \frac{1}{2f^2}\overleftarrow{\Gamma}_t(f, f),$$

where $\overleftarrow{\Gamma}_t$ is the "carré du champ" operator associated with $\overleftarrow{\mathcal{L}}_t$ defined by $\overleftarrow{\Gamma}_t(f, f)(x) = \beta(t)|\nabla f(x)|^2$. We then obtain

$$\partial_t \mathrm{KL}(q_t \| \phi_t) = \int \overleftarrow{\mathcal{L}} f_t(x)\frac{q_t(x)}{f_t(x)}\mathrm{d}x - \int \frac{\beta(t)}{2}\frac{|\nabla f_t(x)|^2}{f_t^2(x)}q_t(x)\mathrm{d}x - \int \overleftarrow{\mathcal{L}} f_t(x)\phi_t(x)\mathrm{d}x$$
$$= -\frac{\beta(t)}{2}\int \frac{|\nabla f_t(x)|^2}{f_t(x)}\phi_t(x)\mathrm{d}x. \tag{29}$$

To obtain a control of the Kullback-Leibler divergence we need a logarithmic Sobolev inequality for the distribution of density $\phi_t = \varphi_{\sigma^2} Q_t$. In this Gaussian case, if $\overleftarrow{X}_0 \sim \mathcal{N}(0, \sigma^2)$ then for all $t \in [0, T]$ the law of $\overleftarrow{X}_t$ is a centered Gaussian with covariance matrix $\overleftarrow{\Sigma}_t$ given by

$$\overleftarrow{\Sigma}_t = \sigma^2 \exp\left( \int_0^t -\frac{\bar{\beta}(s)}{\sigma^2} + 2\bar{\beta}_s \bar{A}_s \mathrm{d}s \right) + \int_0^t \beta(s)\exp\left( \int_s^t -\frac{\bar{\beta}(u)}{\sigma^2} + 2\bar{\beta}(u)\bar{A}_u \mathrm{d}u \right) \mathrm{d}s,$$

where we use the matrix exponential. As mentioned before, if $\lambda_{\max}(\Sigma_0) \leq \sigma^2$, the eigenvalues of $A_s$, for $s \in [0, T]$, are negative. We can easily deduce that $\lambda_{\max}(\overleftarrow{\Sigma}_t) \leq \sigma^2$. We recall the logarithmic Sobolev inequality for a normal distribution (see Chafaï, 2004, Corollary 9)

$$\mathrm{KL}(q_t\|\phi_t) \leq \frac{1}{2}\int \frac{1}{f_t(x)}\nabla f_t(x)^\top \overleftarrow{\Sigma}_t \nabla f_t(x)\phi_t(x)\mathrm{d}x \leq \frac{\lambda_{\max}(\overleftarrow{\Sigma}_t)}{2}\int \frac{|\nabla f_t(x)|^2}{f_t(x)}\phi_t(x)\mathrm{d}x\,.$$

Plugging this into (29) we get

$$\partial_t\mathrm{KL}(q_t\|\phi_t) \leq -\frac{\beta(t)}{\sigma^2}\mathrm{KL}(q_t\|\phi_t)\,.$$

Therefore, recalling that $q_0 = p_T$ and $\phi_0 = \varphi_{\sigma^2}$

$$\mathrm{KL}\left(q_T\|\varphi_{\sigma^2}Q_T\right) \leq \mathrm{KL}(p_T\|\varphi_{\sigma^2})\exp\left(-\int_0^T \frac{\beta(s)}{\sigma^2}\mathrm{d}s\right)\,.$$

We conclude using Lemma B.1. $\qquad\square$

## C.2 Proof of Theorem 4.2

**EI scheme.** Using the fact that

$$\int_{t_k}^t \mathrm{e}^{-\int_s^t \bar\beta(v)/(2\sigma^2)\mathrm{d}v}\bar\beta(s)\mathrm{d}s = 2\sigma^2\left(1 - \mathrm{e}^{-\int_{t_k}^t \bar\beta(v)/(2\sigma^2)\mathrm{d}v}\right)\,,$$

the Exponential Integrator scheme that we consider consists in the following discretization, recursively given with respect to the index $k$,

$$\overleftarrow{X}_t = \mathrm{e}^{-\int_{t_k}^t \bar\beta(s)/(2\sigma^2)\mathrm{d}s}\bar X_{t_k} + 2\sigma^2\left(1 - \mathrm{e}^{-\int_{t_k}^t \bar\beta(s)/(2\sigma^2)\mathrm{d}s}\right)\nabla\log\tilde p_{T-t_k}\left(\bar X_{t_k}\right)$$
$$+ \sigma\sqrt{\left(1 - \mathrm{e}^{-\int_{t_k}^t \bar\beta(s)/\sigma^2\mathrm{d}s}\right)}Z_k\,, \quad (30)$$

where $Z_k$ are i.i.d. Gaussian random vectors $\mathcal{N}(0, I_d)$. In particular, we have that

$$\bar X_t^\theta = \mathrm{e}^{-\int_{t_k}^t \bar\beta(s)/(2\sigma^2)\mathrm{d}s}\bar X_{t_k}^\theta + 2\sigma^2\left(1 - \mathrm{e}^{-\int_{t_k}^t \bar\beta(s)/(2\sigma^2)\mathrm{d}s}\right)s_\theta\left(T - t_k, \bar X_{t_k}^\theta\right)$$
$$+ \sigma\sqrt{\left(1 - \mathrm{e}^{-\int_{t_k}^t \bar\beta(s)/\sigma^2\mathrm{d}s}\right)}Z_k\,, \quad (31)$$

and $\bar X_0^\theta \sim \mathcal{N}\left(0, \sigma^2 I_d\right)$. Note that

$$\mathcal{W}_2\left(\pi_{\mathrm{data}}, \widehat\pi_N^{(\beta,\theta)}\right) \leq \mathcal{W}_2\left(\pi_{\mathrm{data}}, \pi_\infty Q_T\right) + \mathcal{W}_2\left(\pi_\infty Q_T, \pi_\infty Q_T^{N,\theta}\right)\,, \quad (32)$$

where

$$\mathcal{W}_2\left(\pi_{\mathrm{data}}, \pi_\infty Q_T\right) = \mathcal{W}_2\left(p_T Q_T, \pi_\infty Q_T\right)\,,$$

which corresponds to the discrepancy between the same process (3) with two different initializations. The first term of (32) is upper bounded by Proposition C.2.

**Proposition C.2.** *Assume that $\mathcal{W}_2\left(\pi_{\text{data}}, \pi_\infty\right)^2 < +\infty$. The marginal distribution at the end of the forward phase satisfies*

$$\mathcal{W}_2\left(p_T, \pi_\infty\right)^2 \leq \mathcal{W}_2\left(\pi_{\text{data}}, \pi_\infty\right)^2 \exp\left(-\int_0^T \frac{\beta(t)}{\sigma^2}\mathrm{d}t\right). \tag{33}$$

*Assume that H4(ii) holds. Then,*

$$\mathcal{W}_2\left(\pi_{\text{data}}, \pi_\infty Q_T\right)^2 \leq \mathcal{W}_2\left(p_T, \pi_\infty\right)^2 \exp\left(-\int_0^T \frac{\beta(t)}{\sigma^2}\left(1 - 2L_t\sigma^2\right)\mathrm{d}t\right)$$

$$\leq \mathcal{W}_2\left(\pi_{\text{data}}, \pi_\infty\right)^2 \exp\left(-\int_0^T \frac{\beta(t)}{\sigma^2}\left(2 - 2L_t\sigma^2\right)\mathrm{d}t\right). \tag{34}$$

*Moreover, under Assumption H4(i), we have*

$$\mathcal{W}_2\left(\pi_{\text{data}}, \pi_\infty Q_T\right)^2 \leq \mathcal{W}_2\left(p_T, \pi_\infty\right)^2 \exp\left(-\int_0^T \frac{\beta(t)}{\sigma^2}\left(1 + 2C_t\sigma^2\right)\mathrm{d}t\right)$$

$$\leq \mathcal{W}_2\left(\pi_{\text{data}}, \pi_\infty\right)^2 \exp\left(-\int_0^T \frac{\beta(t)}{\sigma^2}\left(2 + 2C_t\sigma^2\right)\mathrm{d}t\right). \tag{35}$$

*Proof of Proposition C.2.* Let $x \in \mathbb{R}^d$ (resp. $y \in \mathbb{R}^d$) and denote by $\overrightarrow{X}^x$ (resp. $\overrightarrow{X}^y$) the solution of (1), with initial condition $\overrightarrow{X}_0^x = x$ (resp. $\overrightarrow{X}_0^x = y$). Applying the chain rule, we get

$$\left\|\overrightarrow{X}_t^x - \overrightarrow{X}_t^y\right\|^2 = \|x - y\|^2 + 2\int_0^t -\frac{\bar{\beta}(s)}{2\sigma^2}\left\|\overrightarrow{X}_s^x - \overrightarrow{X}_s^y\right\|^2 \mathrm{d}s.$$

Therefore, applying Grönwall's lemma, we obtain

$$\mathbb{E}\left[\sup_{t\in[0,T]}\left\|\overrightarrow{X}_t^x - \overrightarrow{X}_t^y\right\|^2\right] \leq \exp\left(-\int_0^T \frac{\bar{\beta}(t)}{\sigma^2}\mathrm{d}t\right)\|x - y\|^2.$$

From this, we can show contraction (33) in 2–Wasserstein distance by taking the infimum over all couplings.

Now, let $x \in \mathbb{R}^d$ (resp. $y \in \mathbb{R}^d$) and denote by $\overleftarrow{X}^x$ (resp. $\overleftarrow{X}^y$) the solution of (3), with initial condition $\overleftarrow{X}_0^x = x$ (resp. $\overleftarrow{X}_0^x = y$). Applying the chain rule and using Cauchy-Schwarz inequality, we get

$$\left\|\overleftarrow{X}_t^x - \overleftarrow{X}_t^y\right\|^2 = \|x - y\|^2 + 2\int_0^t -\frac{\bar{\beta}(s)}{2\sigma^2}\left\|\overleftarrow{X}_s^x - \overleftarrow{X}_s^y\right\|^2 \mathrm{d}s$$

$$+ 2\int_0^t \bar{\beta}(s)\left(\nabla\log\tilde{p}_{T-s}\left(\overleftarrow{X}_s^x\right) - \nabla\log\tilde{p}_{T-s}\left(\overleftarrow{X}_s^y\right)\right)^\top\left(\overleftarrow{X}_s^x - \overleftarrow{X}_s^y\right)\mathrm{d}s$$

$$\leq \|x - y\|^2 - \int_0^t \frac{\bar{\beta}(s)}{\sigma^2}\left(1 - 2\bar{L}_s\sigma^2\right)\left\|\overleftarrow{X}_s^x - \overleftarrow{X}_s^y\right\|^2 \mathrm{d}s.$$

Therefore, applying Grönwall's lemma, we obtain

$$\mathbb{E}\left[\sup_{t\in[0,T]}\left\|\overleftarrow{X}_t^x - \overleftarrow{X}_t^y\right\|^2\right] \leq \exp\left(-\int_0^T \frac{\bar{\beta}(t)}{\sigma^2}\left(1 - 2\bar{L}_t\sigma^2\right)\mathrm{d}t\right)\|x - y\|^2\,.$$

From this, we can show contraction (34) in 2–Wasserstein distance by taking the infimum over all couplings.

To establish (35) note that, under Assumption H4(i), we have

$$\left\|\overleftarrow{X}_t^x - \overleftarrow{X}_t^y\right\|^2 = \|x - y\|^2 + 2\int_0^t -\frac{\bar{\beta}(s)}{2\sigma^2}\left\|\overleftarrow{X}_s^x - \overleftarrow{X}_s^y\right\|^2 \mathrm{d}s$$

$$+ 2\int_0^t \bar{\beta}(s)\left(\nabla\log\tilde{p}_{T-s}\left(\overleftarrow{X}_s^x\right) - \nabla\log\tilde{p}_{T-s}\left(\overleftarrow{X}_s^y\right)\right)^\top\left(\overleftarrow{X}_s^x - \overleftarrow{X}_s^y\right)\mathrm{d}s$$

$$\leq \|x - y\|^2 - \int_0^t \frac{\bar{\beta}(s)}{\sigma^2}\left(1 + 2\bar{C}_s\sigma^2\right)\left\|\overleftarrow{X}_s^x - \overleftarrow{X}_s^y\right\|^2 \mathrm{d}s\,.$$

Therefore, applying Grönwall's lemma, we obtain

$$\mathbb{E}\left[\sup_{t\in[0,T]}\left\|\overleftarrow{X}_t^x - \overleftarrow{X}_t^y\right\|^2\right] \leq \exp\left(-\int_0^T \frac{\bar{\beta}(t)}{\sigma^2}\left(1 + 2\bar{C}_t\sigma^2\right)\mathrm{d}t\right)\|x - y\|^2\,.$$

From this, we can show contraction (35) in the 2–Wasserstein distance by taking the infimum over all couplings. $\qquad\square$

Note that a similar assumption as Assumption H4(i) is used in De Bortoli et al. (2021, Proposition 10,11,12), in particular to bound the conditional moments of $\overleftarrow{X}_0$ given $\overleftarrow{X}_t$ for $t > 0$. However, in this paper the authors also require additional assumptions, in particular that the score of $\pi_{\mathrm{data}}$ has a linear growth.

**Second term.** The second term of (36) can be handled as follows

$$\mathcal{W}_2\left(\pi_\infty Q_T, \pi_\infty Q_T^{N,\theta}\right) \leq \left\|\overleftarrow{X}_T^\infty - \bar{X}_T^\theta\right\|_{L_2}\,.$$

To upper bound $\|\overleftarrow{X}_T^\infty - \bar{X}_T^\theta\|_{L_2}$, we aim at controlling $\|\overleftarrow{X}_{t_{k+1}}^\infty - \bar{X}_{t_{k+1}}^\theta\|_{L_2}$ by $\|\overleftarrow{X}_{t_k}^\infty - \bar{X}_{t_k}^\theta\|_{L_2}$ to resort subsequently to a telescopic sum.

**Proposition C.3.** *Assume that H4, H5 and H6 hold. Consider the regular discretization $\{t_k, 0 \leq k \leq N\}$ of $[0, T]$ of constant step size $h$ such that for all $t_k$ with $0 \leq k \leq N - 1$,*

$$h < \frac{2\bar{C}_t}{\bar{\beta}(t_k)\left(\max_{t_k \leq s \leq t_{k+1}}\bar{L}_s\right)\bar{L}_t}\frac{\widetilde{m}_{t_{k+1}}}{\widetilde{m}_{t_k}}\,,$$

*where $\widetilde{m}_t := \exp(-\int_0^t \bar{\beta}(s)\mathrm{d}s/(2\sigma^2))$, $m_t := \exp(-\int_0^t \beta(s)\mathrm{d}s/(2\sigma^2))$. Then,*

$$\left\|\overleftarrow{X}_T^\infty - \bar{X}_T^\theta\right\|_{L_2} \leq \varepsilon T\beta(T) + MhT\beta(T)\left(1 + 2B\right)$$

$$+ \sum_{k=0}^{N-1}\left(\int_{t_k}^{t_{k+1}}\bar{L}_t\frac{\widetilde{m}_t}{\widetilde{m}_{t_k}}\bar{\beta}(t)\mathrm{d}t\right)\left(\frac{\sqrt{2h\beta(T)}}{\sigma} + m_T\int_{t_k}^{t_{k+1}}\left(\frac{1}{2\sigma^2} + 2\bar{L}_t\right)\bar{\beta}(t)\mathrm{d}t\right)B\,,$$

where $M$ is defined in H6 and $B := (\mathbb{E}[\|X_0\|^2] + \sigma^2 d)^{1/2}$.

*Proof.* Using (31) and the triangular inequality, we have

$$
\left\| \overleftarrow{X}^\infty_{t_{k+1}} - \bar{X}^\theta_{t_{k+1}} \right\|_{L_2}
$$

$$
= \left\| \frac{\widetilde{m}_{t_{k+1}}}{\widetilde{m}_{t_k}} \overleftarrow{X}^\infty_{t_k} - \frac{\widetilde{m}_{t_{k+1}}}{\widetilde{m}_{t_k}} \bar{X}^\theta_{t_k} + \int_{t_k}^{t_{k+1}} \frac{\widetilde{m}_t}{\widetilde{m}_{t_k}} \bar{\beta}(t) \left( \nabla \log \tilde{p}_{T-t} \left( \overleftarrow{X}^\infty_t \right) - \tilde{s}_\theta \left( T - t_k, \bar{X}^\theta_{t_k} \right) \right) dt \right\|_{L_2}
$$

$$
\leq \left\| \frac{\widetilde{m}_{t_{k+1}}}{\widetilde{m}_{t_k}} \overleftarrow{X}^\infty_{t_k} - \frac{\widetilde{m}_{t_{k+1}}}{\widetilde{m}_{t_k}} \bar{X}^\theta_{t_k} + \int_{t_k}^{t_{k+1}} \frac{\widetilde{m}_t}{\widetilde{m}_{t_k}} \bar{\beta}(t) \left( \nabla \log \tilde{p}_{T-t} \left( \overleftarrow{X}^\infty_{t_k} \right) - \nabla \log \tilde{p}_{T-t} \left( \bar{X}^\theta_{t_k} \right) \right) dt \right\|_{L_2}
$$

$$
+ \left\| \int_{t_k}^{t_{k+1}} \frac{\widetilde{m}_t}{\widetilde{m}_{t_k}} \bar{\beta}(t) \left( \nabla \log \tilde{p}_{T-t} \left( \overleftarrow{X}^\infty_t \right) - \nabla \log \tilde{p}_{T-t} \left( \overleftarrow{X}^\infty_{t_k} \right) \right) dt \right\|_{L_2} \tag{36}
$$

$$
+ \left\| \int_{t_k}^{t_{k+1}} \frac{\widetilde{m}_t}{\widetilde{m}_{t_k}} \bar{\beta}(t) \left( \nabla \log \tilde{p}_{T-t} \left( \bar{X}^\theta_{t_k} \right) - \tilde{s}_\theta \left( T - t_k, \bar{X}^\theta_{t_k} \right) \right) dt \right\|_{L_2} .
$$

Using the strong concavity and Lipschitz properties of the modified score function, we have that the first term of r.h.s. of (36) can be bounded as follows

$$
\left\| \frac{\widetilde{m}_{t_{k+1}}}{\widetilde{m}_{t_k}} \overleftarrow{X}^\infty_{t_k} - \frac{\widetilde{m}_{t_{k+1}}}{\widetilde{m}_{t_k}} \bar{X}^\theta_{t_k} + \int_{t_k}^{t_{k+1}} \frac{\widetilde{m}_t}{\widetilde{m}_{t_k}} \bar{\beta}(t) \left( \nabla \log \tilde{p}_{T-t} \left( \overleftarrow{X}^\infty_{t_k} \right) - \nabla \log \tilde{p}_{T-t} \left( \bar{X}^\theta_{t_k} \right) \right) dt \right\|^2
$$

$$
= \frac{\widetilde{m}^2_{t_{k+1}}}{\widetilde{m}^2_{t_k}} \left\| \overleftarrow{X}^\infty_{t_k} - \bar{X}^\theta_{t_k} \right\|^2 + \left\| \int_{t_k}^{t_{k+1}} \frac{\widetilde{m}_t}{\widetilde{m}_{t_k}} \bar{\beta}(t) \left( \nabla \log \tilde{p}_{T-t} \left( \overleftarrow{X}^\infty_{t_k} \right) - \nabla \log \tilde{p}_{T-t} \left( \bar{X}^\theta_{t_k} \right) \right) dt \right\|^2
$$

$$
+ \frac{\widetilde{m}_{t_{k+1}}}{\widetilde{m}_{t_k}} 2 \int_{t_k}^{t_{k+1}} \frac{\widetilde{m}_t}{\widetilde{m}_{t_k}} \bar{\beta}(t) \left[ \overleftarrow{X}^\infty_{t_k} - \bar{X}^\theta_{t_k} \right]^\top \left[ \nabla \log \tilde{p}_{T-t} \left( \overleftarrow{X}^\infty_{t_k} \right) - \nabla \log \tilde{p}_{T-t} \left( \bar{X}^\theta_{t_k} \right) dt \right]
$$

$$
\leq \left\| \overleftarrow{X}^\infty_{t_k} - \bar{X}^\theta_{t_k} \right\|^2 \left( \frac{\widetilde{m}^2_{t_{k+1}}}{\widetilde{m}^2_{t_k}} + \left( \int_{t_k}^{t_{k+1}} \bar{L}_t \frac{\widetilde{m}_t}{\widetilde{m}_{t_k}} \bar{\beta}(t) dt \right)^2 - 2 \frac{\widetilde{m}_{t_{k+1}}}{\widetilde{m}_{t_k}} \int_{t_k}^{t_{k+1}} \bar{C}_t \frac{\widetilde{m}_t}{\widetilde{m}_{t_k}} \bar{\beta}(t) dt \right) .
$$

Using the Lipschitz property of the modified score and Proposition C.6, the second term of the r.h.s. of (36) can be controlled as follows

$$
\left\| \int_{t_k}^{t_{k+1}} \frac{\widetilde{m}_t}{\widetilde{m}_{t_k}} \bar{\beta}(t) \left( \nabla \log \tilde{p}_{T-t} \left( \overleftarrow{X}^\infty_t \right) - \nabla \log \tilde{p}_{T-t} \left( \overleftarrow{X}^\infty_{t_k} \right) \right) dt \right\|_{L_2}
$$

$$
\leq \left( \int_{t_k}^{t_{k+1}} L_{T-t} \frac{\widetilde{m}_t}{\widetilde{m}_{t_k}} \bar{\beta}(t) dt \right) \sup_{t_k \leq t \leq t_{k+1}} \left\| \overleftarrow{X}^\infty_t - \overleftarrow{X}^\infty_{t_k} \right\|_{L_2}
$$

$$
\leq \left( \int_{t_k}^{t_{k+1}} \bar{L}_t \frac{\widetilde{m}_t}{\widetilde{m}_{t_k}} \bar{\beta}(t) dt \right) 2 \left( \frac{1}{\sigma} \sqrt{h \beta(T)} + \exp \left( - \int_0^{t_k} \frac{\bar{\beta}(s)}{\sigma^2} \left( 1 + \bar{C}_s \sigma^2 \right) ds \right) \right) B .
$$

Using Assumption H5, we can control the third term of the r.h.s. of (36) as follows

$$\left\| \int_{t_k}^{t_{k+1}} \frac{\widetilde{m}_t}{\widetilde{m}_{t_k}} \bar{\beta}(t) \left( \nabla \log \tilde{p}_{T-t} \left( \bar{X}_{t_k}^\theta \right) - \tilde{s}_\theta \left( T - t_k, \bar{X}_{t_k}^\theta \right) \right) dt \right\|_{L_2}$$

$$\leq \left\| \int_{t_k}^{t_{k+1}} \frac{\widetilde{m}_t}{\widetilde{m}_{t_k}} \bar{\beta}(t) \left( \nabla \log \tilde{p}_{T-t_k} \left( \bar{X}_{t_k}^\theta \right) - \tilde{s}_\theta \left( T - t_k, \bar{X}_{t_k}^\theta \right) \right) dt \right\|_{L_2}$$

$$+ \left\| \int_{t_k}^{t_{k+1}} \frac{\widetilde{m}_t}{\widetilde{m}_{t_k}} \bar{\beta}(t) \left( \nabla \log \tilde{p}_{T-t} \left( \bar{X}_{t_k}^\theta \right) - \nabla \log \tilde{p}_{T-t_k} \left( \bar{X}_{t_k}^\theta \right) \right) dt \right\|_{L_2}$$

$$\leq \varepsilon \int_{t_k}^{t_{k+1}} \frac{\widetilde{m}_t}{\widetilde{m}_{t_k}} \bar{\beta}(t) dt + \int_{t_k}^{t_{k+1}} \frac{\widetilde{m}_t}{\widetilde{m}_{t_k}} \bar{\beta}(t) \left\| \nabla \log \tilde{p}_{T-t} \left( \bar{X}_{t_k}^\theta \right) - \nabla \log \tilde{p}_{T-t_k} \left( \bar{X}_{t_k}^\theta \right) \right\| dt$$

$$\leq \varepsilon \int_{t_k}^{t_{k+1}} \frac{\widetilde{m}_t}{\widetilde{m}_{t_k}} \bar{\beta}(t) dt + hM \left( 1 + \left\| \bar{X}_{t_k}^\theta \right\|_{L_2} \right) \int_{t_k}^{t_{k+1}} \frac{\widetilde{m}_t}{\widetilde{m}_{t_k}} \bar{\beta}(t) dt \,.$$

Note that $\overrightarrow{X}_t$ has the same law as $m_t X_0 + \sigma \sqrt{(1 - m_t^2)} G$, with $G$ a standard Gaussian random variable independent of $X_0$. We have that $\overleftarrow{X}_0^\infty \sim \mathcal{N}(0, \sigma^2 I_d)$. Define $(\overleftarrow{X}_t)_{t \in [0,T]}$ satisfying (3) but initialized at

$$\overleftarrow{X}_0 = m_T X_0 + \sqrt{(1 - m_T^2)} \overleftarrow{X}_0^\infty \,,$$

with $X_0 \sim \pi_{\text{data}}$. Employing Proposition C.5 and (42), we obtain

$$\left\| \bar{X}_{t_k}^\theta \right\|_{L_2} \leq \left\| \bar{X}_{t_k}^\theta - \overleftarrow{X}_{t_k}^\infty \right\|_{L_2} + \left\| \overleftarrow{X}_{t_k}^\infty - \overleftarrow{X}_{t_k} \right\|_{L_2} + \left\| \overleftarrow{X}_{t_k} \right\|_{L_2} \leq \left\| \bar{X}_{t_k}^\theta - \overleftarrow{X}_{t_k}^\infty \right\|_{L_2} + 2B \,.$$

Therefore, combining the previous bounds, together with (36), we obtain

$$\left\| \overleftarrow{X}_{t_{k+1}}^\infty - \bar{X}_{t_{k+1}}^\theta \right\|_{L_2}$$

$$\leq \left\| \overleftarrow{X}_{t_k}^\infty - \bar{X}_{t_k}^\theta \right\| \left( \frac{\widetilde{m}_{t_{k+1}}^2}{\widetilde{m}_{t_k}^2} + \left( \int_{t_k}^{t_{k+1}} \bar{L}_t \frac{\widetilde{m}_t}{\widetilde{m}_{t_k}} \bar{\beta}(t) dt \right)^2 - 2 \frac{\widetilde{m}_{t_{k+1}}}{\widetilde{m}_{t_k}} \int_{t_k}^{t_{k+1}} \bar{C}_t \frac{\widetilde{m}_t}{\widetilde{m}_{t_k}} \bar{\beta}(t) dt \right)^{1/2}$$

$$+ \left( \int_{t_k}^{t_{k+1}} \bar{L}_t \frac{\widetilde{m}_t}{\widetilde{m}_{t_k}} \bar{\beta}(t) dt \right) \left( \frac{1}{\sigma} \sqrt{2h\beta(T)} + m_T \int_{t_k}^{t_{k+1}} \left( \frac{1}{2\sigma^2} + 2\bar{L}_t \right) \bar{\beta}(t) dt \right) B$$

$$+ \varepsilon \int_{t_k}^{t_{k+1}} \frac{\widetilde{m}_t}{\widetilde{m}_{t_k}} \bar{\beta}(t) dt + hM \left( 1 + \left\| \bar{X}_{t_k}^\theta - \overleftarrow{X}_{t_k}^\infty \right\|_{L_2} + 2B \right) \int_{t_k}^{t_{k+1}} \frac{\widetilde{m}_t}{\widetilde{m}_{t_k}} \bar{\beta}(t) dt \,.$$

By the assumption on $h$ and Proposition C.4,

$$0 < 1 + \frac{\widetilde{m}_{t_k}^2}{\widetilde{m}_{t_{k+1}}^2} \left( \int_{t_k}^{t_{k+1}} \bar{L}_t \frac{\widetilde{m}_t}{\widetilde{m}_{t_k}} \bar{\beta}(t) dt \right)^2 - 2 \frac{\widetilde{m}_{t_k}}{\widetilde{m}_{t_{k+1}}} \int_{t_k}^{t_{k+1}} \bar{C}_t \frac{\widetilde{m}_t}{\widetilde{m}_{t_k}} \bar{\beta}(t) dt < 1 \,,$$

and, using that $\sqrt{1-x} \leq 1 - x/2$ for $x \in [0,1]$, we conclude that

$$
\left\| \overleftarrow{X}_{t_{k+1}}^{\infty} - \bar{X}_{t_{k+1}}^{\theta} \right\|_{L_2}
$$
$$
\leq \left\| \overleftarrow{X}_{t_k}^{\infty} - \bar{X}_{t_k}^{\theta} \right\| \frac{\widetilde{m}_{t_{k+1}}^2}{\widetilde{m}_{t_k}^2}
$$
$$
\times \left( 1 + \frac{1}{2} \frac{\widetilde{m}_{t_k}^2}{\widetilde{m}_{t_{k+1}}^2} \left( \int_{t_k}^{t_{k+1}} \bar{L}_t \frac{\widetilde{m}_t}{\widetilde{m}_{t_k}} \bar{\beta}(t)\mathrm{d}t \right)^2 - \frac{\widetilde{m}_{t_k}}{\widetilde{m}_{t_{k+1}}} \int_{t_k}^{t_{k+1}} \bar{C}_t \frac{\widetilde{m}_t}{\widetilde{m}_{t_k}} \bar{\beta}(t)\mathrm{d}t + \right.
$$
$$
\left. \frac{\widetilde{m}_{t_k}^2}{\widetilde{m}_{t_{k+1}}^2} M h \int_{t_k}^{t_{k+1}} \frac{\widetilde{m}_t}{\widetilde{m}_{t_k}} \bar{\beta}(t)\mathrm{d}t \right)
$$
$$
+ \left( \int_{t_k}^{t_{k+1}} \bar{L}_t \frac{\widetilde{m}_t}{\widetilde{m}_{t_k}} \bar{\beta}(t)\mathrm{d}t \right) \left( \frac{1}{\sigma}\sqrt{2h\beta(T)} + m_T \int_{t_k}^{t_{k+1}} \left( \frac{1}{2\sigma^2} + 2\bar{L}_t \right) \bar{\beta}(t)\mathrm{d}t \right) B
$$
$$
+ \varepsilon \int_{t_k}^{t_{k+1}} \frac{\widetilde{m}_t}{\widetilde{m}_{t_k}} \bar{\beta}(t)\mathrm{d}t + h M (1 + 2B) \int_{t_k}^{t_{k+1}} \frac{\widetilde{m}_t}{\widetilde{m}_{t_k}} \bar{\beta}(t)\mathrm{d}t \, .
$$

Define

$$
\delta_k := \frac{\widetilde{m}_{t_{k+1}}^2}{\widetilde{m}_{t_k}^2} \left( 1 + \frac{1}{2} \frac{\widetilde{m}_{t_k}^2}{\widetilde{m}_{t_{k+1}}^2} \left( \int_{t_k}^{t_{k+1}} \bar{L}_t \frac{\widetilde{m}_t}{\widetilde{m}_{t_k}} \bar{\beta}(t)\mathrm{d}t \right)^2 - \frac{\widetilde{m}_{t_k}}{\widetilde{m}_{t_{k+1}}} \int_{t_k}^{t_{k+1}} \bar{C}_t \frac{\widetilde{m}_t}{\widetilde{m}_{t_k}} \bar{\beta}(t)\mathrm{d}t \right.
$$
$$
\left. + \frac{\widetilde{m}_{t_k}^2}{\widetilde{m}_{t_{k+1}}^2} M h \int_{t_k}^{t_{k+1}} \frac{\widetilde{m}_t}{\widetilde{m}_{t_k}} \bar{\beta}(t)\mathrm{d}t \right)
$$
$$
\leq \left( 1 + \frac{1}{2} \frac{\widetilde{m}_{t_k}^2}{\widetilde{m}_{t_{k+1}}^2} \left( \int_{t_k}^{t_{k+1}} \bar{L}_t \frac{\widetilde{m}_t}{\widetilde{m}_{t_k}} \bar{\beta}(t)\mathrm{d}t \right)^2 - \frac{\widetilde{m}_{t_k}}{\widetilde{m}_{t_{k+1}}} \int_{t_k}^{t_{k+1}} \bar{C}_t \frac{\widetilde{m}_t}{\widetilde{m}_{t_k}} \bar{\beta}(t)\mathrm{d}t \right.
$$
$$
\left. + \frac{\widetilde{m}_{t_k}^2}{\widetilde{m}_{t_{k+1}}^2} M h \int_{t_k}^{t_{k+1}} \frac{\widetilde{m}_t}{\widetilde{m}_{t_k}} \bar{\beta}(t)\mathrm{d}t \right) \, .
$$

By Proposition C.4, $\delta_k \leq 1$ for any $0 \leq k \leq N-1$ , which yields

$$
\left\| \overleftarrow{X}_T^{\infty} - \bar{X}_T^{\theta} \right\|_{L_2} \leq \prod_{k=0}^{N-1} \delta_k \left\| \overleftarrow{X}_0^{\infty} - \bar{X}_0^{\theta} \right\|_{L_2} + \left( \varepsilon h \beta(T) + M h^2 \beta(T) (1 + 2B) \right) \sum_{k=0}^{N-1} \prod_{\ell=k}^{N-1} \delta_\ell
$$
$$
+ \sum_{k=0}^{N-1} \left( \int_{t_k}^{t_{k+1}} \bar{L}_t \frac{\widetilde{m}_t}{\widetilde{m}_{t_k}} \bar{\beta}(t)\mathrm{d}t \right) \left( \frac{1}{\sigma}\sqrt{2h\beta(T)} + m_T \int_{t_k}^{t_{k+1}} \left( \frac{1}{2\sigma^2} + 2\bar{L}_t \right) \bar{\beta}(t)\mathrm{d}t \right) B \prod_{\ell=k}^{N-1} \delta_\ell
$$
$$
\leq \varepsilon T \beta(T) + M h T \beta(T) (1 + 2B)
$$
$$
+ \sum_{k=0}^{N-1} \left( \int_{t_k}^{t_{k+1}} \bar{L}_t \frac{\widetilde{m}_t}{\widetilde{m}_{t_k}} \bar{\beta}(t)\mathrm{d}t \right) \left( \frac{1}{\sigma}\sqrt{2h\beta(T)} + m_T \int_{t_k}^{t_{k+1}} \left( \frac{1}{2\sigma^2} + 2\bar{L}_t \right) \bar{\beta}(t)\mathrm{d}t \right) B \, .
$$

$\square$

**Final bound.** Finally, combining the results of Proposition C.2 and Proposition C.3, we conclude that

$$
\mathcal{W}_2\left(\pi_{\text{data}}, \widehat{\pi}_N^{(\beta,\theta)}\right) \leq \mathcal{W}_2\left(\pi_{\text{data}}, \pi_\infty\right) \exp\left(-\int_0^T \frac{\beta(t)}{\sigma^2}\left(1 + C_t \sigma^2\right) \mathrm{d}t\right)
$$
$$
+ \sum_{k=0}^{N-1}\left(\int_{t_k}^{t_{k+1}} \bar{L}_t \frac{\widetilde{m}_t}{\widetilde{m}_{t_k}} \bar{\beta}(t)\mathrm{d}t\right)\left(\frac{1}{\sigma}\sqrt{2h\beta(T)} + m_T \int_{t_k}^{t_{k+1}}\left(\frac{1}{2\sigma^2} + 2\bar{L}_t\right)\bar{\beta}(t)\mathrm{d}t\right) B
$$
$$
+ \varepsilon T \beta(T) + MhT\beta(T)\left(1 + 2B\right).
$$

## C.3 Technical results for Wasserstein upper bound

**Proposition C.4.** *Assume that H4 and H6 hold. Consider the regular discretization $\{t_k, 0 \leq k \leq N\}$ of $[0, T]$ of constant step size $h$. Assume that $h > 0$ is such that for all $t_k$ with $0 \leq k \leq N-1$,*

$$
h < \frac{2\bar{C}_t}{\bar{\beta}(t_k)\left(\max_{t_k \leq s \leq t_{k+1}} \bar{L}_s\right) \bar{L}_t} \frac{\widetilde{m}_{t_{k+1}}}{\widetilde{m}_{t_k}}, \tag{37}
$$

*where $\widetilde{m}_t := \exp(-\int_0^t \bar{\beta}(s)\mathrm{d}s/(2\sigma^2))$, $m_t := \exp(-\int_0^t \beta(s)\mathrm{d}s/(2\sigma^2))$. Then, for all $0 \leq k \leq N-1$,*

$$
0 < 1 + \frac{\widetilde{m}_{t_k}^2}{\widetilde{m}_{t_{k+1}}^2}\left(\int_{t_k}^{t_{k+1}} \bar{L}_t \frac{\widetilde{m}_t}{\widetilde{m}_{t_k}} \bar{\beta}(t)\mathrm{d}t\right)^2 - 2\frac{\widetilde{m}_{t_k}}{\widetilde{m}_{t_{k+1}}}\int_{t_k}^{t_{k+1}} \bar{C}_t \frac{\widetilde{m}_t}{\widetilde{m}_{t_k}} \bar{\beta}(t)\mathrm{d}t < 1.
$$

*In addition, if*

$$
h < \frac{2\bar{C}_t}{M + \bar{\beta}(t_k)\left(\max_{t_k \leq s \leq t_{k+1}} \bar{L}_s\right) \bar{L}_t} \frac{\widetilde{m}_{t_{k+1}}}{\widetilde{m}_{t_k}}, \tag{38}
$$

*then, for all $0 \leq k \leq N-1$,*

$$
0 < 1 + \frac{1}{2}\frac{\widetilde{m}_{t_k}^2}{\widetilde{m}_{t_{k+1}}^2}\left(\int_{t_k}^{t_{k+1}} \bar{L}_t \frac{\widetilde{m}_t}{\widetilde{m}_{t_k}} \bar{\beta}(t)\mathrm{d}t\right)^2 - \frac{\widetilde{m}_{t_k}}{\widetilde{m}_{t_{k+1}}}\int_{t_k}^{t_{k+1}} \bar{C}_t \frac{\widetilde{m}_t}{\widetilde{m}_{t_k}} \bar{\beta}(t)\mathrm{d}t
$$
$$
+ \frac{\widetilde{m}_{t_k}^2}{\widetilde{m}_{t_{k+1}}^2} Mh \int_{t_k}^{t_{k+1}} \frac{\widetilde{m}_t}{\widetilde{m}_{t_k}} \bar{\beta}(t)\mathrm{d}t < 1.
$$

*Proof.* Denote $\epsilon_1$ and $\epsilon_2$ the following quantities

$$
\epsilon_1 = 1 + \frac{\widetilde{m}_{t_k}^2}{\widetilde{m}_{t_{k+1}}^2}\left(\int_{t_k}^{t_{k+1}} \bar{L}_t \frac{\widetilde{m}_t}{\widetilde{m}_{t_k}} \bar{\beta}(t)\mathrm{d}t\right)^2 - 2\frac{\widetilde{m}_{t_k}}{\widetilde{m}_{t_{k+1}}}\int_{t_k}^{t_{k+1}} \bar{C}_t \frac{\widetilde{m}_t}{\widetilde{m}_{t_k}} \bar{\beta}(t)\mathrm{d}t, \tag{39}
$$
$$
\epsilon_2 = 1 + \frac{\widetilde{m}_{t_k}^2}{\widetilde{m}_{t_{k+1}}^2}\left(\int_{t_k}^{t_{k+1}} \bar{L}_t \frac{\widetilde{m}_t}{\widetilde{m}_{t_k}} \bar{\beta}(t)\mathrm{d}t\right)^2 - 2\frac{\widetilde{m}_{t_k}}{\widetilde{m}_{t_{k+1}}}\int_{t_k}^{t_{k+1}} \bar{C}_t \frac{\widetilde{m}_t}{\widetilde{m}_{t_k}} \bar{\beta}(t)\mathrm{d}t
$$
$$
+ \frac{\widetilde{m}_{t_k}^2}{\widetilde{m}_{t_{k+1}}^2} Mh \int_{t_k}^{t_{k+1}} \frac{\widetilde{m}_t}{\widetilde{m}_{t_k}} \bar{\beta}(t)\mathrm{d}t. \tag{40}
$$

First, we prove that $\epsilon_1$ is positive. Completing the square, we obtain

$$\epsilon_1 = \left(1 - \frac{\widetilde{m}_{t_k}}{\widetilde{m}_{t_{k+1}}}\int_{t_k}^{t_{k+1}} \bar{L}_t \frac{\widetilde{m}_t}{\widetilde{m}_{t_k}}\bar{\beta}(t)\mathrm{d}t\right)^2 + 2\frac{\widetilde{m}_{t_k}}{\widetilde{m}_{t_{k+1}}}\int_{t_k}^{t_{k+1}} \bar{L}_t \frac{\widetilde{m}_t}{\widetilde{m}_{t_k}}\bar{\beta}(t)\mathrm{d}t$$
$$- 2\frac{\widetilde{m}_{t_k}}{\widetilde{m}_{t_{k+1}}}\int_{t_k}^{t_{k+1}} \bar{C}_t \frac{\widetilde{m}_t}{\widetilde{m}_{t_k}}\bar{\beta}(t)\mathrm{d}t$$
$$= \left(1 - \frac{\widetilde{m}_{t_k}}{\widetilde{m}_{t_{k+1}}}\int_{t_k}^{t_{k+1}} \bar{L}_t \frac{\widetilde{m}_t}{\widetilde{m}_{t_k}}\bar{\beta}(t)\mathrm{d}t\right)^2 + 2\frac{\widetilde{m}_{t_k}}{\widetilde{m}_{t_{k+1}}}\int_{t_k}^{t_{k+1}} \left(\bar{L}_t - \bar{C}_t\right)\frac{\widetilde{m}_t}{\widetilde{m}_{t_k}}\bar{\beta}(t)\mathrm{d}t.$$

The first term if the r.h.s. of the previous equality is a square, therefore always positive. The second term is always positive as well, as $\bar{L}_t \geq \bar{C}_t$ for any $t$, as the Lipschitz constant and the log-concavity coefficient of the score function respectively. Moreover, the previous is always strictly positive as

$$\frac{\widetilde{m}_{t_k}}{\widetilde{m}_{t_{k+1}}}\int_{t_k}^{t_{k+1}} \bar{L}_t \frac{\widetilde{m}_t}{\widetilde{m}_{t_k}}\bar{\beta}(t)\mathrm{d}t > 0\,.$$

Secondly, proving that the previous quantity is smaller than 1 is equivalent to show that

$$\frac{\widetilde{m}_{t_k}^2}{\widetilde{m}_{t_{k+1}}^2}\left(\int_{t_k}^{t_{k+1}} \bar{L}_t \frac{\widetilde{m}_t}{\widetilde{m}_{t_k}}\bar{\beta}(t)\mathrm{d}t\right)^2 - 2\frac{\widetilde{m}_{t_k}}{\widetilde{m}_{t_{k+1}}}\int_{t_k}^{t_{k+1}} \bar{C}_t \frac{\widetilde{m}_t}{\widetilde{m}_{t_k}}\bar{\beta}(t)\mathrm{d}t < 0\,.$$

As $\bar{\beta}(t)$ is a decreasing function, we obtain the following bound

$$\frac{\widetilde{m}_{t_k}^2}{\widetilde{m}_{t_{k+1}}^2}\left(\int_{t_k}^{t_{k+1}} \bar{L}_t \frac{\widetilde{m}_t}{\widetilde{m}_{t_k}}\bar{\beta}(t)\mathrm{d}t\right)^2 - 2\frac{\widetilde{m}_{t_k}}{\widetilde{m}_{t_{k+1}}}\int_{t_k}^{t_{k+1}} \bar{C}_t \frac{\widetilde{m}_t}{\widetilde{m}_{t_k}}\bar{\beta}(t)\mathrm{d}t$$
$$\leq \left(\frac{\widetilde{m}_{t_k}}{\widetilde{m}_{t_{k+1}}}\max_{t_k \leq s \leq t_{k+1}} \bar{L}_s\bar{\beta}(t_k)h\right)\frac{\widetilde{m}_{t_k}}{\widetilde{m}_{t_{k+1}}}\int_{t_k}^{t_{k+1}} \bar{L}_t \frac{\widetilde{m}_t}{\widetilde{m}_{t_k}}\bar{\beta}(t)\mathrm{d}t - 2\frac{\widetilde{m}_{t_k}}{\widetilde{m}_{t_{k+1}}}\int_{t_k}^{t_{k+1}} \bar{C}_t \frac{\widetilde{m}_t}{\widetilde{m}_{t_k}}\bar{\beta}(t)\mathrm{d}t$$

$$= \frac{\widetilde{m}_{t_k}}{\widetilde{m}_{t_{k+1}}}\int_{t_k}^{t_{k+1}} \left(\left(\frac{\widetilde{m}_{t_k}}{\widetilde{m}_{t_{k+1}}}\max_{t_k \leq s \leq t_{k+1}} \bar{L}_s\bar{\beta}(t_k)h\right)\bar{L}_t - 2\bar{C}_t\right)\frac{\widetilde{m}_t}{\widetilde{m}_{t_k}}\bar{\beta}(t)\mathrm{d}t\,.$$

This means that, if we have

$$\frac{\widetilde{m}_{t_k}}{\widetilde{m}_{t_{k+1}}}\left(\max_{t_k \leq s \leq t_{k+1}} \bar{L}_s\right)\bar{\beta}(t_k)h\bar{L}_t - 2\bar{C}_t < 0$$

for $t_k \leq t \leq t_{k+1}$, we have $\epsilon_1 < 1$. Isolating $h$ in the previous inequality, we obtain that it is equivalent to the condition (37).

Now we focus on $\epsilon_2$. This quantity is clearly positive as the $\epsilon_2 \geq \epsilon_1$. Moreover, following the same lines as to prove that $\epsilon_1 < 1$, we have

$$\epsilon_2 - 1 \leq \frac{\widetilde{m}_{t_k}}{\widetilde{m}_{t_{k+1}}}\int_{t_k}^{t_{k+1}} \left(\frac{\widetilde{m}_{t_k}}{\widetilde{m}_{t_{k+1}}}\left(\max_{t_k \leq s \leq t_{k+1}} \bar{L}_s\right)\bar{\beta}(t_k)h\bar{L}_t + \frac{\widetilde{m}_{t_k}}{\widetilde{m}_{t_{k+1}}}Mh - 2\bar{C}_t\right)\frac{\widetilde{m}_t}{\widetilde{m}_{t_k}}\bar{\beta}(t)\mathrm{d}t\,.$$

This means that, if we have

$$\frac{\widetilde{m}_{t_k}}{\widetilde{m}_{t_{k+1}}} \left( \max_{t_k \leq s \leq t_{k+1}} \bar{L}_s \right) \bar{\beta}(t_k) h \bar{L}_t + \frac{\widetilde{m}_{t_k}}{\widetilde{m}_{t_{k+1}}} M h - 2\bar{C}_t < 0$$

for $t_k \leq t \leq t_{k+1}$, we have $\epsilon_2 < 1$. Isolating $h$ in the previous inequality, we obtain that it is equivalent to the condition (38).

$\square$

**Proposition C.5.** *Assume that H2 holds. For all $t \geq 0$,*

$$\sup_{0 \leq t \leq T} \left\| \overleftarrow{X}_t \right\|_{L_2} \leq \sup_{0 \leq t \leq T} \left( m_t^2 \mathbb{E} \left[ \|X_0\|^2 \right] + (1 - m_t^2)\sigma^2 d \right)^{1/2} \leq \left( \mathbb{E} \left[ \|X_0\|^2 \right] + \sigma^2 d \right)^{1/2},$$

*where $m_t = \exp(-\int_0^t \beta(s)\mathrm{d}s/2\sigma^2)$.*

*Proof.* Recall the following equality in law

$$\overrightarrow{X}_t = m_t X_0 + \sigma\sqrt{(1 - m_t^2)}G.$$

with $X_0 \sim \pi_{\mathrm{data}}$ and $G \sim \mathcal{N}(0, I_d)$.

Therefore, for any $t \in [0, T]$

$$\mathbb{E}\left[ \left\| \overleftarrow{X}_{T-t} \right\|^2 \right] = \mathbb{E}\left[ \left\| \overrightarrow{X}_t \right\|^2 \right] \leq m_t^2 \mathbb{E}\left[ \|X_0\|^2 \right] + \sigma^2 \left(1 - m_t^2\right) \mathbb{E}\left[ \|G\|^2 \right]$$

$$\leq m_t^2 \mathbb{E}\left[ \|X_0\|^2 \right] + \sigma^2 \left(1 - m_t^2\right) d.$$

$\square$

**Proposition C.6.** *Assume that H2 holds. For all $t_k \leq t \leq t_{k+1}$,*

$$\sup_{t_k \leq t \leq t_{k+1}} \left\| \overleftarrow{X}_t^\infty - \overleftarrow{X}_{t_k}^\infty \right\|_{L_2} \leq \left( \frac{1}{\sigma}\sqrt{2h\beta(T)} + m_T \int_{t_k}^{t_{k+1}} \left( \frac{1}{2\sigma^2} + 2\bar{L}_t \right) \bar{\beta}(t)\mathrm{d}t \right) B, \tag{41}$$

$$\sup_{0 \leq t \leq T} \left\| \overleftarrow{X}_t^\infty - \overleftarrow{X}_t \right\|_{L_2} \leq \left( \left[ \|X_0\|^2 \right] + \sigma^2 d \right)^{1/2} \exp\left( -\int_0^T \frac{\bar{\beta}(s)}{2\sigma^2}\mathrm{d}s \right), \tag{42}$$

*where $m_t = \exp(-\int_0^t \beta(s)\mathrm{d}s/2\sigma^2)$ and $B = (\mathbb{E}[\|X_0\|^2] + \sigma^2 d)^{1/2}$.*

*Proof.* Note that $\overrightarrow{X}_t$ has the same distribution as $m_t X_0 + \sigma\sqrt{(1 - m_t^2)}G$ where $G \sim \mathcal{N}(0, I_d)$ is independent of $X_0$. We have that $\overleftarrow{X}_0^\infty = G \sim \mathcal{N}(0, \sigma^2 I_d)$. Define $(\overleftarrow{X}_t)_{t \in [0,T]}$ satisfying (3) but initialized at

$$\overleftarrow{X}_0 = m_T Y + \sqrt{(1 - m_T^2)}G, \tag{43}$$

with $Y \sim \pi_{\mathrm{data}}$ independent of $G$ ($G$ being shared by $\overleftarrow{X}_0$ and $\overleftarrow{X}_0^\infty$).

On the one hand, following the same proof as in Proposition C.2, we have that

$$\left\|\overleftarrow{X}_t^\infty - \overleftarrow{X}_t\right\|_{L_2} \leq \left\|\overleftarrow{X}_0^\infty - \overleftarrow{X}_0\right\|_{L_2} \exp\left(-\int_0^t \frac{\bar{\beta}(s)}{2\sigma^2}\left(1 + 2\bar{C}_s\sigma^2\right)\mathrm{d}s\right)$$
$$\leq \left(\left[\|Y\|^2\right] + \sigma^2 d\right)^{1/2} m_T\,,$$

where we have used (43) as well as the fact that

$$\|X_0 - G\|_{L_2} = \left(\left[\|Y\|^2\right] + \left[\|G\|^2\right]\right)^{1/2} = B\,.$$

Therefore,

$$\sup_{0 \leq t \leq T}\left\|\overleftarrow{X}_t^\infty - \overleftarrow{X}_t\right\|_{L_2} \leq \left(\left[\|Y\|^2\right] + \sigma^2 d\right)^{1/2}\exp\left(-\int_0^T \frac{\bar{\beta}(s)}{2\sigma^2}\mathrm{d}s\right),$$

corresponding to (42).

On the other hand, we have that

$$\left\|\overleftarrow{X}_t^\infty - \overleftarrow{X}_{t_k}^\infty\right\|_{L_2} \leq \left\|\overleftarrow{X}_t - \overleftarrow{X}_{t_k}\right\|_{L_2} + \left\|\left(\overleftarrow{X}_t^\infty - \overleftarrow{X}_t\right) - \left(\overleftarrow{X}_{t_k}^\infty - \overleftarrow{X}_{t_k}\right)\right\|_{L_2}.$$

The process $(\overleftarrow{X}_t^\infty - \overleftarrow{X}_t)_{t\geq 0}$ is determined by the following ODE:

$$d\left(\overleftarrow{X}_t^\infty - \overleftarrow{X}_t\right) = \left(-\frac{\bar{\beta}(t)}{2\sigma^2}\left(\overleftarrow{X}_t^\infty - \overleftarrow{X}_t\right) + 2\bar{\beta}(t)\left(\nabla\log\tilde{p}_{T-t}\left(\overleftarrow{X}_t^\infty\right) - \nabla\log\tilde{p}_{T-t}\left(\overleftarrow{X}_t\right)\right)\right)\mathrm{d}t\,.$$

Then,

$$\left\|\left(\overleftarrow{X}_t^\infty - \overleftarrow{X}_t\right) - \left(\overleftarrow{X}_{t_k}^\infty - \overleftarrow{X}_{t_k}\right)\right\|_{L_2}$$
$$= \left\|\int_{t_k}^t \left(-\frac{\bar{\beta}(s)}{2\sigma^2}\left(\overleftarrow{X}_s^\infty - \overleftarrow{X}_s\right) + 2\bar{\beta}(s)\left(\nabla\log\tilde{p}_{T-s}\left(\overleftarrow{X}_s^\infty\right) - \nabla\log\tilde{p}_{T-s}\left(\overleftarrow{X}_s\right)\right)\right)\mathrm{d}s\right\|_{L_2}$$
$$\leq \sup_{t_k \leq t \leq t_{k+1}}\left\|\overleftarrow{X}_t^\infty - \overleftarrow{X}_t\right\|_{L_2}\int_{t_k}^{t_{k+1}}\left(\frac{1}{2\sigma^2} + 2\bar{L}_t\right)\bar{\beta}(t)\mathrm{d}t$$
$$\leq Bm_T\int_{t_k}^{t_{k+1}}\left(\frac{1}{2\sigma^2} + 2\bar{L}_t\right)\bar{\beta}(t)\mathrm{d}t\,.$$

Write $(\overrightarrow{X}_t)_{t\in[0,T]}$ the time reversal of $(\overleftarrow{X}_t)_{t\in[0,T]}$, which clearly satisfies (1). Using the following equality in law

$$\overrightarrow{X}_{T-t_k} = \frac{m_{T-t_k}}{m_{T-t}}\overrightarrow{X}_{T-t} + \left(1 - \left(\frac{m_{T-t_k}}{m_{T-t}}\right)^2\right)^{1/2}\sigma G\,,$$

with $G \sim \mathcal{N}(0, I_d)$, we get

$$\left\| \overleftarrow{X}_t - \overleftarrow{X}_{t_k} \right\|_{L_2} = \left\| \overrightarrow{X}_{T-t_k} - \overrightarrow{X}_{T-t} \right\|_{L_2} = \left( 1 - \left( \frac{m_{T-t_k}}{m_{T-t}} \right)^2 \right)^{1/2} \left( \left[ \left\| \overrightarrow{X}_{T-t} \right\|^2 \right] + \sigma^2 d \right)^{1/2}$$

$$\leq \left( 1 - \left( \frac{m_{T-t_k}}{m_{T-t}} \right)^2 \right)^{1/2} \sqrt{2}B \,,$$

where we have applied Proposition C.5 in the last inequality. Since

$$1 - \left( \frac{m_{T-t_k}}{m_{T-t}} \right)^2 = 1 - \exp \left( -\frac{1}{\sigma^2} \int_{T-t}^{T-t_k} \beta(s) \mathrm{d}s \right)$$

$$= \frac{1}{\sigma^2} \int_{T-t}^{T-t_k} \exp \left( -\frac{1}{\sigma^2} \int_{T-t}^{T-u} \beta(s) \mathrm{d}s \right) \beta(u) \mathrm{d}u$$

$$\leq \frac{1}{\sigma^2} h \beta(T) \,,$$

which concludes the proof of (41). $\qquad \square$

## D   Discussion on the hypotheses

**Proposition D.1.** *Assume that* $\log \pi_{\mathrm{data}}$ *is* $C_*$-*strongly concave and that* $C_* > 1/\sigma^2$. *Then, the modified score function* $\log \tilde{p}_t(x)$ *is, for any* $t \in (0, T]$, $C_t$-*strongly concave, with*

$$m_t = \exp \left( -\frac{1}{2\sigma^2} \int_0^t \beta(s) \mathrm{d}s \right) \,,$$

$$C_t = \frac{1}{m_t^2 / C_* + \sigma^2 (1 - m_t^2)} - \frac{1}{\sigma^2} \,.$$

*Moreover, we have that* $C_t \leq C_* - 1/\sigma^2$ *for any* $t \geq 0$.

*Proof.* This result is also proved in Saremi et al. (2023). We provide an alternative proof here for completeness. For all $1 \leq t \leq T$, $\overrightarrow{X}_t$ has the same law has $m_t X_0 + \sigma \sqrt{1 - m_t^2} Z$ where $X_0 \sim \pi_{\mathrm{data}}$ and $Z \sim \mathcal{N}(0, I_d)$ are independent. Therefore, writing $p_0 = \pi_{\mathrm{data}}$,

$$p_t(y) = \int_{\mathbb{R}^d} (2\pi \sigma^2 (1 - m_t^2))^{-d/2} \exp \left\{ \frac{-\|y - x_0 m_t\|^2}{2\sigma^2 (1 - m_t^2)} \right\} p_0(x_0) \mathrm{d}x_0 \,. \tag{44}$$

This implies that

$$\log p_t(y) = -\frac{d}{2} \log \left( 2\pi \sigma^2 (1 - m_t^2) \right) + \log \left( \int_{\mathbb{R}^d} \exp \left\{ -\frac{\|y - x_0 m_t\|^2}{2\sigma^2 (1 - m_t^2)} \right\} p_0(x_0) \mathrm{d}x_0 \right)$$

$$= -\frac{d}{2} \log \left( 2\pi \sigma^2 (1 - m_t^2) \right) + \log \left( \int_{\mathbb{R}^d} \exp \left\{ -\frac{\|y - u\|^2}{2\sigma^2 (1 - m_t^2)} \right\} p_0 \left( \frac{u}{m_t} \right) \mathrm{d}u \right)$$

$$+ \frac{d}{2\sigma^2} \int_0^t \beta(s) \mathrm{d}s \,.$$

Since $\log p_0$ is $C_*$-strongly concave, the function $x \mapsto p_0 \left(u/m_t\right)$ is $C_*/m_t^2$-strongly log-concave. Moreover, we have that the function $y \mapsto \exp\{-\|y\|^2/(2\sigma^2(1-m_t^2))\}$ is $(\sigma^2(1-m_t^2)^{-1}$-strongly log-concave. Applying Saumard and Wellner (Proposition 7.1 2014), since $p_t$ is a convolution of the previous two functions up to terms independent in space, we have that $\log p_t$ is $\left(m_t^2/C_* + \sigma^2 \left(1-m_t^2\right)\right)^{-1}$-strongly concave. Note that if $C_* \geq 1/\sigma^2$,

$$\frac{C_*}{m_t^2 + \sigma^2 C_* \left(1 - m_t^2\right)} \geq \frac{1}{\sigma^2}.$$

This entails that $\log \tilde{p}_t$ is $C_t$-strongly concave, with

$$C_t = \frac{1}{m_t^2/C_* + \sigma^2 \left(1 - m_t^2\right)} - \frac{1}{\sigma^2}.$$

Finally, finding the maximum $t \mapsto C_t$, is equivalent to find the maximum of the following function on $[0,1]$:

$$\psi : z \mapsto \frac{C_*}{z + \sigma^2 C_*(1-z)} - \frac{1}{\sigma^2}.$$

We have that $\psi(0) = C_* - 1/\sigma^2$, $\psi(1) = 0$ and for all $z \in [0,1]$,

$$\psi'(z) = \frac{\sigma^2 - 1/C_*}{\left(z/C_* + \sigma^2(1-z)\right)^2},$$

which is negative since $C_* \geq 1 \leq 1/\sigma^2$. Therefore, we get $0 \leq C_t \leq C_* - 1/\sigma^2$. $\qquad \square$

**Proposition D.2.** *If $\log \pi_{\mathrm{data}}$ is $L_*$-smooth, then for all $0 \leq t \leq T$, $\nabla \log \tilde{p}_t$ is $L_t$-Lipschitz in the space variable with*

$$L_t = \min\left\{\frac{1}{\sigma^2 \left(1 - m_t^2\right)}; \frac{L_*}{m_t^2}\right\} + \frac{1}{\sigma^2}.$$

*Moreover, if $L_* > 1/\sigma^2$, we can choose $L_t$ as follows:*

$$L_t = \min\left\{\frac{1}{\sigma^2 \left(1 - m_t^2\right)}; \frac{L_*}{m_t^2}\right\} - \frac{1}{\sigma^2}.$$

*Moreover, in this case, we have that $L_t \leq L_*$ for any $t \geq 0$.*

*Proof.* In the proof of Proposition D.1, we proved that, if $\log \pi_{\mathrm{data}}$ is $C_*$-strongly concave, $\log p_t$ is $\left(m_t^2/C_* + \sigma^2 \left(1-m_t^2\right)\right)^{-1}$-strongly concave i.e.,

$$\nabla^2 \left(-\log p_t\right)(x) \succcurlyeq \frac{1}{m_t^2/C_* + \sigma^2 \left(1 - m_t^2\right)} I_d.$$

For $p_0 := \pi_{\mathrm{data}}$, we have that $p_t$ is given by (44). This means that $p_t$ is the density of the sum of two independent random variables $X_1 + X_0$ of density respectively $q_0$ and $q_1$, such that

$$q_0(x) := \frac{1}{m_t^d} p_0 \left(\frac{u}{m_t^d}\right) = e^{-\phi_0(x)},$$

$$q_1(x) := \frac{1}{\left(2\pi\sigma^2 \left(1-m_t^2\right)\right)^{d/2}} \exp\left\{-\frac{\|y\|^2}{2\sigma^2 \left(1-m_t^2\right)}\right\} = e^{-\phi_1(x)},$$

for two functions $\phi_0$ and $\phi_1$. Therefore, as in the proof of Saumard and Wellner (Proposition 7.1 2014), we get

$$\nabla^2 \left( -\log p_t \right) (x) = -\mathrm{Var}(\nabla \phi_0(X_0)|X_0 + X_1 = x) + \mathbb{E}[\nabla^2 \phi_0(X_0)|X_0 + X_1 = x]$$
$$= -\mathrm{Var}(\nabla \phi_1(X_1)|X_0 + X_1 = x) + \mathbb{E}[\nabla^2 \phi_1(X_1)|X_0 + X_1 = x].$$

Since $\nabla \log p_0$ is $L_*$-Lipschitz and from the definition of $q_1$,

$$\nabla^2 \phi_0 \preccurlyeq \frac{L_*}{m_t^2} I_d, \quad \nabla^2 \phi_1 \preccurlyeq \frac{1}{\sigma^2 \left( 1 - m_t^2 \right)} I_d.$$

Hence,

$$\nabla^2 \left( -\log p_t \right) (x) \preccurlyeq \min \left\{ \frac{1}{\sigma^2 \left( 1 - m_t^2 \right)}; \frac{L_0}{m_t^2} \right\} I_d.$$

Therefore, since the difference between $\nabla \log p_t$ and $\nabla \log \tilde{p}_t$ is a linear function, we can choose $L_t$ as follows:

$$L_t = \min \left\{ \frac{1}{\sigma^2 \left( 1 - m_t^2 \right)}; \frac{L_*}{m_t^2} \right\} + \frac{1}{\sigma^2}.$$

Clearly we have that $0 \le m_t^2 \le 1$, therefore $1/m_t^2 \ge 1$ and $1/\left( 1 - m_t^2 \right) \ge 1$. This means that, if $L_* \ge 1/\sigma^2$,

$$\min \left\{ \frac{1}{\sigma^2 \left( 1 - m_t^2 \right)}; \frac{L_*}{m_t^2} \right\} \ge \frac{1}{\sigma^2}.$$

Thus, we can choose $L_t$ to be

$$L_t = \min \left\{ \frac{1}{\sigma^2 \left( 1 - m_t^2 \right)}; \frac{L_*}{m_t^2} \right\} - \frac{1}{\sigma^2}.$$

Finally, since $m_0 = 1$, we have that $L_0 = L_* - 1/\sigma^2$. This function increases up to the point where $L_*/m_t^2 = (\sigma^2(1 - m_t^2)^{-1}$, achieved for $m_{t^*}^2 = (\sigma^2 L_*)/(\sigma^2 L_* + 1)$. At this point, we have that $L_{t^*} = L_*$. After this point the Lipschitz constant decreases to 0, as $m_t \to 0$ for $t \to \infty$. This means that for any $t$, $L_t$ is bounded by $L_*$. $\qquad\square$

**Proposition D.3.** *Assume that $\log \pi_{\mathrm{data}}$ is $L_*$-smooth and $C_*$-strongly concave. Consider the regular discretization $\{t_k, 0 \le k \le N\}$ of $[0, T]$ of constant step size $h$. By choosing $h > 0$ such that for all $t_k$ with $0 \le k \le N - 1$,*

$$h \le \min \left\{ \frac{\log(2)2\sigma^2}{\beta(T)}; \frac{\sigma^2 C_* - 1}{\sigma^2 C_* \left( \sigma^2 L_* + 1 \right) L_* \beta(T)}; \frac{\sigma^2 C_* - 1}{\left( \sigma^2 L_* - 1 \right) L_* \beta(T)} \right\}, \tag{45}$$

*then, for all $0 \le k \le N - 1$,*

$$0 < 1 + \frac{\widetilde{m}_{t_k}^2}{\widetilde{m}_{t_{k+1}}^2} \left( \int_{t_k}^{t_{k+1}} \bar{L}_t \frac{\widetilde{m}_t}{\widetilde{m}_{t_k}} \bar{\beta}(t) \mathrm{d}t \right)^2 - 2 \frac{\widetilde{m}_{t_k}}{\widetilde{m}_{t_{k+1}}} \int_{t_k}^{t_{k+1}} \bar{C}_t \frac{\widetilde{m}_t}{\widetilde{m}_{t_k}} \bar{\beta}(t) \mathrm{d}t < 1.$$

*In addition, if*

$$h \leq \min \left\{ \frac{\log(2)2\sigma^2}{\beta(T)}; \frac{\sigma^2 C_* - 1}{\sigma^2 M + \beta(T)L_* \left(\sigma^2 L_* - 1\right)}; \right.$$
$$\left. \frac{\sigma^2 C_* - 1}{\sigma^2 C_*} \frac{m_T^2 \left(1 - m_T^2\right)}{\sigma^2 M \left(1 - m_T^2\right) + \beta(T)L_* m_T^2}; \frac{\left(\sigma^2 C_* - 1\right) L_*}{\sigma^2 C_* \left(\sigma^2 L_* + 1\right) \left(M + \beta(T)L_*^2\right)} \right\} , \quad (46)$$

*then, for all $0 \leq k \leq N - 1$,*

$$0 < 1 + \frac{1}{2} \frac{\widetilde{m}_{t_k}^2}{\widetilde{m}_{t_{k+1}}^2} \left( \int_{t_k}^{t_{k+1}} \bar{L}_t \frac{\widetilde{m}_t}{\widetilde{m}_{t_k}} \bar{\beta}(t)\mathrm{d}t \right)^2 - \frac{\widetilde{m}_{t_k}}{\widetilde{m}_{t_{k+1}}} \int_{t_k}^{t_{k+1}} \bar{C}_t \frac{\widetilde{m}_t}{\widetilde{m}_{t_k}} \bar{\beta}(t)\mathrm{d}t$$
$$+ \frac{\widetilde{m}_{t_k}^2}{\widetilde{m}_{t_{k+1}}^2} Mh \int_{t_k}^{t_{k+1}} \frac{\widetilde{m}_t}{\widetilde{m}_{t_k}} \bar{\beta}(t)\mathrm{d}t < 1 .$$

*Proof.* Define $\epsilon_1$ and $\epsilon_2$ as in (39)-(40). From Proposition C.4, we have that $\epsilon_i \in (0, 1)$, for $i = 1, 2$, if we have (37)-(38).

First, we prove that (45) implies (37). From Proposition D.2, we have that $L_t$ is bounded by $L_*$ everywhere. Moreover, since $\widetilde{m}_{t_{k+1}}/\widetilde{m}_{t_k} = \exp\left(-\int_{t_k}^{t_{k+1}} \bar{\beta}(s)/2\sigma^2 \mathrm{d}s\right)$, we can find $h$ small enough such that $2\widetilde{m}_{t_k}/\widetilde{m}_{t_{k+1}} \geq 1$. This is equivalent to $\int_{t_k}^{t_{k+1}} \bar{\beta}(s)/2\sigma^2 \mathrm{d}s \leq \log(2)$ and it is implied by

$$h \leq \frac{\log(2)2\sigma^2}{\beta(T)} .$$

Now, we study the function $t \mapsto C_t/L_t$. From the proof of the Proposition D.1, we have that

$$C_t = \frac{1}{m_t^2/C_* + \sigma^2 \left(1 + m_t^2\right)} - \frac{1}{\sigma^2} ,$$

which is a decreasing function. Moreover, from the proof of the Proposition D.2, we have that

$$L_t = \min \left\{ \frac{1}{\sigma^2 \left(1 - m_t^2\right)}; \frac{L_*}{m_t^2} \right\} - \frac{1}{\sigma^2} ,$$

which is an increasing function from 0 up to $t^*$, such that $m_{t^*}^2 = \frac{\sigma^2 L_*}{\sigma^2 L_* + 1}$ and decreasing for $t \geq t^*$. On the one hand, this means that for $t \in [0, t^*]$, the function $t \mapsto C_t/L_t$ is decreasing, therefore reaching its minimum $\left(\sigma^2 C_* - 1\right)/\left(\sigma^2 L_* - 1\right)$ in 0, which is a positive quantity. On the other hand, for $t \geq t^*$, we have that

$$\frac{C_t}{L_t} = \frac{1}{m_t^2/C_* + \sigma^2 \left(1 + m_t^2\right)} \frac{1}{\sigma^2} \frac{\left(\sigma^2 C_* - 1\right) m_t^2}{C_*} \frac{\sigma^2 \left(1 - m_t^2\right)}{m_t^2}$$
$$\geq \frac{1}{\sigma^2} \frac{\sigma^2 C_* - 1}{C_*} \left(1 - m_t^2\right)$$
$$\geq \frac{1}{\sigma^2} \frac{\sigma^2 C_* - 1}{C_*} \left(1 - m_{t^*}^2\right) = \frac{\sigma^2 C_* - 1}{\sigma^2 C_* \left(\sigma^2 L_* + 1\right)} .$$

Therefore, combining the previous inequalities, we have that condition (45) implies (37).

Secondly, we prove (46) implies (38). Take $h$ to satisfy

$$h \leq \frac{\log(2)2\sigma^2}{\beta(T)} \, .$$

We now need to study the function $t \mapsto \frac{C_t}{M+\beta(T)L_*\bar{L}_t}$. On the one hand, this function is decreasing for $t \in [0, t^*]$, therefore reaching its minimum $\frac{\sigma^2 C_* - 1}{\sigma^2 M + \beta(T)L_*(\sigma^2 L_* - 1)}$ in 0, which is a positive quantity. On the other hand, for $t \geq t^*$, we have that

$$\frac{C_t}{M+\beta(T)L_*L_t} = \frac{1}{m_t^2/C_* + \sigma^2(1+m_t^2)} \frac{1}{\sigma^2} \frac{(\sigma^2 C_* - 1)m_t^2}{C_*} \frac{\sigma^2(1-m_t^2)}{\sigma^2 M(1-m_t^2) + \beta(T)L_* m_t^2}$$

$$\geq \frac{1}{\sigma^2} \frac{\sigma^2 C_* - 1}{C_*} \frac{m_t^2(1-m_t^2)}{\sigma^2 M(1-m_t^2) + \beta(T)L_* m_t^2} \, .$$

Controlling from below the previous quantity, boils down to control from below the function $\psi(y) = \frac{y(1-y)}{\sigma^2 M(1-y)+\beta(T)L_* y}$ for $y \in [m_{t^*}^2, m_T^2]$. We see that $\psi$ in this interval can be bounded by $\min\{\psi(m_{t^*}^2), \psi(m_T^2)\}$. Therefore, we get

$$\frac{C_t}{M+\beta(T)L_*L_t} \geq \frac{\sigma^2 C_* - 1}{\sigma^2 C_*} \min \left\{ \frac{m_T^2(1-m_T^2)}{\sigma^2 M(1-m_T^2)+\beta(T)L_* m_T^2}; \frac{L_*}{(\sigma^2 L_0 + 1)(M+\beta(T)L_*^2)} \right\} \, .$$

Therefore, combining the previous inequalities, we have that condition (46) implies (38).

$\square$

# E   Details on numerical experiments

This section is divided into two parts. The first part is dedicated to providing detailed implementation choices for the numerical experiments presented in Section 5. The second part displays additional experiments and more details about the experiments of Section 5. All experiments were conducted on a local computer CPU equipped with an Apple M3 processor (8GB of unified memory). This setup is sufficient to replicate the experiments of this paper.

## E.1   Implementation choices

### E.1.1   Exact score and metrics in the Gaussian case

**Lemma E.1.** *Assume that the forward process defined in (1) :*

$$\mathrm{d}\overrightarrow{X}_t = -\frac{\beta(t)}{2\sigma^2}\overrightarrow{X}_t \mathrm{d}t + \sqrt{\beta(t)}\mathrm{d}B_t, \quad \overrightarrow{X}_0 \sim \pi_0,$$

*is initialised with $\pi_0$ the Gaussian probability density function with mean $\mu_0$ and variance $\Sigma_0$. Then, the score function of (1) is:*

$$\nabla \log p_t : x \mapsto -(m_t^2 \Sigma_0 + \sigma_t^2 \mathrm{I}_d)^{-1}(x - m_t \mu_0),$$

*where $p_t$ is the probability density function of $\overrightarrow{X}_t$, $m_t = \exp\{-\int_0^t \beta(s)\mathrm{d}s/(2\sigma^2)\}$ and $\sigma_t^2 = \sigma^2(1-m_t^2)$.*

*Proof.* Note that $\overrightarrow{X}_t$ has the same law as $m_t X_0 + \sigma_t Z$ where $Z \sim \mathcal{N}(0, \mathrm{I}_d)$ is independent of $X_0$. Therefore $\overrightarrow{X}_t \sim \mathcal{N}(m_t \mu_0, \overrightarrow{\Sigma_t})$ with $\overrightarrow{\Sigma_t} = m_t^2 \Sigma_0 + \sigma_t^2 \mathrm{I}_d$ which concludes the proof.

$\square$

**Lemma E.2.** *Let* $\mu_1, \mu_2$ *in* $\mathbb{R}^d$ *and* $\Sigma_1$ *and* $\Sigma_2$ *be two definite positive matrices in* $\mathbb{R}^{d \times d}$. *Then,*

$$\mathrm{KL}(\varphi_{\mu_1, \Sigma_1} \| \varphi_{\mu_2, \Sigma_2}) = \frac{1}{2} \left( \log \frac{|\Sigma_2|}{|\Sigma_1|} - d + \mathrm{Tr}\left(\Sigma_2^{-1} \Sigma_1\right) + (\mu_2 - \mu_1)^\top \Sigma_2^{-1} (\mu_2 - \mu_1) \right). \quad (47)$$

**Lemma E.3.** *Let* $\mu_1, \mu_2$ *in* $\mathbb{R}^d$ *and* $\Sigma_1$ *and* $\Sigma_2$ *be two definite positive matrices in* $\mathbb{R}^{d \times d}$. *Then,*

$$\mathcal{W}_2^2(\varphi_{\mu_1, \Sigma_1}, \varphi_{\mu_2, \Sigma_2}) = \|\mu_2 - \mu_1\|^2 + \mathrm{Tr}\left(\Sigma_1 + \Sigma_2 - 2\left(\Sigma_2^{1/2} \Sigma_2 \Sigma_1^{1/2}\right)^{1/2}\right). \quad (48)$$

**Lemma E.4.** *The relative Fisher information between* $X_0 \sim \mathcal{N}(\mu_0, \Sigma_0)$ *and* $X_\infty \sim \mathcal{N}(0, \sigma^2 \mathrm{I}_d)$ *is given by:*

$$\mathcal{I}\left(\varphi_{\mu_0, \Sigma_0} \| \varphi_{\sigma^2}\right) = \frac{1}{\sigma^4}\left(\mathrm{Tr}\left(\Sigma_0\right) + \|\mu_0\|^2\right) - \frac{2d}{\sigma^2} + \mathrm{Tr}\left(\Sigma_0^{-1}\right).$$

*Proof.* The relative Fisher information between $X_0$ and $X_\infty$ is given by

$$\mathcal{I}\left(\varphi_{\mu_0, \Sigma_0} \| \varphi_{\sigma^2}\right) = \int \left\| \nabla \log \left( \frac{\varphi_{\mu_0, \Sigma_0}(x)}{\varphi_{\sigma^2}(x)} \right) \right\|^2 \varphi_{\mu_0, \Sigma_0}(x) \mathrm{d}x.$$

Write

$$\nabla \log \frac{\varphi_{\mu_0, \Sigma_0}(x)}{\varphi_{\sigma^2}(x)} = \frac{x}{\sigma^2} - \Sigma_0^{-1}(x - \mu_0),$$

so that,

$$\left\| \nabla \log \frac{\varphi_{\mu_0, \Sigma_0}(x)}{\varphi_{\sigma^2}(x)} \right\|^2 = \left\| \frac{x}{\sigma^2} - \Sigma_0^{-1}(x - \mu_0) \right\|^2$$

$$= \left( \frac{x}{\sigma^2} - \Sigma_0^{-1}(x - \mu_0) \right)^\top \left( \frac{x}{\sigma^2} - \Sigma_0^{-1}(x - \mu_0) \right)$$

$$= \frac{\|x\|^2}{\sigma^4} - \frac{2}{\sigma^2} x^\top \Sigma_0^{-1}(x - \mu_0) + (x - \mu_0)^\top \Sigma_0^{-2}(x - \mu_0).$$

First,

$$\mathbb{E}\left[ \frac{\|X_0\|^2}{\sigma^4} \right] = \frac{1}{\sigma^4}\left(\mathrm{Tr}\left(\Sigma_0\right) + \|\mu_0\|^2\right).$$

Then,

$$\mathbb{E}\left[ \frac{2}{\sigma^2} X_0^T \Sigma_0^{-1}(X_0 - \mu_0) \right] = \frac{2}{\sigma^2}\left(\mathrm{Tr}\left(\Sigma_0^{-1} \mathbb{E}\left[X_0 X_0^\top\right]\right) - \mu_0^\top \Sigma_0^{-1} \mu_0\right).$$

Using that $\mathbb{E}\left[X_0 X_0^\top\right] = \Sigma_0 + \mu_0 \mu_0^\top$ yields

$$
\begin{aligned}
\mathbb{E}\left[\frac{2}{\sigma^2} X_0^\top \Sigma_0^{-1}(X_0 - \mu_0)\right] &= \frac{2}{\sigma^2}\left(\text{Tr}\left(\Sigma_0^{-1}\left(\Sigma_0 + \mu_0 \mu_0^\top\right)\right) - \mu_0^\top \Sigma_0^{-1}\mu_0\right) \\
&= \frac{2}{\sigma^2}\left(d + \text{Tr}\left(\Sigma_0^{-1}\mu_0 \mu_0^\top\right) - \mu_0^\top \Sigma_0^{-1}\mu_0\right) \\
&= \frac{2d}{\sigma^2}.
\end{aligned}
$$

Finally,

$$
\begin{aligned}
\mathbb{E}\left[(X_0 - \mu_0)^\top \Sigma_0^{-2}(X_0 - \mu_0)\right] &= \mathbb{E}\left[\text{Tr}\left((X_0 - \mu_0)^\top \Sigma_0^{-2}(X_0 - \mu_0)\right)\right] \\
&= \mathbb{E}\left[\text{Tr}\left(\Sigma_0^{-2}(X_0 - \mu_0)(X_0 - \mu_0)^\top\right)\right] \\
&= \text{Tr}\left(\Sigma_0^{-2}\mathbb{E}\left[(X_0 - \mu_0)(X_0 - \mu_0)^\top\right]\right) \\
&= \text{Tr}\left(\Sigma_0^{-2}\Sigma_0\right) \\
&= \text{Tr}\left(\Sigma_0^{-1}\right),
\end{aligned}
$$

which concludes the proof. $\qquad\square$

**Proposition E.5.** *Under the same assumptions as in Lemma E.1, the Euclidean norm of the score function admits the following upper bound for $t_1 \leq t_2$:*

$$
\sup_{t_1 \leq t \leq t_2} \|\nabla \log p_{t_1}(x) - \nabla \log p_t(x)\| \leq (t_2 - t_1)\max\left\{\|\mu_0\|\,\kappa_2; \kappa_1\right\}(1 + \|x\|),
$$

*with*

$$
\kappa_1 := \frac{m_{t_1}^2 \frac{\beta(t_2)}{\sigma^2}\left|\lambda_{\min} - \sigma^2\right|}{\left|\left(\sigma^2 + m_{t_1}^2\left(\lambda_{\min} - \sigma^2\right)\right)\left(\sigma^2 + m_{t_2}^2\left(\lambda_{\min} - \sigma^2\right)\right)\right|},
$$

*and*

$$
\kappa_2 := \frac{m_{t_1}\frac{\beta(t_2)}{2\sigma^2}\left|m_{t_1} m_{t_2}\left(\lambda_{\min} - \sigma^2\right) - \sigma^2\right|}{\left|\left(\sigma^2 + m_{t_1}^2\left(\lambda_{\min} - \sigma^2\right)\right)\left(\sigma^2 + m_{t_2}^2\left(\lambda_{\min} - \sigma^2\right)\right)\right|},
$$

*where $\lambda_{\min}$ is the smallest eigenvalue of $\Sigma_0$.*

*Proof.* Let $t_1 \leq t_2$,

$$
\begin{aligned}
\|\nabla \log p_{t_1}(x) - \nabla \log p_{t_2}(x)\| &= \left\|-(m_{t_1}^2 \Sigma_0 + \sigma_{t_1}^2 \mathrm{I}_d)^{-1}(x - m_{t_1}\mu_0) + (m_{t_2}^2 \Sigma_0 + \sigma_{t_2}^2 \mathrm{I}_d)^{-1}(x - m_{t_2}\mu_0)\right\| \\
&\leq \left\|\left(m_{t_1}\left(m_{t_1}^2 \Sigma_0 + \sigma_{t_1}^2 \mathrm{I}_d\right)^{-1} - m_{t_2}\left(m_{t_2}^2 \Sigma_0 + \sigma_{t_2}^2 \mathrm{I}_d\right)^{-1}\right)\mu_0\right\| \\
&\quad + \left\|\left(\left(m_{t_1}^2 \Sigma_0 + \sigma_{t_1}^2 \mathrm{I}_d\right)^{-1} - \left(m_{t_2}^2 \Sigma_0 + \sigma_{t_2}^2 \mathrm{I}_d\right)^{-1}\right)x\right\|.
\end{aligned}
$$

Writing $M_t = \left(m_t^2 \Sigma_0 + \sigma_t^2 \mathrm{I}_d\right)^{-1}$ we have, for $t_1 \leq t_2$,

$$
\begin{aligned}
\|M_{t_1} - M_{t_2}\| &\leq \left| \frac{1}{m_{t_1}^2 \lambda_{\min} + \sigma_{t_1}^2} - \frac{1}{m_{t_2}^2 \lambda_{\min} + \sigma_{t_2}^2} \right| \\
&\leq \left| \frac{\left(m_{t_2}^2 - m_{t_1}^2\right)\left(\lambda_{\min} - \sigma^2\right)}{\left(\sigma^2 + m_{t_1}^2 \left(\lambda_{\min} - \sigma^2\right)\right)\left(\sigma^2 + m_{t_2}^2 \left(\lambda_{\min} - \sigma^2\right)\right)} \right| \\
&\leq (t_2 - t_1) \underbrace{\frac{m_{t_1}^2 \frac{\beta(t_2)}{\sigma^2} \left|\lambda_{\min} - \sigma^2\right|}{\left|\left(\sigma^2 + m_{t_1}^2 \left(\lambda_{\min} - \sigma^2\right)\right)\left(\sigma^2 + m_{t_2}^2 \left(\lambda_{\min} - \sigma^2\right)\right)\right|}}_{\kappa_1}.
\end{aligned}
$$

Moreover, for $t_1 \leq t_2$,

$$
\begin{aligned}
\|m_{t_1} M_{t_1} - m_{t_2} M_{t_2}\| &\leq \left| \frac{m_{t_1}}{m_{t_1}^2 \lambda_{\min} + \sigma_{t_1}^2} - \frac{m_{t_2}}{m_{t_2}^2 \lambda_{\min} + \sigma_{t_2}^2} \right| \\
&\leq \left| \frac{\left(m_{t_1} m_{t_2}^2 - m_{t_2} m_{t_1}^2\right)\left(\lambda_{\min} - \sigma^2\right) + \sigma^2\left(m_{t_1} - m_{t_2}\right)}{\left(\sigma^2 + m_{t_1}^2 \left(\lambda_{\min} - \sigma^2\right)\right)\left(\sigma^2 + m_{t_2}^2 \left(\lambda_{\min} - \sigma^2\right)\right)} \right| \\
&\leq \frac{\left|m_{t_2} - m_{t_1}\right| \left|m_{t_1} m_{t_2}\left(\lambda_{\min} - \sigma_2\right) - \sigma^2\right|}{\left|\left(\sigma^2 + m_{t_1}^2 \left(\lambda_{\min} - \sigma^2\right)\right)\left(\sigma^2 + m_{t_2}^2 \left(\lambda_{\min} - \sigma^2\right)\right)\right|} \\
&\leq (t_2 - t_1) \underbrace{\frac{m_{t_1} \frac{\beta(t_2)}{2\sigma^2} \left|m_{t_1} m_{t_2}\left(\lambda_{\min} - \sigma^2\right) - \sigma^2\right|}{\left|\left(\sigma^2 + m_{t_1}^2 \left(\lambda_{\min} - \sigma^2\right)\right)\left(\sigma^2 + m_{t_2}^2 \left(\lambda_{\min} - \sigma^2\right)\right)\right|}}_{\kappa_2}.
\end{aligned}
$$

Finally,

$$
\begin{aligned}
\|\nabla \log p_{t_1}(x) - \nabla \log p_{t_2}(x)\| &\leq (t_2 - t_1) \|\mu_0\| \kappa_2 + (t_2 - t_1) \kappa_1 \|x\| \\
&\leq (t_2 - t_1) \max\left\{\|\mu_0\| \kappa_2; \kappa_1\right\}\left(1 + \|x\|\right).
\end{aligned}
$$

$\square$

### E.1.2 Stochastic differential equation exact simulation

In certain cases, exact simulation of stochastic differential equations is possible. In particular, due to the linear nature of the drift the forward process (1) can be simulated exactly. Indeed, the marginal distribution of (1) at time $t$ writes as

$$
\overrightarrow{X}_t = m_t X_0 + \sigma_t Z,
$$

with $Z \sim \mathcal{N}\left(0, \mathrm{I}_d\right)$ independent of $X_0$, $X_0 \sim \pi_0$, $m_t = \exp\{-\int_0^t \beta(s)\mathrm{d}s/(2\sigma^2)\}$ and $\sigma_t^2 = \sigma^2(1 - \exp\{-\int_0^t \beta(s)/\sigma^2 \mathrm{d}s\})$. Therefore, sampling from the forward process only necessitates access to samples from $\pi_0$ and $\mathcal{N}(0, \mathrm{I}_d)$.

### E.1.3 Noise schedules

**Linear and parametric noise schedules.**   In Section 5, we introduced parametric noise schedules of the form

$$\beta_a(t) \propto (e^{at} - 1)/(e^{aT} - 1) \, ,$$

with $a \in \mathbb{R}$ ranging from $-10$ to $10$ (see Figure 6). For all $a$, with a time horizon of $T = 1$, the initial and final values have been set to match exactly the schedule prescribed by Song et al. (2021) (i.e. $\beta_a(0) = 0.1$ and $\beta_a(1) = 20$) when $a = 0$ (linear schedule).

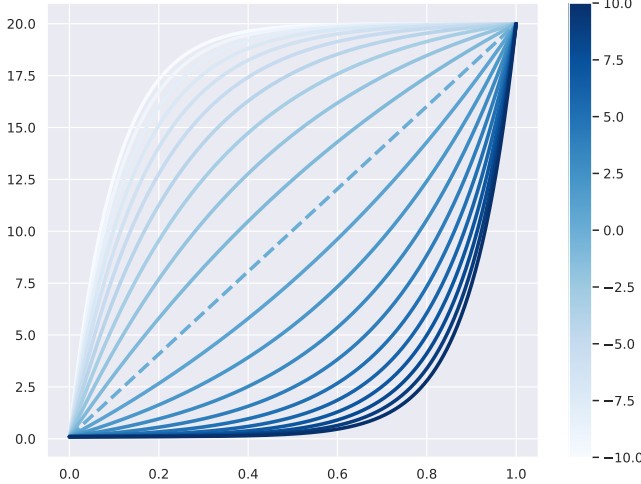

Figure 6: Evolution of noise schedules $\beta_a$ w.r.t. time, for different values of parameter between $-10$ to $10$. The linear case $a = 0$ (Song and Ermon, 2019; Song et al., 2021) is dashed.

As shown in Section E.1.2 $m_t$ and $\sigma_t$ are the two quantities of interest in the calibration of the noising procedure of the forward proces. Their values for different choices of $a$ are displayed in Figure 7.

**Cosine noise schedule.**   We consider the cosine schedule introduced in Nichol and Dhariwal (2021) for which the forward process is defined for $t \in \{1, ..., T\}$ as

$$X_t := \sqrt{\bar{\alpha}_t} X_0 + \sqrt{1 - \bar{\alpha}_t} Z \, ,$$

with $X_0 \sim \pi_{\text{data}}$, $Z \sim \mathcal{N}(0, I_d)$ and with

$$\bar{\alpha}_t = \frac{f(t)}{f(0)}; \ f(t) = \cos\left(\frac{t/T + s}{1 + s}\frac{\pi}{2}\right)^2 \, .$$

To use this noise schedule in the SDE setting we notice that the forward process writes, for $t \in [0, T]$,

$$\overrightarrow{X}_t = m_t X_0 + \sigma_t Z \, ,$$

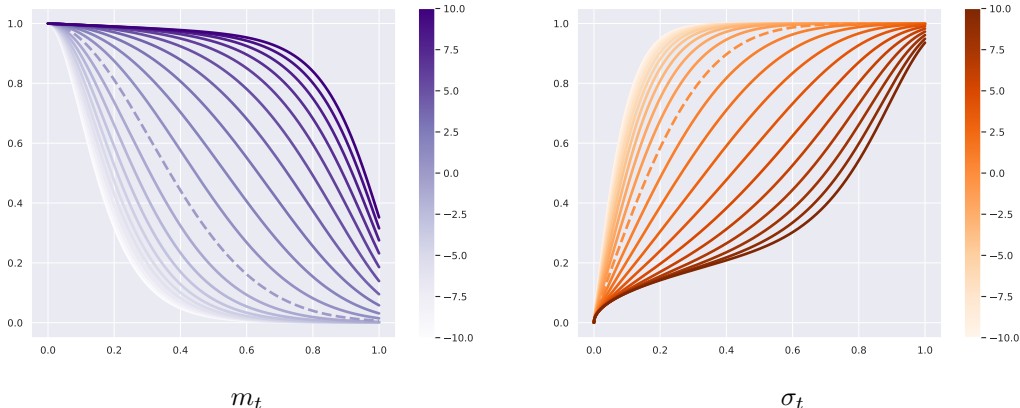

$$m_t \qquad \qquad \qquad \qquad \sigma_t$$

Figure 7: Evolution of $m_t$ and $\sigma_t$ over time, for different choices of $a$ in the noise schedule $\beta_a$ used in see Section 5. The stationary distribution of the forward process $\sigma^2$ is set to 1. The range for $a$ spans from -10 to 10, with the dashed line representing the linear schedule as proposed originally in the VPSDE models (Song et al., 2021).

with $m_t = \exp\{-\int_0^t \beta(s)\mathrm{d}s/(2\sigma^2)\}$, $\sigma_t^2 = \sigma^2(1 - m_t^2)$ and $Z \sim \mathcal{N}(0, I_d)$. Therefore, we can simply identify $\beta_{\cos}$ by solving

$$-\int_0^t \frac{\beta_{cos}(s)}{2\sigma^2}\mathrm{d}s = \log(\bar{\alpha}_t) \,,$$

which yields the following noising function:

$$\beta_{\cos}(t) := \sigma^2 \frac{\pi}{T(s+1)} \tan\left(\frac{\pi(s+t/T)}{2(s+1)}\right) \,. \tag{49}$$

Finally, to ensure fair comparison with the linear schedule and the parametric schedules defined in Section 5, we set in all our experiments $s = 0.021122$ so that $\beta_{\cos}(0) \approx \beta_a(0) = 0.1$ for any $a$.

### E.1.4 Discretization details of the diffusion SDE

In contrast to the forward process, described in Equation (1), which is simulated exactly, the backward process needs to be discretized. Recall that the backward process of (1) is given by:

$$\mathrm{d}\overleftarrow{X}_t = -\frac{\bar{\beta}(t)}{2\sigma^2}\overleftarrow{X}_t + \bar{\beta}(t)\nabla\log p_{T-t}\left(\overleftarrow{X}_t\right)\mathrm{d}t + \sqrt{\bar{\beta}(t)}\mathrm{d}B_t, \quad \overleftarrow{X}_0 \sim \pi_\infty.$$

Consider time intervals $0 \le t_k \le t \le t_{k+1} \le T$, with $t_k = \sum_{\ell=1}^k \gamma_\ell$ and $T = \sum_{k=1}^N \gamma_k$.

In our theoretical analysis, we have considered the Exponential Integrator discretization, defined recursively for $t \in [t_k, t_{k+1}]$ by

$$\mathrm{d}\overleftarrow{X}_t^{EI} = \bar{\beta}(t)\left(-\frac{1}{2\sigma^2}\overleftarrow{X}_t^{EI} + \nabla\log p_{T-t_k}\left(T - t_k, \overleftarrow{X}_{t_k}^{EI}\right)\right)\mathrm{d}t + \sqrt{\bar{\beta}(t)}\mathrm{d}B_t, \quad \overleftarrow{X}_0^{EI} \sim \pi_\infty \,.$$

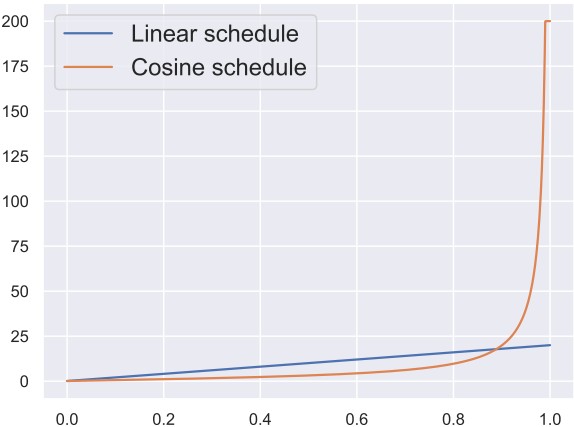

Figure 8: Evolution of noising functions under the cosine schedule (orange, $\beta_{\cos}$) compared to the linear schedule ($\beta_0$, blue) over time with $\sigma^2 = 1$ and $s = 0.021122$. Additionally, since $\beta_0$ increases unboundedly near $T$, we clip its value to 200 for better visualization.

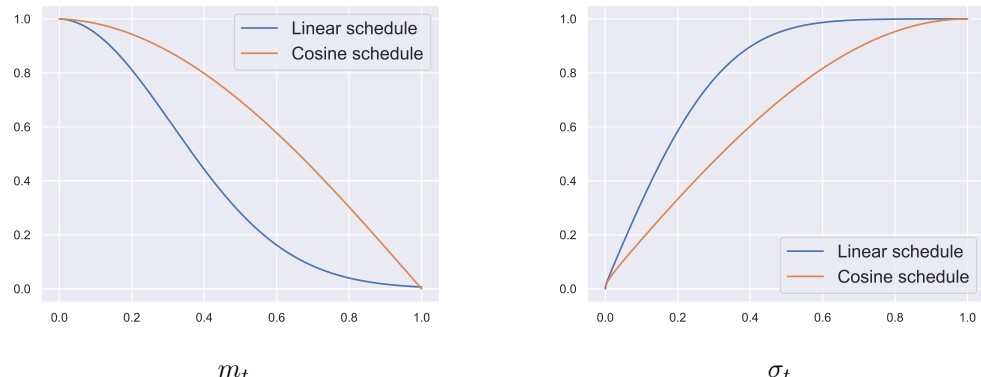

Figure 9: Evolution of $m_t$ and $\sigma_t$ for both the cosine schedule (orange) and the linear schedule (blue) w.r.t. time, with $s = 0.021122$ and $\sigma^2 = 1$. We clip the value of $\beta_{\cos}$ by 200 for better visualization.

In the numerical experiments, we have given priority to the Euler-Maruyama discretization, which is widely used, and defined recursively for $t \in [t_k, t_{k+1}]$ by

$$\mathrm{d}\overleftarrow{X}_t^{EM} = -\frac{\bar{\beta}(t_k)}{2\sigma^2}\overleftarrow{X}_{t_k}^{EM} + \bar{\beta}(t_k)\nabla\log p_{T-t_k}\left(\overleftarrow{X}_{t_k}^{EM}\right)\mathrm{d}t + \sqrt{\bar{\beta}(t_k)}\mathrm{d}B_t, \quad \overleftarrow{X}_0^{EM} \sim \pi_\infty. \tag{50}$$

To ensure transparency, the graphs presented in Figure 2 of Section 5.1 are reproduced in Figure 10 using an Exponential Integrator discretization scheme. As expected for fine discretization steps (here 500 steps were used) the two schemes produce nearly identical results.

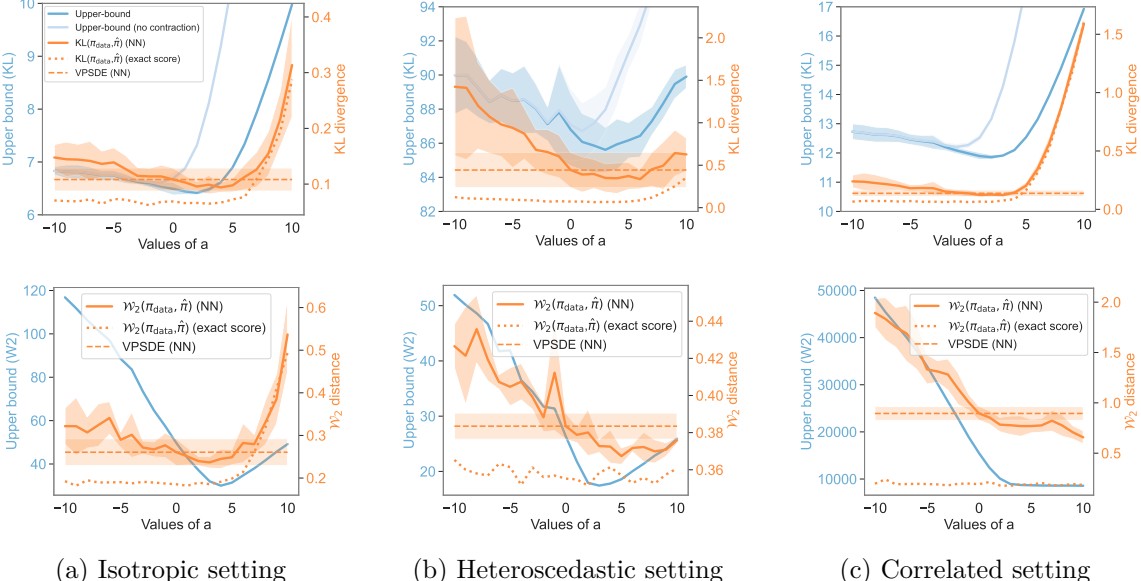

(a) Isotropic setting (b) Heteroscedastic setting (c) Correlated setting

Figure 10: Comparison of the empirical KL divergence (mean $\pm$ std over 30 runs) (top) and $\mathcal{W}_2$ distance (mean $\pm$ std over 10 runs) (bottom) between $\pi_{\text{data}}$ and $\widehat{\pi}_N^{(\beta,\theta)}$ (orange) and the related upper bounds (blue) from Theorem 3.1 and Theorem 4.2 across parameter $a$ for noise schedule $\beta_a$, $d = 50$ with **Exponential Integrator** discretization scheme. We also show the metrics for the linear VPSDE model (dashed line) and our model (dotted line) with exact score evaluation.

### E.1.5 Implementation of the score approximation in the Gaussian setting

Although the score function is explicit when $\pi_{\text{data}}$ is Gaussian (see Lemma E.1), we implement SGMs as done in applications, i.e., we train a deep neural network to witness the effect of the noising function on the approximation error. We train a neural network architecture $s_\theta(t, x) \in [0, T] \times \mathbb{R}^d \mapsto \mathbb{R}^d$ using the actual score function as a target:

$$\mathcal{L}_{\text{explicit}}(\theta) = \mathbb{E}\left[\left\|s_\theta\left(\tau, \overrightarrow{X}_\tau\right) - \nabla \log p_\tau\left(\overrightarrow{X}_\tau\right)\right\|^2\right]$$

$$= \mathbb{E}\left[\left\|s_\theta\left(\tau, \overrightarrow{X}_\tau\right) - (m_\tau^2 \Sigma_0 + \sigma_\tau^2 \mathrm{I}_d)^{-1}(\overrightarrow{X}_\tau + m_\tau \mu_0)\right\|^2\right],$$

where $t \to m_t$ and $t \to \sigma_t$ are defined in Lemma E.1 and $\tau \sim \mathcal{U}(0, T)$ is independent of $\overrightarrow{X}$. The neural network architecture chosen for this task is described in Figure 11. The width of each dense layer `mid_features` is set to 256 throughout the experiments.

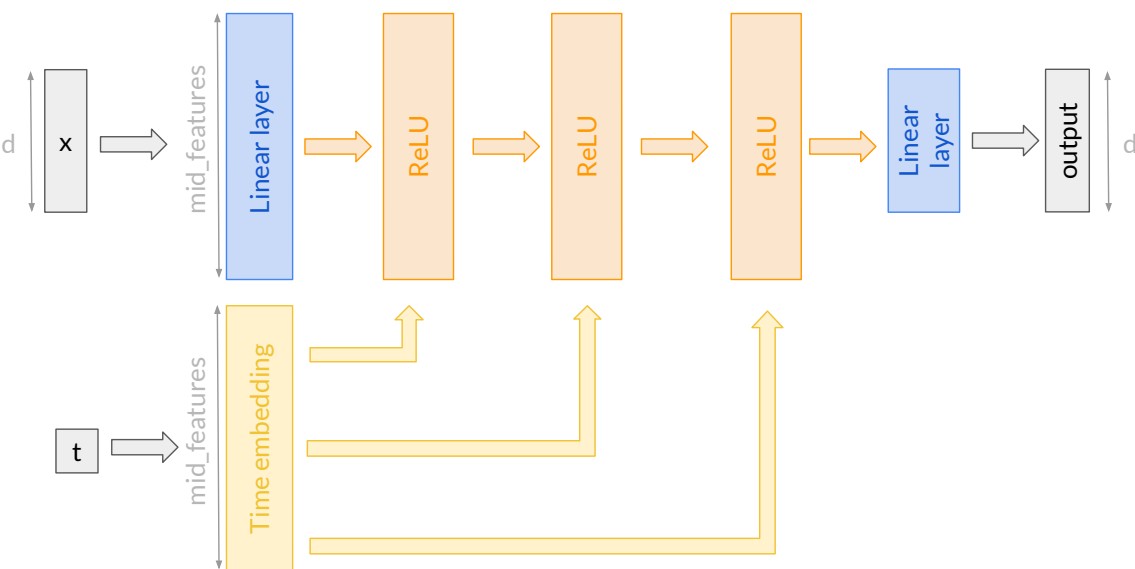

Figure 11: Neural network architecture. The input layer is composed of a vector $x$ in dimension $d$ and the time $t$. Both are respectively embedded using a linear transformation or a sine/cosine transformation (Nichol and Dhariwal, 2021) of width `mid_features`. Then, 3 dense layers of constant width `mid_features` followed by ReLu activations and skip connections regarding the time embedding. The output layer is linear resulting in a vector of dimension $d$.

### E.2 Details on the experiments and additional results

#### E.2.1 Illustration of the KL bound in the Gaussian setting

**Target distributions.** We investigate the relevancy of the upper bound from Theorem 3.1 for different noise schedules in the Gaussian setting. We use as a training sample $10^4$ samples with distribution $\mathcal{N}\left(\mathbf{1}_d, \Sigma\right)$ for $d$ the dimension of the target distribution with different choices of covariance structure.

1. (Isotropic) $\Sigma^{\text{iso}} = 0.5 \mathrm{I}_d$.

2. (Heteroscedastic) $\Sigma^{\text{heterosc}} \in \mathbb{R}^{d \times d}$ is a diagonal matrix such that $\Sigma^{\text{heterosc}}_{jj} = 1$ for $1 \leq j \leq d$, and $\Sigma^{\text{heterosc}}_{jj} = 0.01$ otherwise.

3. (Correlated) $\Sigma^{\text{corr}} \in \mathbb{R}^{d \times d}$ is a full matrix whose diagonal entries are equal to one and the off-diagonal terms are given by $\Sigma^{\text{corr}}_{jj'} = 1/\sqrt{|j - j'|}$ for $1 \leq j \neq j' \leq d$.

The resulting data distributions are respectively denoted by $\pi^{(\text{iso})}_{\text{data}}$, $\pi^{(\text{heterosc})}_{\text{data}}$ and $\pi^{(\text{corr})}_{\text{data}}$.

**Upper bound evaluation.** We leverage the Gaussian nature of the target distribution to compute explicitly all the terms in the bound. On the one hand, the relative entropy in $\mathcal{E}^{\text{KL}}_1$, $\mathrm{KL}\left(\pi_{\text{data}} \| \pi_\infty\right)$

is computed using the analytical formula for KL-divergence between two random Gaussian variable (Lemma E.2). On the other, the relative Fisher information in $\mathcal{E}_3^{\mathrm{KL}}$, $\mathcal{I}(\pi_{\mathrm{data}}|\pi_\infty)$, is computed using Lemma (E.4). Moreover, as the noise schedule function $\beta_a$ and its primitive are analytically known, every occurrences of either of them are explicitly computed. Finally, it remains to estimate the expectation in $\mathcal{E}_2^{\mathrm{KL}}(\theta, \beta)$. This is done via Monte Carlo estimation on 500 samples from the forward process (see Section E.1.2) for every step forward:

$$\frac{1}{500} \sum_{k=0}^{N-1} \sum_{i=1}^{500} \left\| \nabla \log \tilde{p}_{T-t_k} \left( \overrightarrow{X}_{T-t_k}^{(i)} \right) - \tilde{s}_\theta \left( T - t_k, \overrightarrow{X}_{T-t_k}^{(i)} \right) \right\|^2 \int_{T-t_{k+1}}^{T-t_k} \beta_a(t) \mathrm{d}t \, .$$

**SGM data generation in dimension 50.** In Figures 2 (top) of the main paper, we represent the following quantities in the same graph, in dimension $d = 50$, for different values of $a$.

- In blue the upper bound from Theorem 3.1. The dark blue color is used to refer to the upper bound with the contraction argument in equation (7) from Proposition C.1 while the ligther blue bound is the same bound without the contraction argument.

- In orange (dotted line) the KL divergence between the target distribution $\pi_{\mathrm{data}}$ and the empirical mean and covariance of the data generated using the true score function from Lemma E.1.

- In orange (plain line) we represent $\mathrm{KL}(\pi_{\mathrm{data}}\|\widehat{\pi}_N^{(\beta_a, \theta)})$ for $a \in \{-10, -9, -8, \dots, 10\}$. That is, the KL divergence between the target distribution $\pi_{\mathrm{data}}$ and the empirical mean and covariance of the data generated using the neural network architecture described in Figure 11 to approximate the score function.

- In orange (dashed line) we represent $\mathrm{KL}(\pi_{\mathrm{data}}\|\widehat{\pi}_N^{(\beta_0, \theta)})$. That is, the KL divergence between the target data $\pi_{\mathrm{data}}$ and the empirical mean and covariance of the data generated by the linear schedule VPSDE presented in Song et al. (2021) with the neural network architecture described in Figure 11.

We generate 10 000 samples. The batch size is set to 64 and neural networks are optimized with Adam. All the KL divergences written above are computed using Lemma E.2. Due to the stochastic nature of our experiments, they are repeated 30 times so that the corresponding mean value and standard deviations of these results are respectively depicted using plain and fill-in-between plots.

To disentangle the effect of each error term it is possible to plot the mixing time error $\mathcal{E}_1^{\mathrm{KL}}(\beta)$, the approximation error $\mathcal{E}_2^{\mathrm{KL}}(\theta, \beta)$ and the discretization error $\mathcal{E}_3^{\mathrm{KL}}(\beta)$ on a same graph for different values of $a$. However, for the schedule choice presented in Figure 1 as $\beta(T)$ is set to be 20 for every $a$ values it is pointless to display $\mathcal{E}_3^{\mathrm{KL}}(\beta)$ as it would not vary for different choices of schedule from $\beta_a$. The three error terms for Theorem 3.1, corresponding to the example in Figure 2 (top) are provided below in Figure 12.

**Optimal schedule versus classical choices.** We investigate the gain from using the parametric schedule with $a^\star$ minimising the upper bound from Theorem 3.1 for $d \in \{5, 10, 25, 50\}$ compared to using the linear and cosine schedules (see Appendix E.1.3) in the isotropic and correlated settings

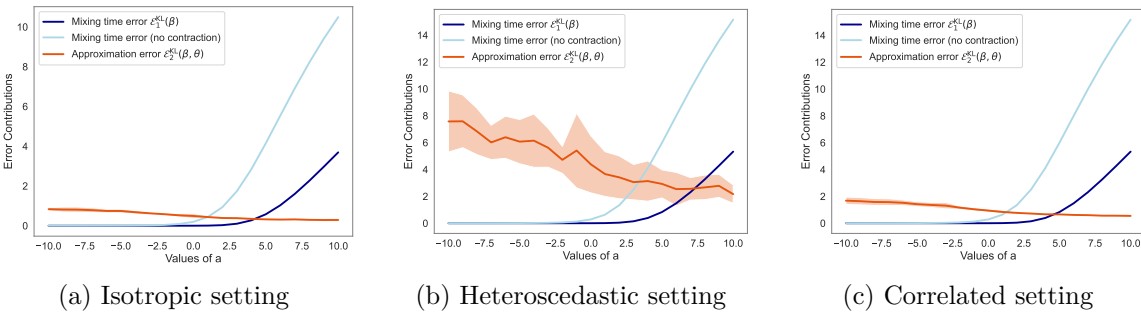

(a) Isotropic setting      (b) Heteroscedastic setting      (c) Correlated setting

Figure 12: Error terms contribution from Theorem 3.1 displayed from the same examples as in Figure 2 (top).

(as mentioned in Section 5.1, up to rescaling the heteroscedastic setting boils down to an isotropic setting).

To determine the optimal value $a^\star$, upper bounds were initially calculated across various dimensions for a range of $a$ values from $\{-10, -9, \ldots, 10\}$. This initial calculation aimed to identify a preliminary minimum value. Subsequently, the search was refined around these preliminary values using finer step-sizes of 0.25 to more precisely locate $a^\star$.

Results are given in tabular form in Table 1 and in Figure 13. The parametric schedule optimized to minimize the upper bound $\beta_{a^\star}$ consistently surpasses the linear schedule, delivering significant improvements. This enhanced performance is shown by lower average Kullback-Leibler divergence between $\pi_{\text{data}}$ and the generated sample distribution, as well as a reduction in the standard deviation of these divergences, which contributes to more stable generation. These results are competitive with or even exceed those obtained with state-of-the-art schedules such as the cosine schedule, particularly in higher dimensions $d = 25$ and $d = 50$. However, one should note that this comparison may not be entirely fair, as the cosine schedule increases unboundedly near $T$, whereas we capped the parametric schedule at $\beta(T) = 20$ to align with the linear schedule described in Song et al. (2021).

### E.2.2    Illustration of the Wasserstein bound in the Gaussian setting

**Target distributions.** The target distributions are Gaussian and are the same as for the the Kullback-Leibler bound: $\pi_{\text{data}}^{(\text{iso})}$, $\pi_{\text{data}}^{(\text{heterosc})}$ and $\pi_{\text{data}}^{(\text{corr})}$.

**Upper bound evaluation.** We leverage the Gaussian nature of the target distribution to compute explicitly all the terms in the bound from Theroem 4.2. For the mixing time $\mathcal{E}_1^{\mathcal{W}_2}$, the strong log-concavity constant $\bar{C}_t$ is derived using Lemma 4.1 and $\mathcal{W}_2(\pi_{\text{data}}, \pi_\infty)$ is derived using Lemma E.3. For $\mathcal{E}_2^{\mathcal{W}_2}$, the analytical expressions for $\bar{L}_t$ is given in Lemma 4.1 and an upper bound to $M$ is derived in Proposition E.5. All non analytically solvable integrals estimated numerically using the trapezoidal rule, implemented with the built-in PyTorch function `torch.trapezoid`. To estimate $\varepsilon$, we use Monte-Carlo simulations with 500 samples (in the same manner as for the Kullback-Leibler

|  | Dimension | 5 | 10 | 25 | 50 |
|---|---|---|---|---|---|
| Isotropic | Upper bound min $a^\star$ | 1.75 | 1.00 | 1.50 | 2.00 |
|  | Generation value in $a^\star$ | $0.001607 \pm 0.000462$ | $0.005343 \pm 0.001155$ | $\mathbf{0.026724} \pm 0.004046$ | $\mathbf{0.095981} \pm 0.005485$ |
|  | VPSDE (linear sched.) | $0.001935 \pm 0.000405$ | $0.005594 \pm 0.001377$ | $0.031748 \pm 0.006158$ | $0.105592 \pm 0.019529$ |
|  | Cosine schedule | $\mathbf{0.001390} \pm 0.000296$ | $\mathbf{0.005097} \pm 0.001064$ | $0.026900 \pm 0.001859$ | $0.099917 \pm 0.004375$ |
|  | % gain (vs VPSDE) | +16.93 % | +4.48 % | +15.80 % | +9.10 % |
|  | % gain (vs Cosine) | -15.61 % | -4.83 % | +0.66 % | +3.94 % |
| Correlated | Upper bound min $a^\star$ | 2.25 | 1.75 | 1.75 | 2.25 |
|  | Generation value in $a^\star$ | $0.001861 \pm 0.000880$ | $0.005871 \pm 0.001165$ | $\mathbf{0.033156} \pm 0.003785$ | $\mathbf{0.109649} \pm 0.008056$ |
|  | VPSDE (linear sched.) | $0.002568 \pm 0.002708$ | $0.006210 \pm 0.001816$ | $0.038434 \pm 0.010313$ | $0.134716 \pm 0.016541$ |
|  | Cosine schedule | $\mathbf{0.001197} \pm 0.000332$ | $\mathbf{0.005515} \pm 0.000775$ | $0.040430 \pm 0.003475$ | $0.110515 \pm 0.004646$ |
|  | % gain (vs VPSDE) | +27.53 % | +5.46 % | +13.74 % | +18.63 % |
|  | % gain (vs Cosine) | -55.47 % | -6.46 % | +17.98 % | +0.78 % |
| Parameters | Learning rate | 1e-4 | 1e-4 | 1e-3 | 1e-3 |
|  | Epochs | 20 | 30 | 75 | 150 |

Table 1: Comparison of the KL divergence between the target value and the generated value at $a^\star$ (the minimum value of the upper bound from Theorem 3.1) with the KL divergence between the generated value by VPSDE with linear schedule and the target distribution. We display average KL divergences plus or minus standard deviations over 10 runs. The target distributions are chosen to be Gaussian with different covariance structures: isotropic ($\pi_{\mathrm{data}}^{(\mathrm{iso})}$), heteroscedastic ($\pi_{\mathrm{data}}^{(\mathrm{heterosc})}$) and correlated ($\pi_{\mathrm{data}}^{(\mathrm{corr})}$).

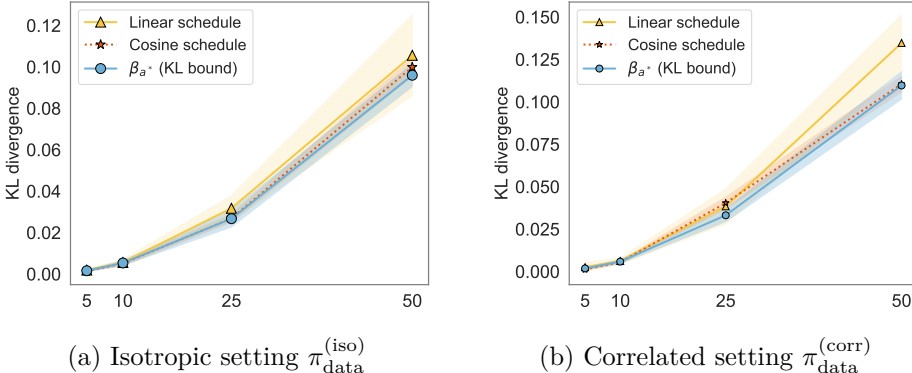

(a) Isotropic setting $\pi_{\mathrm{data}}^{(\mathrm{iso})}$      (b) Correlated setting $\pi_{\mathrm{data}}^{(\mathrm{corr})}$

Figure 13: Comparison of the empirical KL divergence (mean value $\pm$ std over 10 runs) between $\pi_{\mathrm{data}}$ and the generative distribution $\hat{\pi}$ for different values of the dimension. The generative distributions considered are $\hat{\pi}_N^{(\beta_{a^\star},\theta)}$ obtained by the time-inhomogeneous SGM for $\beta_{a^\star}$ (blue plain), $\hat{\pi}_N^{(\beta_0,\theta)}$ obtained by a standard linear VPSDE model (yellow dashed) and $\hat{\pi}_N^{(\beta_{\cos},\theta)}$ obtained by using a cosine schedule (orange dotted).

bound):

$$\sup_{k \in \{0,\ldots,N-1\}} \sqrt{\frac{1}{500} \sum_{i=1}^{500} \left\| \nabla \log \tilde{p}_{T-t_k}\left(\overrightarrow{X}_{T-t_k}^{(i)}\right) - \tilde{s}_\theta\left(T-t_k, \overrightarrow{X}_{T-t_k}^{(i)}\right) \right\|^2}.$$

**SGM data generation dimension 50.** In Figures 14 (and Figures 2 (bottom) of the main paper) we represent on the same graph, in dimension $d = 50$, for different values of a:

- in blue the upper bound from Theorem 4.2.

- in orange (dotted line) the $\mathcal{W}_2$ distance between the target distribution $\pi_{\text{data}}$ and the empirical mean and covariance of the data generated using the true score function from Lemma E.1.

- in orange (plain line) we represent $\mathcal{W}_2(\pi_{\text{data}}, \widehat{\pi}_N^{(\beta_a, \theta)})$ for $a \in \{-10, -9, -8, .., 10\}$. That is, the $\mathcal{W}_2$ distance between the target distribution $\pi_{\text{data}}$ and the empirical mean and covariance of the data generated using the neural network architecture described in Figure 11 to approximate the score function

- in orange (dashed line) we reprensent $\mathcal{W}_2(\pi_{\text{data}}, \widehat{\pi}_N^{(\beta_0, \theta)})$. That is, the $\mathcal{W}_2$ distance between the target data $\pi_{\text{data}}$ and the empirical mean and covariance of the data generated by the linear schedule VPSDE presented in Song et al. (2021) with the neural network architecture described in Figure 11.

First results. We generate 10 000 samples. The batch size is set to 64 and neural networks are optimized with Adam. All the $\mathcal{W}_2$ distances written above are computed using Lemma E.3. Due to the stochastic nature of our experiments, they are repeated ten times so that the corresponding mean value and standard deviations of these results are respectively depicted using plain and fill-in-between plots.

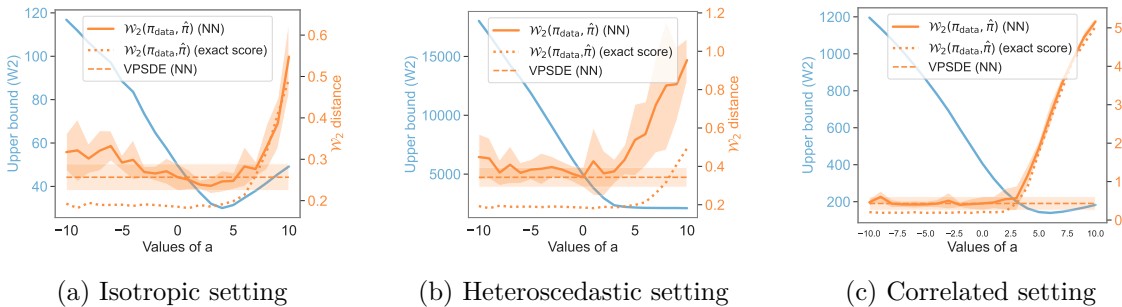

| (a) Isotropic setting | (b) Heteroscedastic setting | (c) Correlated setting |

Figure 14: Comparison of the empirical 2-Wasserstein distance (mean value $\pm$ std over 10 runs) between $\pi_{\text{data}}$ and $\widehat{\pi}_N^{(\beta_a, \theta)}$ (in orange) and the upper bound from Theorem 4.2 (in blue) w.r.t. the parameter $a$ used in the definition of the noise schedule $\beta_a$, for $d = 50$. We also represent the 2-Wasserstein distances obtained with the linear VPSDE model (dashed line) and the one obtained with the parametric model (dotted line) when the score is not approximated but exactly evaluated. The data distribution $\pi_{\text{data}}$ is chosen Gaussian, corresponding to (a) $\pi_{\text{data}}^{(\text{iso})}$, (b) $\pi_{\text{data}}^{(\text{heterosc})}$ and (c) $\pi_{\text{data}}^{(\text{corr})}$.

Performances obtained from raw distributions for $\pi_{\text{data}}^{(\text{iso})}$, $\pi_{\text{data}}^{(\text{heterosc})}$ and $\pi_{\text{data}}^{(\text{corr})}$ are displayed in Figure 14. In the isotropic case (Figure 14 (a)) the curve for the upper bound (blue line) points a global minimum near the minimal values obtain by $\mathcal{W}_2(\pi_{\text{data}}, \widehat{\pi}_N^{(\beta_a, \theta)})$ (plain orange line), which underlines

that the upper bound is indeed informative in such a case. However, the upper bounds obtained for the heteroscedastic and correlated settings (Figure 14 (b,c)) are not in line with the generation results.

These observed discrepancies can be linked to the conditioning of the covariance matrices. In both heteroscedastic and correlated cases, the largest eigenvalue of the covariance matrices is not smaller than the variance stationary distribution of the forward process (set to $\sigma^2 = 1$ in those experiments) violating the requirements of Lemma 4.1 ($\lambda_{\max}\left(\Sigma^{(\text{heterosc})}\right) = 1$ and $\lambda_{\max}\left(\Sigma^{(\text{corr})}\right) \approx 15$). This induces the default of strong log-concavity of the renormalized densities $\tilde{p}_t$. In this way, the Gaussian scenario highlights the critical influence of the covariance matrix conditioning on SGMs. Additionally, a smaller $\lambda_{\min}(\Sigma)$ would increase $L_t$ and $M$, which in turn would increase the bound from Theorem 4.2.

Data preprocessing. As frequently done in practice, we expect better conditioning by running SGMs on a standardized distribution. In this way, note that if $X_0 \sim \pi_{\text{data}}$ we consider the centered standardized distribution $X_{\text{stand}} = D\left(X_0 - \mu\right)$ with $D = \text{diag}(\sigma_1, \ldots, \sigma_d) \in \mathbb{R}^{d \times d}$ a diagonal matrix with diagonal entries $\sigma_j$ corresponding to the standard deviation of the $j$-th component of $X_0$ and with $\mu = \left[\mathbb{E}\left[X_{0,1}\right], \ldots, \mathbb{E}\left[X_{0,d}\right]\right]^\top$. A last transformation shrinks the data into a rescaled version of $X_{\text{stand}}$ defined as $X_{\text{scale}} = \kappa D\left(X_0 - \mu\right)$ with $\kappa := 1/(2\lambda_{\max}\left(\Sigma_{(\text{stand})}\right))^{1/2}$, where $\lambda_{\max}\left(\Sigma_{(\text{stand})}\right)$ is the largest eigenvalue of the covariance matrix of $X_{\text{stand}}$. We then train SGMs to approximate the distribution of $X_{\text{scale}}$. By doing so we ensure the applicability of Lemma 4.1 (with $\sigma^2 = 1$), as the largest eigenvalue of the covariance matrix of $X_{\text{scale}}$ is no larger than 0.5.

Adapted upper bound. We can finally adapt the upper bound from Theorem 4.2 to a rescaled setting by noting that

$$\mathcal{W}_2\left(\pi_{\text{data}}, \tilde{\pi}\right) \leq \frac{1}{\kappa}\left(\max_{1 \leq j \leq d} \sigma_j\right) \mathcal{W}_2\left(\pi_{\text{scale}}, \widehat{\pi}_{N,\text{scale}}^{(\beta_a, \theta)}\right), \tag{51}$$

where

- $\pi_{\text{scale}}$ is the distribution of scaled sample $X_{\text{scale}}$

- $\widehat{\pi}_{N,\text{scale}}^{(\beta_a, \theta)}$ corresponds to the distribution of SGM trained on $X_{\text{scale}}$

- $\tilde{\pi}$ is the distribution of the descaled generated samples, i.e., the distribution of $D^{-1}X/\kappa + \mu$ with $X \sim \widehat{\pi}_{N,\text{scale}}^{(\beta_a, \theta)}$.

Therefore, we can evaluate the upper bound of Theorem 4.2 for scaled samples (r.h.s. of (51)), and transfer it up to a constant to descaled generated samples (l.h.s. of (51)).

Results with scaled data preprocessing. The results are detailed in Figure 15 for the heteroscedastic case (e) $\pi_{\text{scale}}^{(\text{heterosc})}$ and the correlated case (f) $\pi_{\text{scale}}^{(\text{corr})}$, and are discussed extensively in Section 5.1 of the main paper. Note that the minima of the evaluated bounds now align closely with the empirical metrics. However, the upper bound profile for the correlated case has been shifted up. This increase was anticipated due to the effect of rescaling by the largest eigenvalue of $\Sigma_{(\text{stand})}^{\text{corr}}$, approximately 15, which reduces the magnitude of the values in $\pi_{\text{scale}}^{(\text{corr})}$. This tends to increase the values of $L_t$ and $M$ through the effect on $\lambda_{\min}(\Sigma_{(\text{stand})}^{(\text{corr})})$ as explained above. Despite this effect, these experiments confirm the overall utility of the bound for selecting the appropriate noise schedule. The effect of

data rescaling on the Lispschitz continuity and log concavity of the true score function $\nabla \log p_t$ are illustrated in Figure 16 on the Heteroscedastic setting.

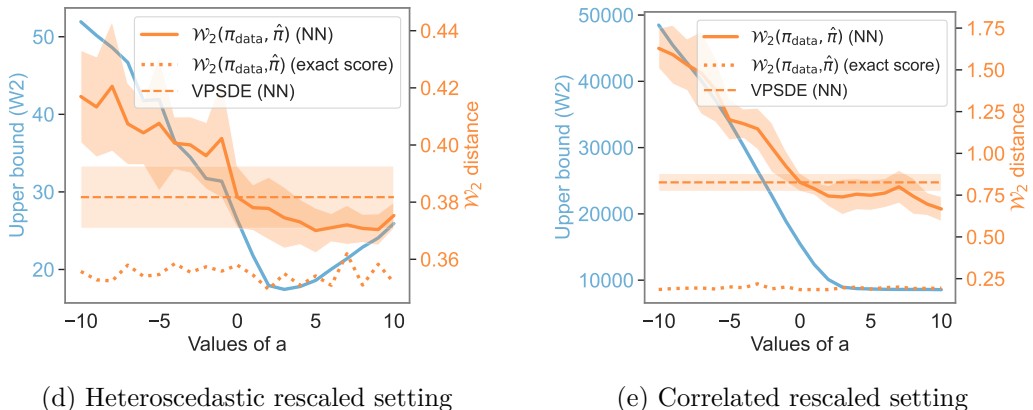

(d) Heteroscedastic rescaled setting          (e) Correlated rescaled setting

Figure 15: Comparison of the empirical 2-Wasserstein distance on rescaled datasets for (d) $\pi_{\text{scale}}^{(\text{heterosc})}$, (e) $\pi_{\text{scale}}^{(\text{corr})}$.

**Optimal schedule versus classical choices.** We investigate the gain from using SGM with the schedule $a^\star$ minimising the upper bound from Theorem 4.2 for $d \in \{5, 10, 25, 50\}$ compared to the linear and cosine schedules (see Appendix E.1.3). To determine the optimal value $a^\star$, upper bounds were initially calculated across various dimensions for a range of $a$ values from $\{-10, -9, \dots, 10\}$. This initial calculation aimed to identify a preliminary minimum value. Subsequently, the search was refined around these preliminary values using finer step-sizes of 0.25 to more precisely locate $a^\star$.

Results for the isotropic, heteroscedastic, and correlated cases are presented in both tabular form in Table 2 and visually in Figure 3 within the main paper. These findings are discussed in Section 5.1 of the main paper.

### E.2.3 Numerical experiments on more complex synthetic data

In the context of complex data distributions, the Kullback-Leibler bound (Theorem 3.1) appears to be of limited practical applicability. Specifically, $\mathcal{E}_2^{\text{KL}}(\theta, \beta)$ implies that for each noise schedule tested, a distinct score approximation $\tilde{s}_\theta(t, x)$ must be trained. This requirement renders the bound computationally intensive and therefore not realistically usable. Additionally, $\mathcal{E}_3^{\text{KL}}(\beta)$ is independent of the schedule choice over $(0, T)$, as it depends solely on its final value $\beta(T)$ which is set constant in our empirical setting (for all $a$, $\beta_a(T) = 20$). As a consequence, the last remaining error term to analyse the bound through the lens of noise schedules is the mixing time $\mathcal{E}_1^{\text{KL}}(\beta)$. However, relying exclusively on $\mathcal{E}_1^{\text{KL}}(\beta)$ would suggest selecting a schedule $t \mapsto \beta(t)$ that maximises $\int_0^T \beta(t)\mathrm{d}t$. As demonstrated in Section 5.1, this approach clearly fails to yield the schedule choices near the optimal solution.

Therefore, a more reliable choice would be to use the $\mathcal{W}_2$ bound of Theorem 4.2 for which most of the terms can be computed explicitly with reasonable computational cost in the Gaussian setting.

| | Dimension | 5 | 10 | 25 | 50 |
|---|---|---|---|---|---|
| Isotropic | Upper bound min $a^\star$ | 4.5 | 4.25 | 3.75 | 4.25 |
| | Generation value in $a^\star$ | $0.039241 \pm 0.012572$ | $\mathbf{0.059274} \pm 0.009438$ | $\mathbf{0.130829} \pm 0.014245$ | $\mathbf{0.233812} \pm 0.010584$ |
| | VPSDE (linear sched.) | $0.036995 \pm 0.004663$ | $0.063939 \pm 0.010876$ | $0.141601 \pm 0.020447$ | $0.256384 \pm 0.032709$ |
| | Cosine schedule | $\mathbf{0.030996} \pm 0.003254$ | $0.060649 \pm 0.007117$ | $0.131234 \pm 0.004794$ | $0.251959 \pm 0.005588$ |
| | % gain (vs VPSDE) | -6.07 % | +7.30 % | +7.61 % | +8.79 % |
| | % gain (vs Cosine) | -26.60 % | +2.26 % | +0.31 % | +7.20 % |
| Heterosc. (with rescaling) | Upper bound min $a^\star$ | 4.00 | 3.25 | 2.00 | 2.75 |
| | Generation value in $a^\star$ | $0.096592 \pm 0.003062$ | $\mathbf{0.143224} \pm 0.004899$ | $\mathbf{0.242493} \pm 0.004769$ | $\mathbf{0.372292} \pm 0.004694$ |
| | VPSDE (linear sched.) | $0.098889 \pm 0.003604$ | $0.147478 \pm 0.009638$ | $0.249144 \pm 0.011394$ | $0.385612 \pm 0.009333$ |
| | Cosine schedule | $\mathbf{0.096437} \pm 0.002380$ | $0.143701 \pm 0.002460$ | $0.250520 \pm 0.004448$ | $0.374868 \pm 0.003243$ |
| | % gain (vs VPSDE) | +2.32 % | +2.89 % | +2.67 % | +3.46 % |
| | % gain (vs Cosine) | -0.16 % | +0.33 % | +3.20 % | +0.69 % |
| Correlated (with rescaling) | Upper bound min $a^\star$ | 8.00 | 8.75 | 10.50 | 11.00 |
| | Generation value in $a^\star$ | $0.066548 \pm 0.013873$ | $\mathbf{0.107291} \pm 0.028454$ | $\mathbf{0.261075} \pm 0.029533$ | $\mathbf{0.676151} \pm 0.123277$ |
| | VPSDE (linear sched.) | $0.072068 \pm 0.019861$ | $0.138240 \pm 0.031119$ | $0.302986 \pm 0.045539$ | $0.897584 \pm 0.079860$ |
| | Cosine schedule | $\mathbf{0.048276} \pm 0.008605$ | $0.112898 \pm 0.011284$ | $0.391753 \pm 0.030112$ | $0.765524 \pm 0.022376$ |
| | % gain (vs VPSDE) | +7.65 % | +22.36 % | +13.81 % | +24.68 % |
| | % gain (vs Cosine) | -37.77 % | +4.96 % | +33.31 % | +11.67 % |
| Parameters | Learning rate | 1e-4 | 1e-4 | 1e-3 (1e-4 for Corr.) | 1e-3 (1e-4 for Corr.) |
| | Epochs | 20 | 30 | 75 | 150 |

Table 2: Comparison of the $\mathcal{W}_2$ distance between the target value and the generated value at $a^\star$ (the minimum value of the upper bound from Theorem 4.2) with the $\mathcal{W}_2$ distance between the generated value by VPSDE and the target distribution. We display averages plus or minus standard deviations over 10 runs. The target distributions are chosen to be Gaussian with different covariance structures: isotropic, heteroscedastic (with rescaling applied), and correlated (with rescaling applied).

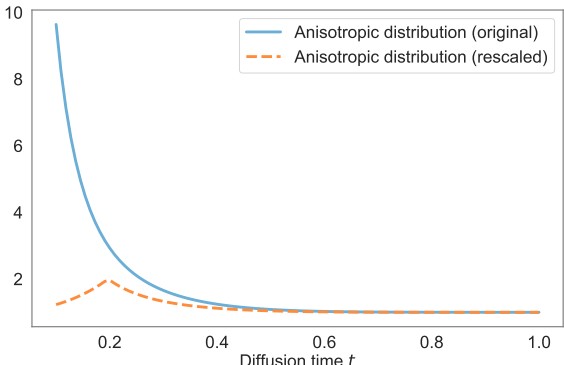

Figure 16: Comparison of the ratio strong concavity / Lipschitz continuity for the true score function $\nabla \log p_t$ in the Heteroscedastic setting before rescaling (Figure 14 (b)) and after rescaling (Figure 15 (d)) throughout diffusion time $t \in (0, T]$.

In particular, we leverage the Gaussian framework to estimate the constant terms and apply the rescaling defined in Appendix E.2.2 to ensure that $C_t$ is non negative for $t \in (0, T]$. More precisely,

- $L_t$ and $C_t$ are given in Lemma 4.1 and are computed using the empirical covariance matrix associated with $\pi_{\text{scale}}$ (and using when applicable the refinements in Propositions D.1 and D.2),

- $M$ is derived with Proposition E.5 with appropriate empirical estimators,

- $\mathcal{W}_2(\pi_{\text{data}}, \pi_\infty)$ is computed using closed-form formulas for Gaussian distributions, involving empirical estimators of the mean and covariance of $\pi_{\text{scale}}$,

- the term $\varepsilon$ is deliberately omitted to avoid the prohibitively high computational costs associated with training distinct models for different noise schedules.

The experiments are run using the same neural network architecture as in the Gaussian illustrations of Appendices E.2.1 and E.2.2 (i.e., a dense neural network with 3 hidden layers of width 256). The network was trained over 200 epochs for $a \in \{-10, -9, \ldots, 19, 20\}$. Contrary to the Gaussian case, conditional score matching $\mathcal{L}_{\text{score}}(\theta)$ (5) is used, as being closer to what is done in practice (explicit scores are now out of reach). To assess the quality of the data generation three metrics are used:

(a) an estimator of the KL-divergence based on $k$-nearest neighbors (Wang et al., 2009) with $\lceil \sqrt{d} \rceil$ neighbors,

(b) the sliced 2-Wasserstein distance (Flamary et al., 2021) with 2000 projections,

(c) the negative log likelihood computed on 1000 samples defined as $-\frac{1}{1000} \sum_{i=1}^{1000} \log \pi_{\text{data}}(x_i)$ with $(x_i)_{1 \leq i \leq 1000}$ samples from the generated distribution and $\pi_{\text{data}}$ the probability density function to be estimated.

**Funnel distribution.** The first distribution considered is the Funnel distribution (Thin et al., 2021) in dimension 50, defined as

$$\pi_{\text{data}}(x) = \varphi_{a^2}(x_1) \prod_{j=2}^{d} \varphi_{\exp(2bx_1)}(x_j),$$

with $a = 1$ and $b = 0.5$. To ensure the applicability of Theorem 4.2 and Lemma 4.1 the samples are standardized and rescaled according to the method described in Appendix E.2.2. The results, illustrated in Figures 4 and 17, show that the upper bounds effectively mirror the generation outcomes across the three metrics considered. Moreover, the generation results for the parametric schedule $a^\star$ (the one that minimizes the upper bound) outperforms in all three metrics both the linear and cosine schedules (see Table 3).

**Gaussian mixture models.** The second distribution considered is a Gaussian mixture model with 25 modes in dimension 50, defined as

$$\pi_{\text{data}}(x) = \frac{1}{25} \sum_{(j,k) \in \{-2,\ldots,2\}^2} \varphi_{\mu_{jk}, \Sigma_d}(x)$$

with $\varphi_{\mu_{jk}, \Sigma_d}$ denoting the probability density function of the Gaussian distribution with covariance matrix $\Sigma_d = \text{diag}(0.01, 0.01, 0.1, ..., 0.1)$ and mean vector $\mu_{jk} = [j, k, 0, 0, 0..., 0]^\top$. The results shown in Figure 18 and Table 3 confirm the relevance of the upper bound even for non-Gaussian datasets.

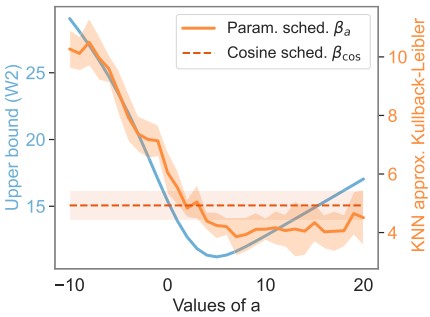 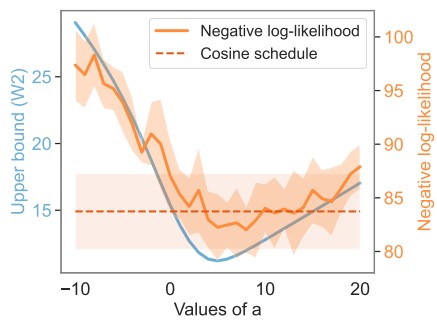

(b) KL divergence with $k$-nearest neighbors estimate     (c) Negative log-likelihood

Figure 17: Upper bound and empirical distances between the data distribution and the generated samples for different metrics on a Funnel dataset in dimension 50.

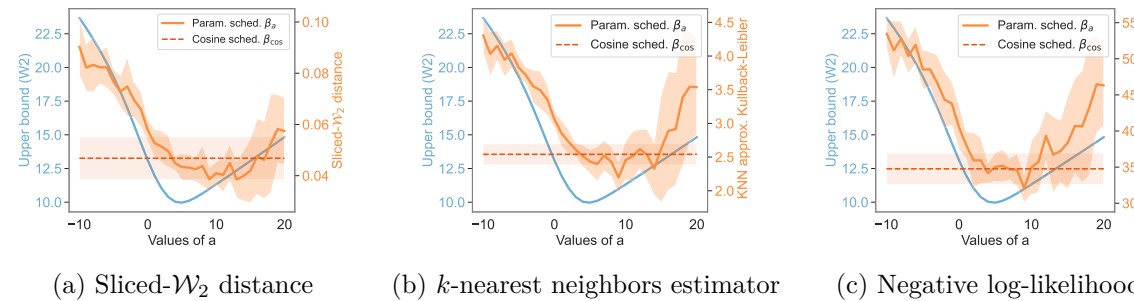

(a) Sliced-$\mathcal{W}_2$ distance     (b) $k$-nearest neighbors estimator     (c) Negative log-likelihood

Figure 18: Upper bound and empirical distances between the data distribution and the generated samples for different metrics on a mixture of 25 Gaussian variables dataset in dimension 50.

## E.3   Numerical experiments on real-world datasets

To evaluate the impact of the noise schedule on the performance of score-based generative models we evaluate the parametric family $\beta_a$ introduced in Equation (9) using CIFAR 10 dataset. We suggest to analyse the FID (Fréchet Inception Distance) score on 50 000 samples generated for different noise schedules (different values of $a$ in $\beta_a$, see Figure 1) on CIFAR 10.

We use pretrained models from Karras et al. (2022) with the recommended hyperparameters designed to replicate the experiments in Song et al. (2021) corresponding to our linear schedule ($a = 0$) as shown in Figure 1. In particular, we let $T = 1$, $\beta(0) = 0.1$, $\beta(T) = 20$, 1000 discretization steps and sample over the diffusion $[\epsilon, 1]$ with $\epsilon = 10^{-3}$.

The training process in Karras et al. (2022) is slightly different, though equivalent, to the original implementation. In particular, the networks are not trained to directly estimate $\nabla \log p_t(\overrightarrow{X}_t)$. Instead, a denoiser function $D_\theta(X, \sigma)$ is trained to isolate the noise from the signal for some noise level (see Equations (2) and (3) in Karras et al. (2022)). With appropriate rescaling this denoiser

|  | Metric | Sliced-Wasserstein | $k$-nn (Kullback-Leibler) | NLL |
|---|---|---|---|---|
| Funnel distribution | Generation value in $a^\star$ | **0.218498** ± 0.049882 | **4.242455** ± 0.450224 | **82.25179** ± 3.12809 |
|  | VPSDE (linear sched.) | 0.240664 ± 0.036578 | 6.048403 ± 0.726221 | 87.02893 ± 3.40642 |
|  | Cosine schedule | 0.221851 ± 0.054309 | 4.927209 ± 0.510968 | 83.73294 ± 3.53262 |
|  | % gain (vs VPSDE) | +9.21 % | +29.88 % | +5.49 % |
|  | % gain (vs Cosine) | +1.51 % | +13.91 % | +1.77 % |
| Gaussian mixture models | Generation value in $a^\star$ | **0.043388** ± 0.005222 | **2.433759** ± 0.180652 | 35.033176 ± 1.97863 |
|  | VPSDE (linear sched.) | 0.057763 ± 0.004450 | 3.063054 ± 0.126697 | 40.49867 ± 3.13705 |
|  | Cosine schedule | 0.046816 ± 0.008402 | 2.541213 ± 0.158563 | **34.76353** ± 2.20980 |
|  | % gain (vs VPSDE) | +24.91 % | +20.55 % | +13.49 % |
|  | % gain (vs Cosine) | +7.32 % | +4.23 % | -0.77 % |
| Parameters | Learning rate | 1e-3 | 1e-3 | 1e-3 |
|  | Epochs | 200 | 200 | 200 |

Table 3: Comparison of the sliced-$\mathcal{W}_2$ distance, KL divergence coupled with $k$-nearest neighbors estimate and negative log-likelihood between the target distribution and the SGM-generated one. For the latter, the SGM is either trained with linear, cosine and $\beta_{a^\star}$ schedules. We display averages plus or minus standard deviations over 10 runs. The target distributions are chosen are Funnel and Gaussian mixture models.

can be used in the VP setting by letting

$$
s_\theta(\overrightarrow{X}_t, t) = \frac{\sigma_t^2}{m_t} \left( D_\theta \left( \frac{\overrightarrow{X}_t}{m_t}, \frac{\sigma_t}{m_t} \right) - \frac{\overrightarrow{X}_t}{m_t} \right) ,
$$

where $s_\theta$ is the score approximation as defined in our paper, $m_t = \exp\{ - \int_0^t \beta(s) \mathrm{d}s / (2\sigma^2) \}$ and $\sigma_t^2 = \sigma^2 (1 - m_t^2)$. This formulation bridges the denoising approach with score-based methods in the VP framework.

Figure 19 displays the FID score for samples generated using the Euler-Maruyama discretization of the backward process for different choices of $\beta_a$ with $a \in \{-10, -9, \ldots, 10\}$ and cosine schedule $\beta_{\cos}$. Although the assumptions of our results cannot be verified in such a setting, it is interesting to note that the empirical performance follows the same dynamics as in the toy numerical experiments. This indicates that the analysis and optimization of noise schedules is an interesting problem to be explored further for complex cases.

## F   Conditional training in the Gaussian setting

Section 5.1 of this paper is dedicated to the illustration of the theoretical upper bounds and their relevance in the Gaussian setting (i.e., when $\pi_{\mathrm{data}}$ is Gaussian). This choice has been motivated by the fact that, under this setting, all constants in the upper bounds from Theorem 3.1 and Theorem 4.2 are either analytically available or could be precisely estimated (see Appendices E.2.1 and E.2.2).

In particular, both upper bounds display error terms proportional to $\mathcal{L}_{\mathrm{explicit}}(\theta)$ (4), which has motivated the use of explicit score matching during the training. To do so, we used a deep neural architecture (see Figure 11) trained to minimize $\mathcal{L}_{\mathrm{explicit}}(\theta)$ (4) using as a target the true score function. This is possible because in the Gaussian setting, the true score function is analytically known (Lemma E.1). However, in most applications the score function is not available, because

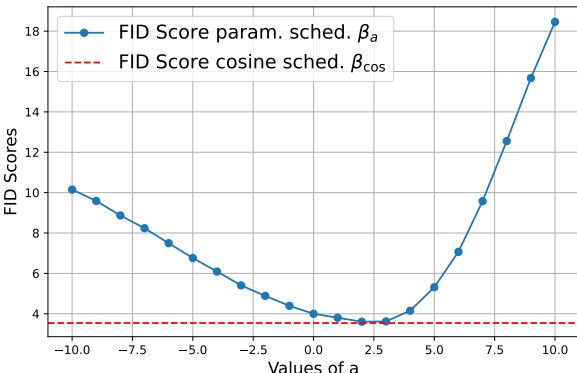

Figure 19: FID scores on 50 000 generated samples using pre-trained models from Karras et al. (2022) with different noise schedules from the parametric family (9) and cosine schedule.

the data distribution is not known and has to be learned. This is the reason why, in practice we rely on conditional score matching (i.e., the minimization of $\mathcal{L}_{\text{score}}(\theta)$ (5)). This approach is particularly relevant given the relationship between the explicit and conditional score functions:

$$\mathcal{L}_{\text{explicit}}(\theta) = \mathcal{L}_{\text{score}}(\theta) - \mathbb{E}\left[\|\nabla \log p_\tau(\overrightarrow{X}_\tau) - \nabla \log p_\tau(\overrightarrow{X}_\tau|X_0)\|^2\right].$$

Consequently, all the theoretical upper bounds discussed in Sections 3 and 4 can be adjusted by a constant (with respect to $\theta$) to account for discrepancies between the score function learned through $\mathcal{L}_{\text{score}}$ or $\mathcal{L}_{\text{explicit}}$.

The rest of this section demonstrates the numerical effects of employing conditional score matching instead of explicit score matching, following the numerical set-up of Appendices E.2.1 and E.2.2. In Figure 20, the Kullback-Leibler upper bound from Theorem 3.1 is depicted in varying shades of blue, while the empirical $\text{KL}(\pi_{\text{data}}||\widehat{\pi}_N^{(\beta_a,\theta)})$ across parameters $a \in \{-10, -9, -8, ..., 10\}$ is shown in varying shades of orange.

In Figure 20, three learning scenarios are presented: one using explicit score matching (which exactly matches the results of Figure 2 (top)), another with conditional score matching over 150 epochs, and a third with 300 epochs. Both the generation results and the upper bounds show diminished performance as the curves are shifted upwards. Nonetheless, the overall curve shapes are similar, and the optimal points remain closely aligned. Interestingly, both the upper bounds and the generation outcomes in the conditional scenarios demonstrate more pronounced peaks near the minimum values. This suggests that precise noise schedule selection may yield even better performance gain when SGMs are trained using conditional score matching.

Additionally, Figure 21 demonstrates that increasing the number of training iterations when using conditional score matching provides results more and more similar to that obtained with explicit score matching. This effect is noticeable in both the KL divergence and the $\mathcal{W}_2$ distance.

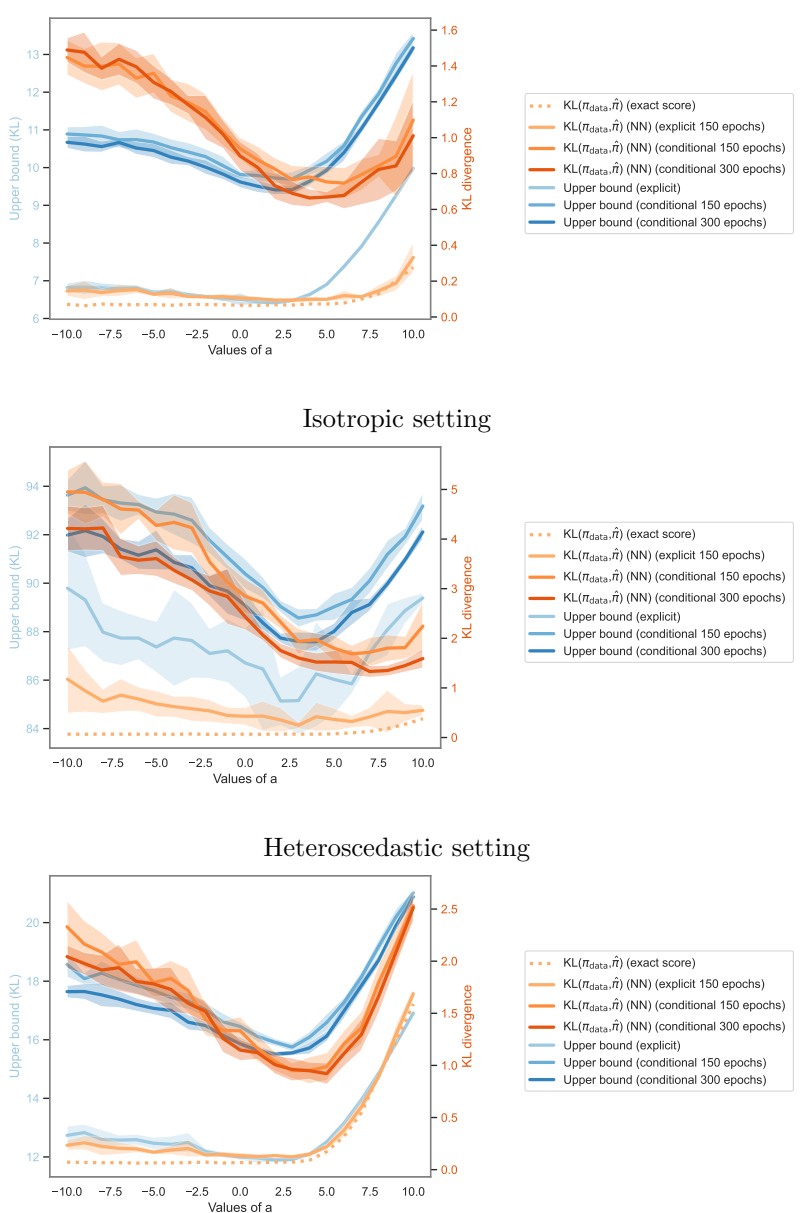

Figure 20: Comparison of the empirical KL divergence (mean value $\pm$ std over 10 runs) between $\pi_{\text{data}}$ and $\widehat{\pi}_N^{(\beta_a,\theta)}$ (in orange) and the upper bound of Theorem 3.1 (in blue) w.r.t. the parameter $a$ used in the definition of the noise schedule $\beta_a$, for $d = 50$.

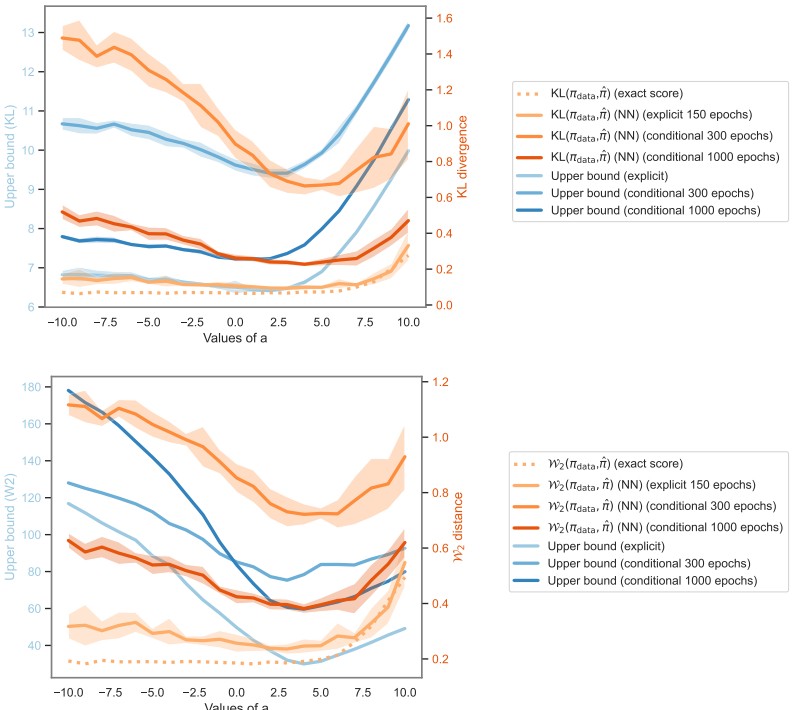

Figure 21: Comparison of the empirical KL divergence (top) and the $\mathcal{W}_2$ distance (bottom) (mean value $\pm$ std over 10 runs) between $\pi_{\text{data}} = \pi_{\text{data}}^{(\text{iso})}$ and $\widehat{\pi}_N^{(\beta_a,\theta)}$ (in orange) and the upper bound of Theorem 3.1 (top) and of Theorem 4.2 (bottom) (in blue) w.r.t. the parameter $a$ used in the definition of the noise schedule $\beta_a$, for $d = 50$.

