# OpenReview forum: "An analysis of the noise schedule for score-based generative models"
_TMLR — Accepted by TMLR_

### Review · Reviewer_425e · 2024-10-29

**Summary Of Contributions:**

This paper provides refined theoretical upper bounds on the KL divergence and Wasserstein distance between samples from a diffusion model and the data distribution. The bounds describe the dependence of the errors on the mixing behavior of the forward process, the approximation error of the (normalized) score, and the discretization error of the numerical integrator. In particular, the bounds are explicit in the chosen (time-inhomogeneous) noise schedule as well as the properties of the target distribution, providing insights into their impact. A series of numerical experiments on toy distributions show that the theoretical bounds can potentially guide the selection of noise schedules and data preprocessing.

**Audience:**

Yes

**Broader Impact Concerns:**

--

**Claims And Evidence:**

Yes

**Requested Changes:**

1. Comparison with existing bounds and related works. It would be good to add a discussion on what the novel proof techniques are, e.g., why the proofs (in particular for the KL divergence) do not follow from time-rescaling arguments, or more careful tracking of the involved quantities related to the noise schedule in previous proofs? Moreover, it would be good to also add sections like "Dissecting the upper bound" and "Comparison with existing bounds" for the $\mathcal{W}_2$-result.

2. The discussion in the last section should provide a more comprehensive summary and outlook.

3. Can one add a more relevant general target distribution in Section 5.2, e.g., a simple image dataset such as CIFAR-10?

4. Fixing the minor issues and comments mentioned above.

**Strengths And Weaknesses:**

### Strengths:

1. Refined theoretical analysis of diffusion models based on time-inhomogeneous VPSDEs, providing upper bounds for both the KL divergence and Wasserstein distance between the target and generated distributions that reveal the role of the noise schedule.

2. Empirical validation of the upper bounds through experiments on toy distributions, indicating potential benefits of optimized noise schedules and data preprocessing.

### Weaknesses:

1. **Limited discussion of novel techniques and related work:** The paper features a short discussion of the KL bounds and mentions that the $\mathcal{W}_2$-bounds leverage the contraction of the backward diffusion. However, apart from that, the main part of the paper does provide many details on comparing their proof techniques and contributions to previous work. Further comments on related works:
	- "Note that all the mathematical theory for diffusion models developed so far covers general time discretizations of time-homogeneous SGMs": What about the work by De Bortoli (2022), which is also cited several times in the manuscript? It would be good to add a corresponding comparison.
	- "an exact simulation can be performed for this process": It should be noted that this is well-known.
	- "The Euler Exponential Integrator (EI, see Durmus and Moulines, 2015), as already used in Conforti et al. (2023)": Note that it has also been proposed or used in https://arxiv.org/abs/2204.13902 and De Bortoli (2022).
	- "It has already been used by Conforti et al. (2023)": To some extend, the trick has also been used in https://arxiv.org/abs/2302.13834.
	- "In the literature, when $\beta(t)$ is constant equal to 2 (meaning that there is no time change), this diffusion process is referred to as the Variance-Preserving SDE": Song et al. (2021) already define the VPSDE for general, non-constant noise schedules. This is also noted later: "the schedule is linear and matches exactly the classical VPSDE implementation"

2. **Limited practical relevance of the empirical results:** While the experiments validate the theoretical results, there is still quite a large gap between relevant, high-dimensional problems and state-of-the-art implementations of diffusion models. It is unclear if and how the findings of this work would translate to practically relevant settings. Further comments on the empirical results:
	- Is there any explanation for the quite large gap to the actual KL divergence/$\mathcal{W}_2$-distance? Given this gap, one might ask why the tendency of the upper bound is still often reflected in the empirical results, which is crucial for claiming that better noise schedules are given by smaller upper bounds, e.g., in "upper bound remains clearly relevant to assess the efficiency of the noise schedule used during training".
	- "values for $a$ improving over the classical linear noise schedule": It should be emphasized earlier that the cosine schedule seems to be a classical, commonly used schedule for VPSDEs as well. Thus, it would also be interesting to see its performance in the plots of Figure 2.
	- "For backward process simulation, we use an Euler-Maruyama scheme with 500 steps, as being the most encountered discretization in practice": Did you also try the EI to follow the theoretical results more closely?
	- "the refinement relying on contraction arguments specific to the Gaussian setting (see (7)) is indeed more informative in such a case": could you show the corresponding bound in the plot to validate this statement?
	- "Moreover, implementing SGM with the optimal parameter a yields consistent improvements of the data generation quality across different metrics w.r.t. to classical noise schedule competitors (linear or cosine)": From Figure 4, it seems that the (untuned) cosine schedule would be approximately as good as the schedule with tuned $a$.
	- "significant improvement in the data generation quality": It seems to rather be a minor improvement.
	 - The experiments show that data preprocessing helps and the lower ratio of $L_0/C_0$ is offered as a potential explanation. I think it would be good to provide numerical evidence and present a plot of performance vs. ratio (for suitably rescaled target distributions).
	- It would be helpful to remove the dashed line in Figure 2 (and similar figures), since it seems not to add additional information but can be confusing to the reader. Moreover, it might be valuable to depict the dotted line (exact score evaluation) in a separate plot and compare against the corresponding upper bound for exact score evaluation.

3. **Limited scope:** The work seems to only cover noise schedules for VPSDEs but not for other SDEs which gained popularity, see, e.g., https://arxiv.org/abs/2206.00364.


**Additional minor issues:**

1. H3 should probably hold in each time interval $[t_k, t_{k+1}]$? Also, without a specific bound, I would not call it a "guarantor of a good approximation of the score by the neural network".
2. "However, knowing the score amounts to knowing the distribution at time t = 0": This seems not even sufficient, since the score cannot be obtained in a tractable fashion even if the density of the distribution at time t=0 was known.
3. "with $h$ [...] small enough": I think it would be helpful to provide an estimate on how small $h$ needs to be. Similar for Thm 4.2.
4. "If the result was derived for the EI discretization scheme": This sounds like the result has not been derived for the EI discretization scheme.
5. "and adjust schedules so that they all start at [...]": It would be good to specify how the schedule is rescaled.
6. "By considering early stopping techniques (typical in the literature on SGM), we could adapt these bounds to more general data scenarios (e.g., not strongly log concave)": How is this exactly meant?

**Typos:**

1. "converge exponentially fast": converges
2. "Isotropic, denoted by $\pi^{(\mathrm{corr})}\_{\mathrm{data}}$": should be $\pi^{(\mathrm{iso})}\_{\mathrm{data}}$
3. "Wasserstein distances provided by (...": Use `\citet` instead of `\citep`.
4. "on Gaussian mixture model": on *a* Gaussian mixture model
5. "multi-dimensional noise schedule": schedules
6. "particular interest what to deal with": how to deal with
7. "restricted to a compact,": restricted to a compact *set*?
8. "in top of": on top of
9. "in function of": as a function of
10. "for Wasserstein bound for": bounds

---

> ### Author Response · Authors · 2024-11-21
> **Answer to the "Requested changes" feedback**
>
> We thank the reviewer for his/her detailed feedback. We have taken into account his/her suggestions in the revised version. We provide specific answers in what follows. Please note that the general comment include also answer to his/her concern.
>
>
> **Requested changes**
>
> 1.
>
> - We agree that the discussion with existing works needed to be revised and completed. In particular, additional details on the novelty of our contribution and on the connection with existing hypotheses and results needed to be deepened.  First, the impact of the noise schedule on the KL term arises indeed from a more careful tracking of the involved quantities in previous works cited in the paper.  Theorem 3.1 is not our main contribution  and was established to highlight the impact of the noise schedule on the upper bounds obtained in previous works. We completed this first result with a refined analysis in the Gaussian case (Proposition C.1) to understand the sharpness of the first term of the upper bound.
>     We updated the "contribution" section of the introduction to insist on the fact that the KL result is a refinement of previous works focusing on the noise schedule. We believe that the detailed analysis in the Gaussian case (Proposition C.1) is interesting and points out natural assumptions to obtain the quantitative bounds in Wasserstein distance.
> - Our main contribution is the analysis for the Wasserstein distance. To derive the Wasserstein bound, the key challenge was to control the Wasserstein distance in a clear manner while incorporating the modified score function analysis, crucial to the analysis of noise schedule and its relationships with the other SGM parameters.
>
> - Moreover, taking into account the backward contraction for the diffusion process (Proposition C.2) provides state-of-the-art results on mixing time convergence for SGM under the Ornstein-Uhlenbeck forward process, whether in-homogeneous or not. This extends the similar result for the KL in the Gaussian setting. These  results are in the same line of work as \cite{bruno2023diffusion,gao2023wasserstein}, incorporating to the time in-homogeneous setting a refinement of the mixing time error based and an analysis of the modified score function.
>
> - These comments will be provided in the revised version of the paper for better clarity on the technical challenges. As suggested and in order to clarify this, we extended Section 4 with a paragraph  "Comparison with other works" in the revised paper and we detailed the "contribution part" of the introduction. We also discuss recent works such as \cite{benton2024nearly} and \cite{chen2023improved} with weak assumptions for the KL upper bounds to highlight possible perspectives to obtain tight Wasserstein bounds in other settings.
>
>
> 2.  The discussions (at the end of the paper and after the main results) were revised to take into account the proposed updates to answer all concerns and comments of the reviewers.
> 3. The revised version contains several additional experiments to illustrate the proposed upper bounds and to illustrate the impact of the noise schedule. In particular, as mentioned in the general response section, we will add in the revised version of the paper an additional experiment using CIFAR-10 to consider a more challenging setting.
> 4. All minor issues, typos,  weaknesses (in particular all precisions regarding the literature) and comments were taken into consideration in the revised version of the paper.

---

> > ### Author Response · Authors · 2024-11-21
> > **Answer to the "Weaknesses" part**
> >
> > **Weaknesses**
> > 1. *"For backward process simulation, we use an Euler-Maruyama scheme with 500 steps, as being the most encountered discretization in practice": Did you also try the EI to follow the theoretical results more closely?*
> >
> > We added a comparison of the empirical KL divergence  with the exponential integrator in Figure 9. The empirical performance are close to the classical Euler-Maruyama discretization scheme in the settings we considered which motivates the use of this scheme in the main paper in line with the vast majority of empirical works. As time was limited, we concentrated on the KL divergence but results for the Wasserstein distance are similar. We can also add illustration for the Wasserstein distance if you think it is necessary.
> >
> > 2. *"the refinement relying on contraction arguments specific to the Gaussian setting (see (7)) is indeed more informative in such a case": could you show the corresponding bound in the plot to validate this statement?*
> >
> > The first term of the error is now plotted along with the theoretical bound in the Gaussian case to highlight the additional contraction. We believe that this illustration highlights the interest of Proposition~C.1, we thank the reviewer for this suggestion.
> >
> > 3. *"Moreover, implementing SGM with the optimal parameter a yields consistent improvements of the data generation quality across different metrics w.r.t. to classical noise schedule competitors (linear or cosine)": From Figure 4, it seems that the (untuned) cosine schedule would be approximately as good as the schedule with tuned and
> > "significant improvement in the data generation quality": It seems to rather be a minor improvement.*
> >
> > In most cases, the tuned schedule performs better than the classical cosine schedule but it is true that this is a slight improvement. Therefore, we tempered our statement in the revised version. We would like to add that the cosine schedule was added as a baseline to display which performance state-of-the-art schedules would lead to. The numerical illustrations are used to highlight the fact that the upper bound can be used to tune the noise schedule but we do not claim that we aim at outperforming the cosine schedule, which, on another note, could also be optimized using our upper bound.
> >
> > 4. *"The experiments show that data preprocessing helps and the lower ratio of $L_0/C_0$ is offered as a potential explanation. I think it would be good to provide numerical evidence and present a plot of performance vs. ratio (for suitably rescaled target distributions)."*
> >
> > We agree with the reviewer that such a numerical evidence would be of interest. Unfortunately, we have not had the time to go into this aspect in detail. However, the initial submission already contained some elements of a response: For example, regarding the $L_0/C_0$ ratio, the difference in upper bound informativeness and data generation quality before and after rescaling was exemplified in the anisotropic case. Specifically, this is illustrated in Figure 13 (b) before rescaling ($L_0/C_0 = 100$) and in Figure 14 (d) after rescaling ($L_0/C_0 = 1$). As discussed in the main response area such result could in theory be propagated throughout the diffusion (see Figure 15).

---

> ### Comment · Reviewer_425e · 2024-11-23
>
> Thank you for your response and revision of the paper, which addresses parts of my concerns.
>
> The response raised the following questions, which seem not to be answered yet:
> 1. Could you provide a comparison to De Bortoli (2022), who also considers the time-inhomogenous case?
> 2. What are potential explanations for the large gaps between the theoretical results and the empirically observed metrics, even in toy examples?
> 3. Why is the performance of the pre-trained models on CIFAR-10 so much worse than reported in the EDM paper? Also, this section is currently not referenced or explained in the paper.
> 4. Could you comment on the relevance of the results for other processes, e.g., VE SDEs?

---

> > ### Author Response · Authors · 2024-11-26
> > **Further comments**
> >
> > Thank you for your questions. We hope the following responses provide clarifications and address any remaining concerns.
> >
> > 1. The article Convergence of Denoising Diffusion Models Under the Manifold Hypothesis (2022) by De Bortoli investigates convergence guarantees for VP SDE when the data distribution is supported  on a compact lower dimensional manifold (in this setting the target density is no longer absolutely continuous with respect to the Lebesgue measure). The main result provides an upper bound on the 1-Wasserstein distance between the generated and target distributions.
> >
> > This setting differs from our work and the other convergence results cited in our paper. Our results focus on $\mathcal{W}^2$ and Kullback-Leibler (KL) divergence bounds for target distributions that admit a density with respect to the Lebesgue measure, and importantly, our bounds do not display exponential dependency on some parameters. Moreover, it is true that De Bortoli (2022) models the forward process as a time-inhomogeneous Ornstein-Uhlenbeck process, but there is no explicit dependency of the upper bound on this schedule. Specifically, in Theorem 1 of their paper, the upper bound on the $\mathcal{W}_1$ distance between the target and generated distributions does not depend on the choice of the noise schedule throughout the diffusion process (i.e. on the values of $\beta_t$ for $t \in [0, T]$) but only on some bound on the initial ($\beta_0$) and terminal ($\beta_T$) values, see $\bf A2$. Our aim is to propose a bound as explicit as possible on the noise schedule dynamics to better understand its influence on the different terms of the bound. In this sense, although it is not yet possible to directly optimize our bound, we believe that our result offers interesting perspectives on noise schedule selection, which was not clear with previous works. This comment will be added in the revised version of our work in the paragraph "Discussion and comparison with other works" after Theorem 4.2.
> >
> > 2. We believe that the large gap may have several explanations. First, there is no guarantee that the proof technique, as any other theoretical work referenced in our paper, provides optimal constants. This proof highlights the impact of several approximations but obtaining lower bounds is a challenging task that should be explored to understand the optimality of the proposed upper bounds. In addition, some constants are obtained by considering a supremum over all discretization steps (see $\bf H6$ for instance). However, the behavior of these quantities may grow significantly for small time steps. A way to balance this detrimental impact would be to analyze early-stopping solutions to stop the backward process before reaching zero and optimally tune this stopping time.
> >
> > 3. The CIFAR-10 results we presented were preliminary and included before the discussion period to demonstrate our ongoing experimental efforts. These initial results represent the beginning of our study due to limited time to complete the experiments in the two weeks period.
> >
> >     We kept similar hyperparameters as those stated in Karras et al. (2022) : $T=1$, $\beta (0) =0.1$, $\beta(T) = 20$, $1000$ discretization steps and sampling over the diffusion $[\epsilon,1]$ with $\epsilon = 10^{-3}$. However, the sampling method is different as we rely on discretizing the backward stochastic process and not on ODE sampling.
> >
> >     In our preliminary experiments, the FID was calculated using only 1,000 generated samples for different choices of $a$ with exact same initialization, whereas Karras et al. (2022) report FID results based on 50,000 samples. Computing FID with a smaller number of samples can lead to very biased estimates (Chong et al, 2020) which explain the large gap when comparing FID scores.
> >
> >     We reproduced these results with an increased sample size of 50,000 to provide a better comparison. We updated the revised version of the paper (see Appendix G) so that you can check we reach performance similar to the baseline. Therefore, in the revised version (within the next few days), we will include all experiments with this set of hyperparameters (and with the cosine schedule) to illustrate that the behavior of the empirical bound follows the same dynamics as a function of the schedule as our toy examples although we cannot check that the assumptions on the data distribution are satisfied.
> >
> >  Depending on what the reviewers consider to be appropriate, we are prepared to include the numerical experiments on CIFAR-10 in the body of the paper by moving the (synthetic) Funnel case to the appendix.
> >
> > Karras, T., Aittala, M., Aila, T., and Laine, S. (2022). Elucidating the design space of diffusion-based generative models. arXiv. https://doi.org/10.48550/arXiv.2206.00364
> >
> > Chong, M. J., and Forsyth, D. (2020). Effectively unbiased FID and Inception Score and where to find them. University of Illinois at Urbana-Champaign. arXiv. https://doi.org/10.48550/arXiv.1911.07023

---

> > ### Author Response · Authors · 2024-11-26
> > **Further comments II**
> >
> > 4. Indeed, our result focuses on the VP SDE formulation of score-based generative models,  where the forward process is given by an Ornstein-Uhlenbeck process. This formulation is closely related to the Denoising Diffusion Probabilistic Model (DDPM) algorithm.
> >
> >     An alternative approach is VE SDE, where the forward process is defined by a scaled Brownian motion. This method has been empirically effective in practice. Theoretically, the main difference lies in the ergodicity of the forward process. In the VE SDE setting, the forward process is no longer ergodic. This means that we cannot initialize the backward process at the stationary distribution of the forward process; instead, the backward process is initialized at a centered Gaussian random variable with large variance.
> >     As a consequence, this affects the mixing time error in our theoretical analysis. Specifically, the mixing time error (for both $\mathrm{KL}$ and $\mathcal{W}_2$ bounds) in the VE SDE case is expected to decay approximately at a rate of  $(\int_0^T \beta(s) d s)^{-1}$ rather than exponentially fast, as it does in the VP SDE setting.
> >
> >     Finally, while our current results are derived for the VP SDE formulation, we anticipate that similar analytical techniques could be extended to VE SDEs. In particular, this appears to be straightforward for the $\mathcal{W}_2$ bound but may be more challenging for the KL bound, in particular, as it requires careful handling of Lemma B.3, where Lemma B.7 is applied to the Ornstein-Uhlenbeck process, and the cancellation of the backward process $\overleftarrow{X}_t$ does not occur.   This comment will be added in the revised version of our work.

---

### Review · Reviewer_goCE · 2024-11-03

**Summary Of Contributions:**

The paper studies score-based generative models theoretically, by presenting upper bounds on the error committed by these models in the estimation of the target measure. The emphasis is put on how the noise schedule chosen for the model affects these bounds.
First, the authors analyse the error using the Kullback-Leibler (KL) divergence and generalise the recent results of Conforti et al. (2023) to find an upper bound for a generic noise schedule.
Then, using the Wasserstein distance, a new upper bound is derived for score functions satisfying some additional regularity assumptions.
Finally, the paper presents numerical simulations on some relevant probability distributions, for which the empirical upper bounds can be computed and compared to the actual values of the error.

**Audience:**

Yes

**Broader Impact Concerns:**

No ethical concerns.

**Claims And Evidence:**

No

**Requested Changes:**

- The "contraction" argument in Proposition D.1, used to prove Theorem 4.2, is similar to the one used in "Chain of Log-Concave Markov Chains" by Saremi et al. (https://arxiv.org/abs/2305.19473) to prove Theorem 1. Therefore, in my opinion, the sentence after Theorem 4.2 that claims "this feature has never been considered" should be revised.
- (See weakness b): Section 5 needs major revision. I would advise to focus more on the theoretical result of section 4 and less on how much the result itself is of practical interest, unless more convincing numerical results can be found.

**Strengths And Weaknesses:**

**Strengths:**
- The paper is clearly written and well-structured.
- Sections 1 and 2 introduce well the framework and the notation used in the rest of the paper.
- The Wasserstein bound presented in Theorem 4.2 is interesting and, as far as I know, a novel contribution to the field.

**Weaknesses**
- (a) The derivation of the Bound on the KL divergence presented in Section 3 follows step by step the one of Conforti et al. (2023). It looks to me that keeping the function $\beta(t)$ generic does not require any additional technical step in the proof. Additionally, from the expression given in Theorem 3.1 it seems that $\beta(t)$ does not play an important role in the bound. This makes the result a very marginal contribution in my opinion.
- (b) The aim of section 5 is supposedly to "numerically illustrate the validity of the bounds" obtained in the previous sections. However, I find the results presented rather inconclusive: First, as can be seen from the left y-axis in Figs. 2 and 4, the upper bounds are always at least one order of magnitude larger than the numerically calculated values. The authors still claim that the bounds are "clearly relevant" and that they "align with the empirical performance" because taking the value of $a$, called $a^*$, that minimises the bound and calculating the error from the resulting noise schedule leads to better performance than typical noise schedules used in practice.
However, I think it should be noted that in all cases analysed in the paper, $a^*$ is almost the same (between 1 and 5), regardless of the data set considered. Moreover, looking at the results in Tables 1 and 2, the performance of the $a=a^*$ schedule is similar to that of typical schedules within the standard error deviations, which means that the cases in which it seems to perform better could just be due to the limited statistics of the experiments (from Fig.3 it seems that only 10 experiments were performed for each point).

---

> ### Author Response · Authors · 2024-11-21
>
> We thank you for your feedback, we provide a detailed answer in what follows. Please note that the general comment includes also answers to your concerns.
>
> **Requested changes**
>
> 1. We thank the reviewer for providing this reference as we were not aware of this result (our result was obtained simultaneously and independently). We therefore removed our statement, and clarified the fact that a similar result (propagation of the log-concavity) was obtained in Saremi et al. (https://arxiv.org/abs/2305.19473). We kept the proof in the revised version for the sake of completeness, since this is a technical result of the appendix and not an important result of the article.
>    We would like to emphasize that this is an auxiliary result of our contribution, serving only to illustrate how H4 can be verified in simple settings which is now stated more clearly.
>    To clear up any misunderstanding, we would like to point out that the backward contractivity argument (Prop. C.1 for KL and C.2 for W2, Eq. (35)) remains new to our knowledge, and allows generic improvement of mixing time errors.
> 2. We agree that a revision of the  numerical section was required. In the revised version of the article, we therefore proposed additional simulations and clarifications of the simulations displayed in the original article.
>     - The aim of the simulations in Tables 1 and 2 was to illustrate how the theoretical results could be used to tune the noise schedule in a proposed parametric family and in a simple setting where the upper bound can be evaluated. In this perspective, we have therefore proposed a parametric family encompassing different types of noise schedule behaviors. We added the cosine schedule for comparison with a schedule used in state-of-the-art implementations. To complete these results, we present the statistics with 30 independent experiments in Tables 1 and 2 of the revised version. To complete these results, we also illustrate the contribution of the different terms in the upper bound in Figure 11 of the revised version.
>     - We would also like to stress that the phenomena identified in the theoretical bounds are not disconnected from observations in practice. Indeed, we illustrate in the case of the KL bound (see Figure 2, top) the different upper bounds with or without taking into account the backward contractivity. The contraction argument turns out to be crucial to properly align the theoretical bound with the observed empirical performance.
>     - We added a comparison of the empirical KL divergence  and Wasserstein distance with the exponential integrator in Figure 9. The empirical performance are close to the classical Euler-Maruyama discretization scheme in the settings we considered which motivates the use of this scheme in the main paper in line with the vast majority of empirical works. As time was limited, we concentrated on the KL divergence but results for the Wasserstein distance are similar. We can also add illustration for the Wasserstein distance if you think it is necessary.
>     - As mentioned in the general response section, we will include in the revised version of the paper an additional experiment on CIFAR-10 to illustrate the impact of the noise schedule in a more challenging setting. In this context, the upper bound cannot be evaluated but the proposed empirical results support the impact of the noise schedule and suggest that optimizing the noise schedule is an interesting research topic.

---

> > ### Author Response · Authors · 2024-11-21
> > **Review (Weaknesses part)**
> >
> > **Weaknesses**
> >
> > - It is true that the derivation of the bound on the KL divergence presented follows the one of Conforti et al. (2023). The proof technique is similar but we believe that detailing precisely how the noise schedule can be traced is crucial to understand both practical and theoretical aspects of these algorithms.
> >
> >   Still, we had to establish the KL upper bound for an inhomogeneous forward diffusion which involved determining a non-asymptotic rate of convergence for the mixing time using Fokker-Planck equations and a log-Sobolev inequality that depends on the noise schedule and not only on the diffusion time horizon.  Although this result (Lemma B.1) is not the most interesting part of our contribution, this was not established in Conforti et al. (2023). Our initial objective was to highlight the impact of the schedule on the KL control and to propose an upper bound for schedule optimization. However, we agree that the main result of the paper is Theorem 4.2 to establish theoretical guarantees in terms of Wasserstein distance.
> >
> >   In the revised version of the paper, this main contribution is highlighted and we clarify the fact that the KL control was a natural first step to establish additional results with other metrics. Thanks to your remarks and the other reviews we detailed this in the "contribution" section of the introduction and on the discussion on the Wasserstein bound in Section~4.
> >
> >   On the other hand, we do not agree that the noise schedule "does not play an important role in the bound" and believe that the additional simulations  provided in the revised paper (both for KL and Wasserstein distances) support the impact of the schedules on the performance of score-based generative models (and motivates research perspectives such as online optimization of noise schedules).
> >
> >   We would also like to add that Proposition C.1 on the contraction of the KL between $\pi_{\mathrm{data}}$ and $\pi_\infty Q_T$ in the Gaussian case was not established in previous works and is an interesting result providing insights for the Wasserstein bounds requiring extra regularity of the score.
> >
> > - Weakness (b). See the proposed changes in "requested changes".

---

### Review · Reviewer_cnqp · 2024-11-07

**Summary Of Contributions:**

The paper derives upper bounds for the KL divergence and 2-Wasserstein distance between the target and estimated distribution of a time-inhomogeneous score-based generative model which explicitly depend on the noise schedule. The numerical experiments show relationships between the choice of noise schedule and quality of the estimated model which align with those suggested by the bounds. The authors also use the upper bounds to propose an alternative noise schedule (based on what would approximately minimize the upper bound) which seems to outperform common noise schedules in synthetic experiments.

**Audience:**

Yes

**Broader Impact Concerns:**

I see no ethical concerns with the work.

**Claims And Evidence:**

Yes

**Requested Changes:**

For major changes, please see the weaknesses above. More broadly, the paper could benefit from further discussion / analysis around the main theoretical results.

Other minor notes:
1. There's an unintentional paragraph break under the paragraph titled "Comparison with existing bounds."
2. Several statements in the introduction could be made more precise, including "It is crucial to note that the complexity of real data prohibits a thorough depiction of the distribution $\pi_\text{data}$ through a conventional parametric model, and its estimation via traditional maximum likelihood methods" and "However, knowing the score amounts to knowing the distribution at time $t = 0$, i.e., knowing the distribution $\pi_\text{data}$ according to which we wish to simulate new examples." Namely, in the former it may be too strong to say "prohibitive" given the empirical success of many generative models that are directly trained via maximum likelihood; in the latter, "knowing the distribution" is a bit imprecise.

**Strengths And Weaknesses:**

Strengths:
1. The contributions, assumptions, and analysis are clear.
2. The paper is well-situated in the related work and extends it significantly.
3. The numerical experiments provide some hints into the potential usefulness of the derived bounds.

Weaknesses:
1. The paper would benefit from expanding the discussion with the practical benefits gained from deriving these tighter theoretical bounds (such as some which have been hinted at in the experiments section).
2. The upper bound is still very loose relative to the exact empirical computations, e.g. Figure 2.
3. The paper suggests a potential improvement to the noise schedule using the upper bound but does not demonstrate whether this schedule has practical benefit in more real-world settings. Even in the synthetic setting, though, the improvement seem minimal at best.

---

> ### Author Response · Authors · 2024-11-21
>
> We thank you for your feedback, and provide point-by-point answer to the weaknesses you mention.
>
> 1. Our first motivation was to provide a rigorous theoretical framework for the introduction of a noise function into diffusion-based generative models. Indeed, this noise function has been identified as a game changer for efficient learning, and has so far only been addressed through empirical intuition.
>  Obtaining a theoretical bound on the quality of generated samples that depends on the noise function finally enables us to guide learning by quantifying the impact of choosing one function over another on the different types of error characteristic of SGMs.
>
>
> 1 and 2.  In the revision, greater emphasis is now placed on the fact that even if the numerical constants of the established bounds appear suboptimal (as evidenced by the discrepancy in order of magnitude between the left-hand and right-hand terms of the bounds evaluated in numerical experiments), the assessment of the bound remains informative for arbitrating two denoising functions.
>
> 3. In the synthetic case, the least trivial example is undoubtedly that based on the Funnel distribution in dimension 50: in Figure 4, the impact of the noise function is visible on the quality of the samples produced, leading us to consider noise schedules with a parameter $a$ around 5.
> To address your concern, we have also initiated new numerical experiments on CIFAR, please refer to the general comment for further details.
>
>
> Requested changes: We also included your minor suggestions.

---

### Author Response · Authors · 2024-11-21

We sincerely thank you for your time and feedback on our paper, which helped us improve our work.
We provide point-by-point answers in the individualized comment sections.
We would like to inform all the reviewers that we took into account your remarks, and modified the paper accordingly. We also expanded the discussion section at the end of the paper and after the main results.

In addition, we conducted as many new numerical experiments as we could within the allotted time. More precisely, we performed the following additional simulations.

1. For the KL bound, we repeated the initial simulations in the Gaussian case over 30 iterations (compared to 10 in the submitted version), as shown in Figure 2 (top). This does not affect the conclusions drawn from the previous results with fewer repetitions. If required by the reviewers, we can extend this approach to include 30 iterations for the other experiments presented in the paper.

2. For the KL and Wasserstein bounds, we provided the simulations in the Gaussian setting using the Exponential Integrator (EI) instead of the Euler-Maruyama discretization scheme (see Figure 9), while maintaining the same number of discretization steps (500). As expected, given the small step size, both discretization techniques lead to comparable results.

3. For the KL bound, we now illustrate the different upper bounds with and without accounting for the backward contraction (Proposition C1, Equation (7)), as shown in Figures 2 (top) and 9 (top). The contraction argument --identified for the first time in this paper to our knowledge-- proves to be essential for aligning the theoretical bound with the observed empirical performance.

4. For the KL bound, in the Gaussian setting, we dissect the different error terms (approximation and mixing time with and without contraction), with respect to the different noise schedules, see Figure 11. This highlights more specifically the role of each error term in the comparison of different schedules from the family $\beta_a$. (Note, that the discretization error has not been plotted as the $\beta_a(T)$ is set to 20 for every $a$.)

5. Regarding the discussion on data conditioning and the resulting $L_0/C_0$ ratio, the difference in upper bound informativeness and data generation quality before and after rescaling is exemplified in the anisotropic case. Specifically, this is illustrated in Figure 13 (b) before rescaling ($L_0/C_0 = 100$) and in Figure 14 (d) after rescaling ($L_0/C_0 = 1$). Such a ratio can, in theory, be propagated through the forward process, given that the expressions for the strong concavity and Lipschitz continuity of the score function for $t \in (0,T]$ can be derived analytically. In particular, in the anisotropic case, for a linear schedule ($a=0$) the corresponding graph would look like Figure 15 in the revised version.

6. To address your concerns regarding real dataset, we suggest to evaluate the FID score for different noise schedules (different values of $a$ in $\beta_a$, see Figure 1) on CIFAR 10.
The obtained results are described in Tables 4 and 5 in the revised version, showing the impact of the noise schedule on the quality of generated images.
Regarding the implementation, we used pretrained models from \cite{karras2022edm} with the recommended hyperparameters designed to replicate the experiments in \cite{song2021score} corresponding to our linear schedule ($a=0$) as shown in Figure 1. \cite{karras2022edm} also suggest to use 1000 discretization steps for the backward process.  It is worth noting that the training process in \cite{karras2022edm} is slightly different, though equivalent, to the original implementation. In particular, the networks are not trained to directly estimate $\nabla \log p_t ( \overrightarrow X_t)$. Instead, a denoiser function $D_{\theta}(X,\sigma)$ is trained to isolate the noise from the signal for some noise level (see Equations (2) and (3) in \cite{karras2022edm}). With appropriate rescaling this could in fact be used in the VP setting by letting
$$ s_{\theta}(\overrightarrow X_t,t) = \frac{\sigma_t^2}{m_t}\left( D_{\theta}  \left(\frac{\overrightarrow X_t}{m_t} , \frac{\sigma_t}{m_t}\right) - \frac{\overrightarrow X_t}{m_t} \right),  $$
where $s_{\theta}$ is the score approximation as defined in our paper, $m_t = \exp\{-\int_0^t \beta(s) \rm d s/(2 \sigma^2)\}$ and $ \sigma^2_t = \sigma^2 (1-m_t^2)$. This formulation bridges the denoising approach with score-based methods in the VP framework.

---

### Decision · Action_Editor_MbDu · 2024-12-16

**Recommendation:** Accept as is

**Comment:**

After careful consideration of the reviews, the authors' responses, and the revised manuscript, my recommendation is for acceptance of the revised manuscript.

The paper presents clear and rigorous theoretical contributions, particularly in deriving novel upper bounds for Wasserstein distance and exploring the role of noise schedules in score-based generative models. While the work's practical relevance remains limited, the authors have thoroughly addressed reviewer concerns, providing additional experiments and clarifications that enhance the manuscript.

Importantly, all reviewers ultimately leaned toward acceptance, recognizing the paper's value to the theoretical understanding of score-based generative models.

**Audience:**

Score-based generative models is a topic of major interest to the TMLR community, and the theoretical insights concerning  the role of noise schedules will be of interest to the readers.

**Claims And Evidence:**

The work provides new insights into the role of noise schedules in score-based generative models. The results are supported by rigorous mathematical analysis, particularly the derivation of upper bounds for KL divergence and Wasserstein distance.

Some concerns were raised by the reviewers:

- The derivation of the KL bound closely follows prior work (e.g., Conforti et al., 2023). The authors acknowledged this and clarified that the KL bound serves as an auxiliary result rather than a primary contribution.

- The gap between theoretical bounds and empirical metrics remains significant, limiting the practical relevance of the results. The authors addressed this by tempering their claims and providing additional experiments.

In summary, while the claims are mathematically solid, the empirical evidence supporting their practical impact is less robust. The authors have acknowledged these limitations and made meaningful revisions.